# CREA: A Collaborative Multi-Agent Framework for Creative Image Editing and Generation

**Kavana Venkatesh**[*]     **Connor Dunlop**[*]     **Pinar Yanardag**
Virginia Tech, Blacksburg, VA
{kavanav, cdunlop, pinary}@vt.edu
https://crea-diffusion.github.io

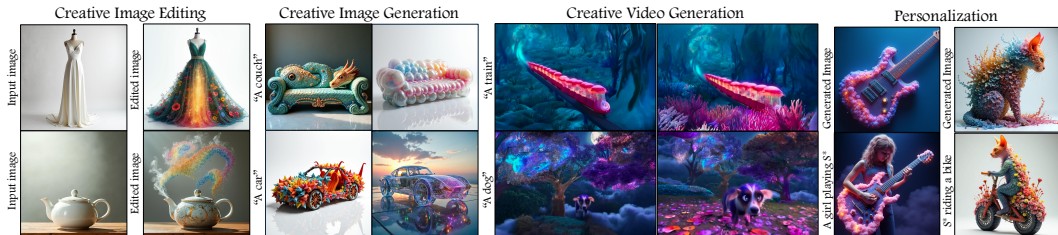

Figure 1: We introduce CREA, an agentic framework inspired by the human creative process for image editing and generation. Our approach can be extended to video domain for creative video generation or can be integrated with personalization methods to further enrich creative workflows.

## Abstract

Creativity in AI imagery remains a fundamental challenge, requiring not only the generation of visually compelling content but also the capacity to add novel, expressive, and artistically rich transformations to images. Unlike conventional editing tasks that rely on direct prompt-based modifications, creative image editing requires an autonomous, iterative approach that balances originality, coherence, and artistic intent. To address this, we introduce CREA, a novel multi-agent collaborative framework that mimics the human creative process. Our framework leverages a team of specialized AI agents who dynamically collaborate to conceptualize, generate, critique, and enhance images. Through extensive qualitative and quantitative evaluations, we demonstrate that CREA significantly outperforms state-of-the-art methods in diversity, semantic alignment, and creative transformation. To the best of our knowledge, this is the first work to introduce the task of creative editing.

## 1   Introduction

Generative AI has significantly transformed the field of image generation, producing high-quality visuals with remarkable detail and realism. Advances in diffusion models [42, 14], GANs [26], and retrieval-augmented techniques [5, 49] have enabled powerful capabilities in content synthesis, making AI-driven image creation an essential tool for artists, designers, and various creative industries. These models have been widely applied in diverse tasks, including image-to-image translation [24, 38, 47], inpainting [42, 33, 25], style transfer [26, 18, 44, 61], and content-aware editing [36, 8, 7, 12], and opening up new frontiers in digital art, advertising, and entertainment, revolutionizing creative workflows.

Despite these advancements, achieving creative and artistically rich compositions still demands significant user effort. Traditional generative approaches primarily follow a prompt-to-image paradigm,

---

[*]Equal contribution.

39th Conference on Neural Information Processing Systems (NeurIPS 2025).

where models synthesize high-quality visuals based on textual descriptions. However, these outputs often lack originality and artistic depth, as the models are trained to replicate patterns from their training data rather than generate novel content. As a result, users must engage in tedious prompt refinements, fine-tune model parameters, or manually edit outputs to infuse genuine creativity. This heavy reliance on user expertise and intervention limits the accessibility of generative AI as an autonomous creative assistant, placing the responsibility for creativity more on the user than on the system itself.

To address this limitation, we propose the task of creative image editing, where the goal is to modify images in a way that enhances their creative and artistic qualities with minimal user intervention. Unlike conventional editing tasks that focus on making explicit, text-driven modifications, creative editing aims to transform images into novel, aesthetically rich compositions in a disentangled way. Our approach mimics the complex process of creative image generation and editing as a collaborative team effort, drawing inspiration from human workflows where specialists iteratively refine ideas to achieve a shared artistic vision. To achieve this, we introduce a novel collaborative multi-agent framework, where agents, each with a distinct role, such as a *Creative Director* or *Art Critic*, work in synergy to conceptualize, generate and refine creative outputs by grounding them in well-established guidelines distilled from established principles in creativity literature [6, 20, 2, 40]. This structured yet flexible approach enables the disentangled creation of highly diverse and imaginative images, ensuring both novelty and coherence at every stage of the generation process. Since creativity is a complicated task to assess and evaluate, our agents leverage the extensive knowledge in multimodal LLMs [1] based on state-of-the-art creativity principles inspired by previous work [6, 20, 40, 2, 4]. Through both qualitative and quantitative evaluations, we demonstrate that our method consistently produces edits perceived as more creative and aesthetically pleasing compared to baseline methods. Our contributions are as follows:

- We introduce a novel agentic framework for the task of creative image editing and generation. To the best of our knowledge, this is the first work to introduce the task of 'creative *editing*'.

- We incorporate a user-in-the-loop generation setting, enabling user guidance to steer the creative direction through optional interventions. This supports a more collaborative human-AI co-creation process while maintaining artistic coherence and control.

- We demonstrate the versatility of our method across image editing and generation, as well as its potential for enhancing personalization workflows and creative video generation.

## 2 Related Work

**Text-to-Image Models.** Recent advances in diffusion models [23, 42, 45] have revolutionized text-to-image synthesis, enabling high-fidelity image generation guided by textual prompts. Models such as DALLE-3 [37], SDXL [39], and Flux [27] demonstrate the ability to generate visually compelling images based on textual prompts [55, 53, 30]. While these models produce high-quality outputs, they lack a structured mechanism to enforce creativity principles in generation. Personalized generation approaches such as DreamBooth [44] and Textual Inversion [18] focus on fine-tuning text-to-image models for specific subjects or styles. However, these methods optimize for style and subject consistency rather than generating creative images.

**Creative Image Generation and Editing.** Research on creative generation in AI has advanced through GANs [19, 26] and diffusion models, leveraging contrastive loss and diversity-based objectives to encourage novel synthesis beyond category constraints [13, 22, 46]. Recent works such as ProCreate [31] use energy-based repulsion to steer diffusion models away from reference images, while Inspiration Tree [50] employs hierarchical decomposition for conceptual hybridization. ConceptLab [41] tackles creative concept synthesis through diffusion priors, iteratively enforcing constraints to generate novel category members. C3 [21] proposes a training-free approach for enhancing creativity by amplifying low-frequency feature maps in early layers of Stable Diffusion models. Despite their advancements, these methods require expensive retraining or optimization, and fail to generalize to broader concepts. For image editing, ControlNet [59] extends Stable Diffusion, introducing external conditioning signals for localized modifications while maintaining structural integrity. SDEdit [36] and Blended Latent Diffusion [3] refine diffusion-based editing for finer control, while InstructPix2Pix [8] allows text-based transformations via user prompts. However, these approaches focus on structural fidelity rather than creativity-driven, minimal-guidance transformations.

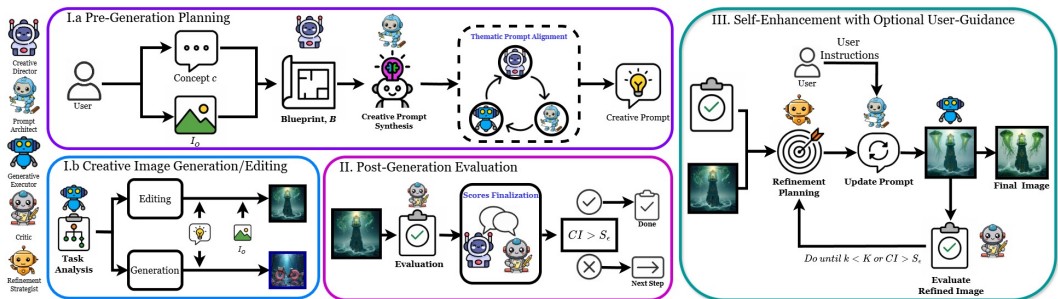

Figure 2: **CREA Framework**. We introduce a collaborative multi-agent framework for creative image editing and generation. Our framework consists of four stages, 1.a Pre-Generation Planning, 1.b Creative Image Generation/Editing, 2. Post-Generation Evaluation and 3. Self-Enhancement. Here, K is the number of maximum iterations.

**Large Language Models** Large Language Models (LLMs) such as GPT-4 [1] and PaLM [10] have demonstrated remarkable capabilities in natural language understanding and generative tasks. These models leverage transformer architectures [48], which allows them to process large amounts of text and generate coherent and contextually relevant outputs. In creative domains, LLMs have been increasingly integrated into AI-driven artistic workflows, assisting with idea generation, structured prompt synthesis, and artistic guidance [35]. GenArtist [52] uses a multimodal single agent for general image generation and editing but does not address creative image editing and fails to leverage the collaborative nature of multiple agents for complex use cases. Several studies have explored LLMs in the context of creativity [32, 62, 29, 11, 28, 16]; however, to the best of our knowledge, creative editing tasks have not been explicitly tackled before. Additionally, agentic frameworks, where AI systems exhibit autonomy in decision-making, iterative refinement, and adaptive goal-setting have not yet been systematically explored in the context of creative image generation or editing.

## 3 Methodology

Creativity is a multifaceted process that spans ideation, critique, refinement, and the delicate balancing of novelty with coherence. Traditional prompt-to-image pipelines handle this spectrum in a single static step, leaving users to face the burden of crafting elaborate prompts, testing variations, and manually disentangling edits. Such manual workflows are both expert-intensive; they assume deep domain knowledge to articulate abstract creative facets, and time-consuming, often requiring many trial-and-error iterations before the desired concept emerges. Our agentic framework replaces this bottleneck with a dynamic, collaborative loop that coordinates specialized AI agents such as a creative director, art critic, and refinement strategist to reason jointly and iteratively evolve an image. By structuring creativity as a multi-agent dialogue, the system mirrors human studio practice: each role contributes focused expertise while shared memory maintains global coherence. The result is higher-quality, more interpretable edits achieved with less human effort and lower computational overhead than a manual, prompt-tweaking regimen. Fig. 2 offers an overview, and the sections that follow detail each component of the framework.

### 3.1 Multi-Agent Framework Design

Our workflow mirrors structured human collaboration, assigning distinct agent roles to ensure iterative, controllable creative image editing and generation where agents are:

**Creative Director** $\mathcal{A}_1$**.** This agent serves as the main decision-maker, interpreting concepts, defining a creativity blueprint, and coordinating with agents to refine, discard, or finalize the image.

**Prompt Architect** $\mathcal{A}_2$**.** Translates conceptual ideas into contrastive prompts [9] for each creativity principle, merging them into a high-creativity prompt for the *Generative Executor* agent for image editing/generation and refining them iteratively based on agent feedback.

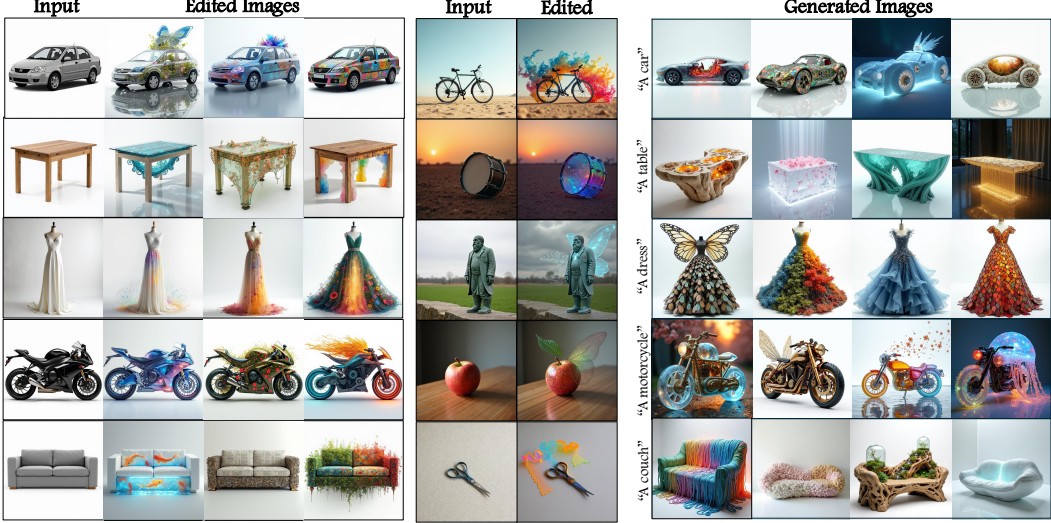

Figure 3: **Qualitative Results for Creative Image Editing and Generation Tasks.** (a) **Creative Image Editing:** CREA takes either a real-world or AI generated input image to produce a conceptually enriched and stylistically novel edit while preserving key structural elements. (b) **Creative Image Generation:** CREA receives only a minimal concept description (e.g., "a couch" or "a car") and generates diverse, imaginative outputs without any visual input, extrapolating rich visual metaphors and materials. These results demonstrate CREA's ability to disentangle and control creativity across editing and generation workflows. For additional results, see the appendix section.

🦾 **Generative Executor** $A_3$**.** This agent uses T2I models such as Flux and ControlNet for image generation or editing, dynamically selecting the appropriate diffusion model and parameters to ensure creative outputs.

🤖 **Art Critic** $\mathcal{A}_4$**.** The Art Critic evaluates the generated image based on the creativity principles, assigning a score for each criterion. Given that LLMs can approximate human judgment in subjective evaluations [63], this agent uses a multimodal LLM judge to ensure sensible evaluation.

🐵 **Refinement Strategist** $\mathcal{A}_5$**.** This agent translates the Critic's feedback into actionable refinements for the next iteration. It identifies weak creative dimensions and suggests precise modifications to the *Prompt Architect*.

Each agent maintains a private memory and utilizes role-specific tools to share task status and relevant information, ensuring informed decision-making at every step. We acknowledge that an agentic pipeline introduces extra orchestration effort compared with a naive single-pass T2I call. However, an agentic design is critical for several aspects (i) modularity - each agent encapsulates a narrow skill (prompt refinement, consistency auditing, local editing) that can be swapped or upgraded independently; (ii) interpretability - inter-agent messages expose an explicit reasoning trace; and (iii) extensibility - new capabilities (e.g., style transfer, safety filtering) may be added to expand the scope of the framework. See Supplementary Material for detailed agent compositions.

## 3.2 Collaborative Agentic Synthesis

Given a user-provided concept $c$ (such as 'a guitar' and an input image $I$, our goal is to modify it to generate $I_c$ in a creative and disentangled manner. For image generation, we follow the same pipeline with minimal modifications to produce a creative image $I_0$ and transform it into $I_c$. We formulate this as an optimization problem that maximizes a Creativity Index (CI), guided by six creativity principles described below. Our method has three modular phases: *Pre-Generation Planning*, *Post-Generation Evaluation*, and *Self-Enhancement with optional User-Guidance*. For clarity, the following sections primarily describe our method in the context of creative editing, as the same framework extends naturally to image generation with minor adjustments.

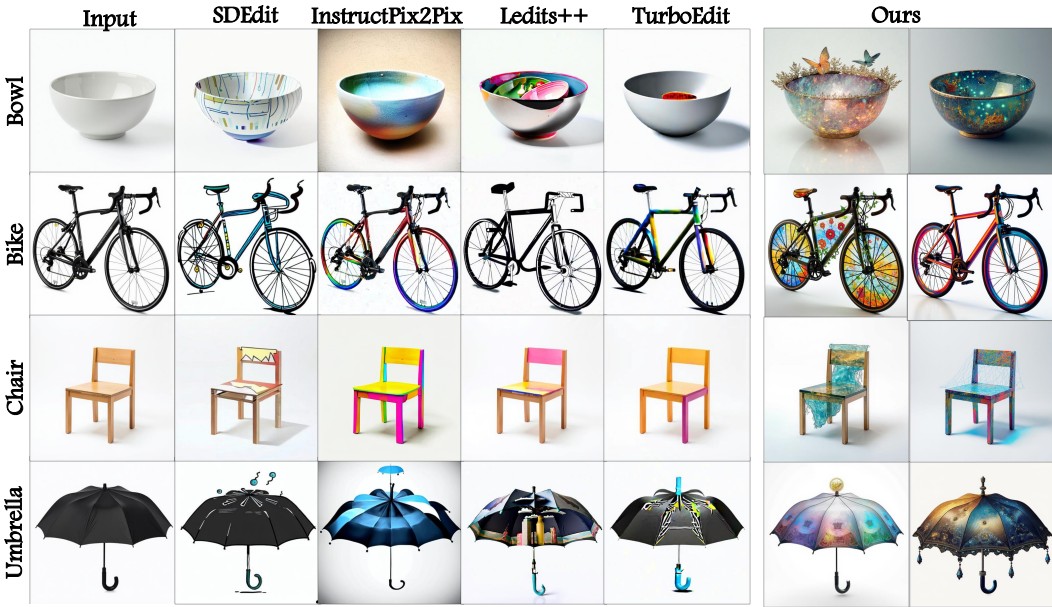

Figure 4: **Qualitative Comparison of Creative Image Editing Task.** We compare CREA with state-of-the-art editing methods. As shown, CREA successfully reimagines objects into creative variants in a disentangled manner, whereas other approaches either fail to produce distinctly creative edits or introduce unintended alterations.

**Creativity Principles**. Our framework leverages six creativity principles, grounded in state-of-the-art creativity theories, to systematically evaluate and measure creative output. *Originality*, measuring novelty and uniqueness, is inspired by Boden's Theory of Creativity [6] and Guilford's Divergent Thinking [20] framework. *Expressiveness*, which captures emotional impact, is influenced by Amabile's Model of Creativity [2] and Ramachandran & Hirstein's Laws of Aesthetics [40]. *Aesthetic Appeal*, assessing composition and harmony, is grounded in Martindale's Aesthetic Model [34] and Berlyne's Aesthetic Theory [4]. *Technical Execution*, evaluating craftsmanship and skill, draws from Amabile's Model and AI Creativity Frameworks [2]. *Unexpected Associations*, reflecting surprise and ingenuity, is supported by the Geneplore Model [15] and Boden's Combinational Creativity [6]. Finally, *Interpretability & Depth*, which considers exploration potential, is informed by Ramachandran's Laws [40] and the Geneplore Model [15] (see Appendix for more details). These six principles are used as a creativity template $T$ for agents to assess and refine the outputs.

## 3.3  Pre-Generation Planning and Image Synthesis

The pre-generation planning phase serves as a structured ideation stage where agents collaboratively establish a creative blueprint, $B$ before image generation begins. This phase involves three key agents: the Creative Director $A_1$, Prompt Architect $A_2$, and Generative Executor $A_3$, who collaboratively devise the creativity prompt $P_c$ that is both creatively rich and technically viable. First, *Creative Director*, $A_1$ interprets the initial image $I_0$, either user-provided or generated using the user-provided concept $c$ to formulate a creativity blueprint $B$, capturing the core theme, stylistic interpretation, visual structure, and necessary constraints to balance artistic flexibility with semantic coherence (see Appendix for more details). Based on $B$, the *Prompt Architect $A_2$* synthesizes a set of contrastive prompts $P = \{p_1, p_2, \ldots, p_6\}$, each conditioned on a distinct creativity principle and are merged into a high-creativity fused prompt $P_c$ through Chain-of-Thought [54] reasoning $P_c = \text{CoT-Fusion}(p_1, p_2, \ldots, p_6)$ where CoT-Fusion extracts salient conceptual and stylistic attributes from each prompt and synthesizes them into a coherent, balanced, and conceptually rich formulation. The *Generative Executor $A_3$* evaluates the feasibility of $P_c$ and determines T2I model-specific constraints-such as ControlNet conditioning scale, image guidance scale to anticipate the nuances of the generated blueprint. Once all agents reach a consensus, the prompt moves to the next phase for creative image generation.

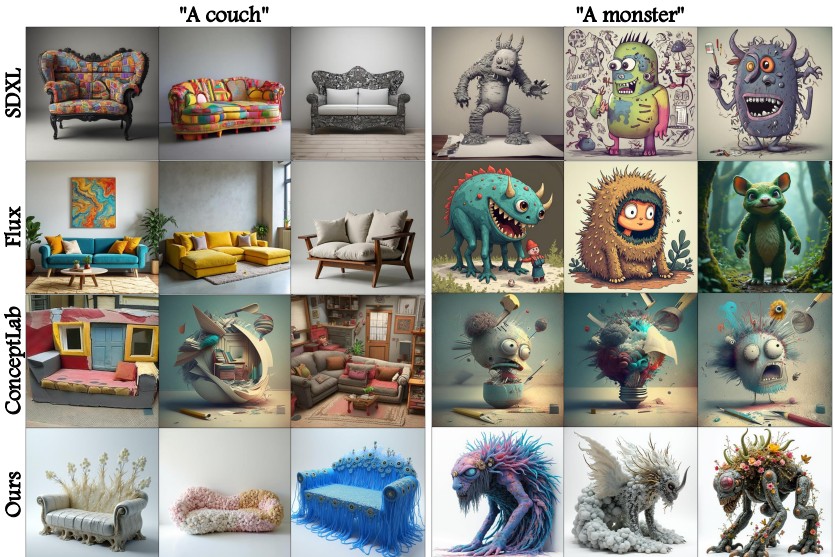

Figure 5: **Qualitative Comparison of Creative Image Generation Task.** We compare CREA with ConceptLab, SDXL and Flux. CREA consistently produces diverse and creative generations across multiple domains.

**Image Generation:** The *Generative Executor*, $A_3$ plays a dual role, either generating a creative image using a T2I model or performing disentangled creative editing on an existing image, whether initially generated ($I_0$) or user-provided ($I_0$). The high-creativity structured prompt formulated during the pre-generation planning phase is then taken by the *Generative Executor* to synthesize the initial creative image, $I_0$ using a text-to-image diffusion model, $G$ as $I_0 = G(P_c, \theta)$ where $\theta$ represents model-specific parameters-such as guidance scale and ControlNet conditioning scale. The generated image, $I_0$ serves as the starting point for the creative editing and iterative refinement process as described in the following sections.

**Image Editing:** If the user provides an image instead of generating one, or if an enhancement to $I_0$ is required, the *Generative Executor* performs disentangled creative edits using $I_e = G(P_c, I_0, \theta)$ where $G$ is the ControlNet model used to perform disentangled edits, $P_c$ is the high-creativity editing prompt generated in the pre-generation planning phase and $\theta$ represents parameters of $G$. After generation or editing, the resulting image progresses to the post-generation evaluation phase to investigate if further iterative refinements are necessary.

### 3.4 Post-Generation Evaluation

The post-generation evaluation assesses the edited creative image, $I_e$ (or $I_0$ for generation) against the creativity template, $T$ to maximize creativity. This Critic $A_4$ and the Creative Director $A_1$ collaborate, with the *Critic* systematically evaluating the initial edited image, $I_e$ based on the creativity template, $T$. The *Critic* utilizes the LLM-as-a-Judge to evaluate the edited image $I_e$ by assigning a creativity score, $S_i$ for each of the six $i^{th}$ creativity criteria on a 1-5 scale. The overall Creativity Index, $CI$ is then computed as: $CI = \sum_{i=1}^{6} S_i$.

If the total creativity score, $CI < S_\epsilon$, where $S_\epsilon$ is a predefined threshold, the edited image $I_e$ is considered suboptimal in creativity. The *Creative Director* then reviews the *Critic*'s evaluation and may challenge its assessment if the assigned scores misalign with the creative blueprint, $B$. Once a consensus is reached, $I_e$ is either finalized or progresses to further enhancement based on the final creativity score $CI$.

#### 3.4.1 Self-Enhancement with Optional User-Guidance

While one round of editing produces optimal creative outputs, further refinement can enhance creativity. In the self-enhancement phase, all agents iteratively refine the edited image, $I_e$ with optional human intervention. Given a maximum of $K$ iterations, each edited image $I_k$ is evaluated by the *Critic* to compute its $CI$. When $CI < S_\epsilon$, the *Refinement Strategist*, $A_5$ identifies the low scoring

| Method | CLIP ↑ | LPIPS* ↑ | VENDI ↑ | DINO ↑ | FID ↓ | KID ↓ | User Study-Q1 | User Study-Q2 |
|---|---|---|---|---|---|---|---|---|
| LEDITS++ | $0.396 \pm 0.028$ | $0.252 \pm 0.074$ | $2.88 \pm 1.11$ | $0.678 \pm 0.160$ | 312.50 | 21.45 | $3.21 \pm 1.27$ | $3.50 \pm 1.25$ |
| InstructPix2Pix | $0.379 \pm 0.032$ | $0.289 \pm 0.126$ | $1.94 \pm 0.61$ | $0.704 \pm 0.189$ | 314.42 | 22.51 | $3.14 \pm 1.23$ | $3.59 \pm 1.17$ |
| SDEdit | $0.381 \pm 0.033$ | $0.308 \pm 0.068$ | $3.19 \pm 1.23$ | $0.737 \pm 0.162$ | 304.37 | 18.78 | $3.31 \pm 1.21$ | $3.15 \pm 1.16$ |
| TurboEdit | $0.389 \pm 0.031$ | $0.192 \pm 0.071$ | $2.34 \pm 0.94$ | $0.735 \pm 0.173$ | 320.53 | 24.79 | $3.23 \pm 1.29$ | $2.63 \pm 1.20$ |
| **Ours (Editing)** | $\mathbf{0.417 \pm 0.030}$ | $\mathbf{0.414 \pm 0.157}$ | $\mathbf{3.70 \pm 1.97}$ | $\mathbf{0.744 \pm 0.185}$ | **294.19** | **14.02** | $\mathbf{3.34 \pm 1.34}$ | $\mathbf{3.74 \pm 1.21}$ |
| SDXL | $\mathbf{0.404 \pm 0.033}$ | $0.636 \pm 0.069$ | $6.63 \pm 2.85$ | N/A | 282.22 | 9.67 | $\mathbf{4.37 \pm 0.99}$ | $3.56 \pm 1.10$ |
| Flux | $0.359 \pm 0.048$ | $0.650 \pm 0.088$ | $5.84 \pm 2.69$ | N/A | 270.69 | 9.94 | $4.11 \pm 1.27$ | $3.00 \pm 1.24$ |
| ConceptLab | $0.334 \pm 0.055$ | $0.663 \pm 0.076$ | $10.38 \pm 2.27$ | N/A | 272.97 | 6.49 | $3.40 \pm 1.52$ | $3.18 \pm 1.30$ |
| **Ours (Generation)** | $0.360 \pm 0.052$ | $\mathbf{0.709 \pm 0.057}$ | $\mathbf{10.44 \pm 2.15}$ | N/A | **248.67** | **5.94** | $4.32 \pm 0.99$ | $\mathbf{4.16 \pm 1.01}$ |

Table 1: **Quantitative Comparison of Creative Image Editing and Generation.** Our method surpasses state-of-the-art methods across multiple metrics for both editing and generation tasks. Note that DINO scores cannot be computed for image generation, as they rely on image-image similarity, and there is no reference image available for this task. * indicates that scores are interpreted in opposition to their conventional usage, as creative generation tasks benefit from greater perceptual distance between original and edited images.

creative dimensions and proposes a refinement plan to enhance the corresponding weak dimensions, which the *Prompt Architect* $A_2$ uses to refine the editing prompt, $P_e$ as follows $P_r = P_e + \Delta P$ where $\Delta P$ represents the prompt adjustment that amplify specific creativity dimensions based on their evaluation scores and $P_r$ is the refined creativity prompt. The refined prompt is used by the *Generative Executor* $A_3$ to regenerate an improved image using $I_r = G(P_r, I_k)$.

The process iterates until $CI \geq S_\epsilon$ or $k \geq K$, where $K$ is the maximum allowed iterations and $I_k$ is an intermediate creative image. Users can provide real-time instructions to enhance creativity, which the *Prompt Architect* integrates into the evolving prompt, while the *Refinement Strategist* ensures artistic coherence. The final goal is $\max_{I_c} \mathbb{E}[CI(I_r)]$ where $I_c$ is the final optimized image, achieved through iterative refinement of the refined current image, $I_r$.

# 4 Experiments

**Experimental Setup** In this section, we evaluate our method's ability to generate highly creative edits and images. All experiments use FLUX.1-dev [27]. For editing, we employ ControlNet [59] with a conditioning scale of 0.4 and Canny as the condition. For image generation, we vary the CFG scale from 3.5 to 40 to explore different levels of creativity and control. Additionally, we utilize Autogen [56] for our agentic framework. All experiments are performed on a 48GB NVIDIA L40 GPU. We use GPT-4o as our MLLM for all agents [1]. We set the Creativity Index ($CI$) threshold to 24 for editing, requiring a majority of criteria to score at least 4 ("very good") for an image to be considered creative. For generation, we use a higher threshold of 26 to reflect greater creative freedom unconstrained by a base image. We cap the number of refinement iterations at $K = 3$. A full run of the pipeline takes approximately 3–5 minutes, depending on the number of self-enhancement rounds. Our source code is publicly available at `https://crea-diffusion.github.io`.

## 4.1 Qualitative Results

**Creative Image Editing** First, we qualitatively showcase how CREA transforms input images into various creative modifications. As seen in Fig. 3, CREA's autonomous agents analyze the input image or concept and generate creative prompts: e.g., focusing on style, color, or thematic twists, without requiring extensive user intervention. Since no existing method is explicitly designed for creative image editing, we adapt state-of-the-art editing models for comparison. Specifically, we evaluate the following baselines: LEDITS++ [7], InstructPix2Pix [8], SDEdit [36] and TurboEdit [12]. For a fair comparison, we apply a "creative <object>" prompt (e.g., "a creative couch") to the baseline methods, mirroring the objective of CREA. As shown in Fig. 4, standard editing methods often fail to generate distinctly creative concepts: InstructPix2Pix typically adds vibrant colors or alters the background extensively without fundamentally reimagining the object. LEDITS++, SDEdit, and TurboEdit struggle to introduce creative features beyond superficial stylistic changes. In contrast, CREA successfully performs creative edits in a disentangled manner. For more experiments such as more fine-grained editing examples, or other image generators, please refer to Appendix D.4.

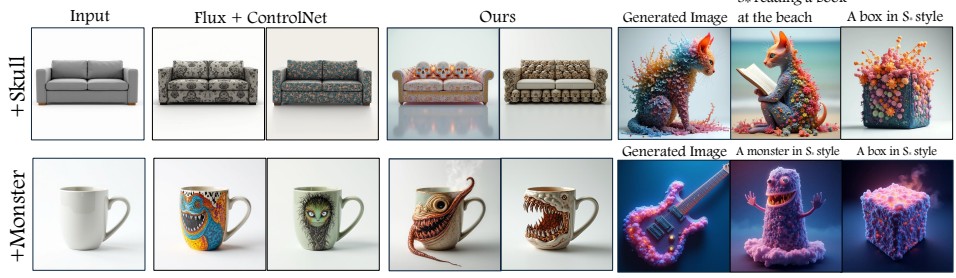

a) User-in-the-loop generation        b) Personalization

Figure 6: **Creative applications of our method beyond image generation and editing.** (a) Users can steer the creative process with additional conditions such as 'Monster'. (b) CREA-generated images can be leveraged for personalization in creative domains.

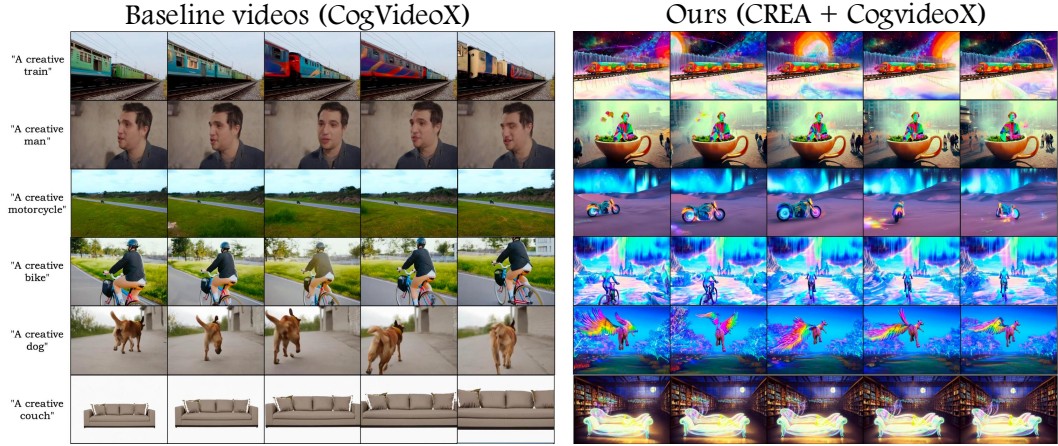

Figure 7: **Video Generation**. Comparison between baseline generations from CogVideoX and CREA. Our method enables the creation of visually diverse and creative video scenes.

**Creative Image Generation** For creative image generation, we compare CREA against ConceptLab [41], which is the closest related approach, as well as two generative baselines: Flux [27] and SDXL [39]. To ensure a fair comparison, we use the same random seed for each method in every evaluated prompt, allowing for direct visual comparisons. As illustrated in Fig. 5, CREA consistently produces diverse and creative generations across multiple domains. In contrast, ConceptLab struggles to maintain the intended concept, particularly for highly abstract or unconventional categories. For example, when generating "a monster," ConceptLab often fails to produce a meaningful interpretation, as it relies on extracting subcategories from the BLIP model. If a given concept lacks well-defined subcategories or consists of highly specific attributes, ConceptLab's fails to perform well. CREA generalizes across a broad range of creative categories without relying on predefined subcategories.

### 4.2 Quantitative Results

For both editing and generation, we utilized 24 different objects with 25 prompts (either for editing or generation), resulting in an evaluation set of 600 images per task.

**Creative Image Editing.** For image editing, we compare our method against state-of-the-art techniques, including LEDITS++, InstructPix2Pix, SDEdit, and TurboEdit, across a range of evaluation metrics. To assess how well the edited objects align with the text descriptions, we compute the CLIP score by measuring the similarity between the generated image and its corresponding text prompt. For evaluating diversity, we use the LPIPS [60] score, which quantifies perceptual distance between images. Unlike conventional usage where lower LPIPS indicates better reconstruction, a higher LPIPS in our case signifies stronger, more transformative edits, a desirable trait in creative tasks. Additionally, we compute the DINO [58] score which measures how well the edited image retains key structural and semantic characteristics of the original image, and Vendi [17] score, which quantifies diversity by calculating the Shannon entropy of the eigenvalues of the similarity matrix among the

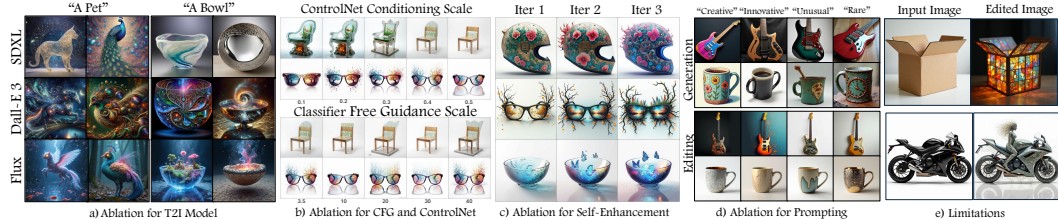

Figure 8: **Ablation Study.** We perform comprehensive ablation studies to analyze the design choices of CREA: (a) Model Generalization: Our method extends effectively to different generative models, such as SDXL and DALL-E. (b) Parameter Sensitivity: We ablate CFG values for Flux and the conditioning scale for ControlNet to evaluate their impact. (c) Iterative Refinement: We demonstrate the benefits of our method's refinement process over multiple iterations. (d) Prompt Variations: We explore alternative prompts beyond 'a creative <obj>'.

generated images. As shown in Table 1, our method consistently outperforms all baselines across all evaluation metrics, demonstrating its superior ability to generate creative and diverse image edits.

**Creative Image Generation.** For creative image generation, we compare CREA against ConceptLab, Flux, and SDXL. As shown in Table 1, our method achieves superior results in LPIPS and VENDI scores, while in CLIP scores, it was outperformed by SDXL. We observe that SDXL tends to generate highly colorful images, which aligns with CLIP's depiction of creativity. However, it produces the lowest LPIPS scores, indicating that its generated images are highly similar to each other, and it also achieves significantly lower VENDI scores compared to CREA and ConceptLab. While ConceptLab and our method achieve comparable results overall, ConceptLab has significantly lower performance CLIP scores, suggesting that its generated images do not align well with the text prompt. This limitation stems from ConceptLab's reliance on well-defined subcategories—a core design principle of their method. As shown in Fig. 5, ConceptLab fails when a concept lacks sufficient subcategories, such as its inability to accurately generate a couch object. While ConceptLab attains a high VENDI score, this failure highlights its trade-off between diversity and semantic alignment.

For additional comparisons with recent methods such as GenArtist [52] and RF-Inversion [43], see Appendix D.3.

**User Study.** To assess the creativity of the generated images from a human perception standpoint, we conducted a user study on Prolific.com with 50 participants (see Table 1. Participants rated each criterion using a 5-point Likert scale, ranging from 1 (Not at all) to 5 (Very well). For the image generation task, User Study-Q1 evaluates Usability, measuring how accurately the generated image represents the specified object. User Study-Q2 evaluates Creativity, assessing the uniqueness and originality of the image. Among all competitors, SDXL performed the best on Q1 since it is generating generic objects (e.g., a simple cup) which is perceived as more useful by participants, whereas our method significantly outperformed others on Q2 due to its ability to produce more creative visuals. For edited images, we evaluate editing consistency—how well the original image characteristics are preserved while performing edits (User Study-Q1)—and the creativity of the edits (User Study-Q2). Our method achieves comparable results to state-of-the-art editing techniques on Q1, demonstrating strong disentangled editing capabilities. Moreover, it surpasses all competitors on Q2, highlighting its superior ability to generate creative edits.

### 4.3 Additional Experiments

**User-Guided Editing and Generation** Our method enables users to guide the creative process according to their preferences (see Fig. 6 (a)). For example, users can specify not only that a creative edit should be applied to a cup but also include additional conditions, such as 'monster', to influence the generation. Our approach incorporates these preferences, steering the creative process accordingly. Compared to the baseline Flux + ControlNet approach, where we provided the prompt 'A creative <condition> <object>', our results demonstrate greater creativity.

**Personalization** We also present personalization results using CREA-generated creative images using an off-the-shelf personalization adapter [51]. Fig. 6 (b) demonstrates how generated subjects can be adapted to various contexts or how their styles can be transferred to create new images. Note that due

| Method | LPIPS-Diversity ↑ | Vendi ↑ |
|---|---|---|
| Base | $0.302 \pm 0.129$ | $3.19 \pm 1.49$ |
| +Principles | $0.312 \pm 0.142$ | $3.59 \pm 2.46$ |
| +Contrastive | $0.391 \pm 0.150$ | $3.50 \pm 1.64$ |
| +Self-Enhancement | $\mathbf{0.414 \pm 0.157}$ | $\mathbf{3.70 \pm 1.97}$ |

Table 2: **Ablation Study for CREA.** CREA achieves the highest performance across all metrics when all components are utilized, demonstrating the effectiveness of our full framework.

to the limitations in personalization models, some fine-grained details were not able to capture by [51], however the main characteristics are still preserved.

**Video Generation** To demonstrate the versatility and extensibility of our agentic framework, we extend CREA to creative video generation, where an initial prompt (e.g., 'a train') is automatically transformed into creative scenes. We conduct experiments using the CogVideoX [57] model as the generative backbone. Instead of initializing the process with a static creative blueprint, our Creative Director agent generates a structured creative *video plan* containing key fields: *Subject*, *Action*, *Setting*, *Style*, and optional *Additional Details*. This plan is passed to the Prompt Architect, who composes a coherent and high-creativity video prompt via contrastive prompt fusion, similar to our image generation pipeline. The Generative Executor then synthesizes the video using CogVideoX. This evaluation serves as a proof of concept for applying our multi-agent creativity principles beyond static imagery. As shown in Figure 7, our method produces significantly more imaginative and visually engaging results compared to baseline model.

**Ablation Studies** We perform comprehensive ablation studies to analyze the design choices of CREA (see Fig. 8). (a) **Ablation on Generalization to other T2I models**: Our method extends effectively to different generative models, such as SDXL and DALL-E. (b) **Ablation on Parameter Sensitivity**: We ablate CFG values (3.5-40) for Flux and the conditioning scale for ControlNet (0.1-0.5) to evaluate their impact. (c) **Ablation on Iterative Refinement**: We demonstrate the benefits of our method's refinement process over multiple iterations. (d) **Ablation on Prompt Variations**: We explore alternative prompts beyond 'a creative <obj>' to steer the model away from conventional generations. (e) **Ablation on Model Components** We also conduct an ablation study on our key components (see Table 2). The base version represents a baseline where the multi-modal LLM model is simply prompted to generate a creative description for a given object. Please see Appendix for more ablations, such as an ablation with negative prompting.

## 5 Discussion

**Limitations and Broader Impact** Since our framework utilizes a T2I model such as Flux, certain biases inherent to the generative backbone can affect the outputs. For instance, we observe cases where the background is unintentionally darkened based on the object's semantics (e.g., editing a "lighted box" often results in darker surroundings to emphasize contrast), or where the model hallucinates human figures even when not prompted (see Fig. 8 (e)). Moreover, we recognize that defining and quantifying visual creativity is itself an open research problem. Instead of introducing a new assessment protocol, we tackle creative image editing and generation through established principles in creativity. Assessing what is truly a creative output, however, remains beyond the scope of this paper. For broader impacts, on the positive side, our agentic framework can democratize high-level creative tools, lowering barriers for artists, educators, and hobbyists to experiment with sophisticated visual storytelling. Potential downsides include possible displacement of human labor in creative industries. We encourage responsible deployment by fostering human-in-the-loop usage that positions the system as a co-creative assistant rather than a replacement for human creators.

**Conclusion** In this work, we introduce a novel agentic framework for creative image editing and generation, pioneering a disentangled approach that enables greater flexibility and artistic control. By leveraging specialized agents that collaborate to refine and enhance outputs, our method overcomes the limitations of traditional prompt-to-image models, reducing the burden on users while fostering creativity. We demonstrate the versatility of our approach across editing and generation, and highlight its potential for creative video generation. Our findings suggest that agentic frameworks can serve as a powerful foundation for more autonomous and creative AI systems, opening new directions for creative collaboration between humans and AI.

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

# Table of Contents

## A  Investigating Creativity Types

We investigate emerging patterns in generated images to identify key attributes—shape, color, and texture that influence creativity. Our analysis reveals that our method achieves significantly higher scores in these aspects, leading to the generation of more visually diverse and imaginative objects (see Fig. 9). Compared to baseline models, our approach demonstrates a stronger ability to manipulate these factors, reinforcing its effectiveness in producing unique and aesthetically rich generations.

We also investigate emerging factors for different objects within our method (see Fig. 10).

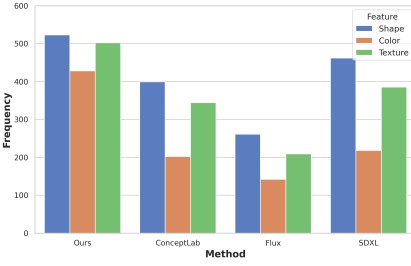

Figure 9: Emerging patterns in the generated images across CREA, ConceptLab, Flux, and SDXL reveal that our method achieves significantly higher scores in shape, color, and texture, enabling the generation of more creative objects.

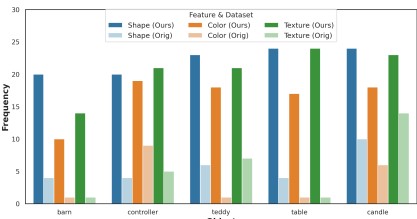

Figure 10: Emerging patterns in the generated images across various objects reveal that our method dynamically emphasizes different factors - such as shape, color, or texture depending on the object, effectively enhancing creativity in a context-aware manner.

| Method | Originality ↑ | Expressiveness ↑ | Aesthetic ↑ | Technical ↑ | Unexpected ↑ | Interpretability ↑ | Total ↑ | Creativity ↑ |
|---|---|---|---|---|---|---|---|---|
| LEDITS++ | 3.04 ± 0.74 | 2.52 ± 0.66 | 3.56 ± 0.55 | 3.80 ± 0.47 | 2.86 ± 0.84 | 2.60 ± 0.71 | 18.38 ± 3.37 | 77.80 ± 8.83 |
| InstructPix2Pix | 2.43 ± 0.91 | 2.28 ± 0.80 | 3.59 ± 0.54 | 3.76 ± 0.51 | 2.12 ± 0.82 | 2.21 ± 0.78 | 16.39 ± 3.70 | 69.78 ± 15.94 |
| SDEdit | 2.50 ± 0.87 | 2.12 ± 0.72 | 3.35 ± 0.50 | 3.59 ± 0.51 | 2.12 ± 0.78 | 2.07 ± 0.72 | 15.74 ± 3.42 | 69.39 ± 13.54 |
| TurboEdit | 2.68 ± 0.76 | 2.12 ± 0.62 | 3.28 ± 0.50 | 3.66 ± 0.47 | 2.44 ± 0.78 | 2.12 ± 0.62 | 16.30 ± 3.11 | 71.34 ± 13.87 |
| Ours (Editing) | **3.77 ± 1.00** | **3.49 ± 1.03** | **4.49 ± 0.62** | **4.61 ± 0.55** | **3.66 ± 1.12** | **3.46 ± 1.03** | **23.48 ± 4.95** | **83.47 ± 6.68** |
| SDXL | 3.45 ± 0.89 | 3.30 ± 0.77 | 4.15 ± 0.52 | 4.31 ± 0.49 | 3.33 ± 1.00 | 3.32 ± 0.79 | 21.85 ± 3.91 | 82.95 ± 6.56 |
| Flux | 2.87 ± 0.69 | 3.05 ± 0.74 | 4.12 ± 0.54 | 4.17 ± 0.47 | 2.59 ± 0.78 | 2.86 ± 0.64 | 19.66 ± 3.14 | 79.46 ± 7.82 |
| ConceptLab | 3.49 ± 0.75 | 3.32 ± 0.84 | 3.75 ± 0.77 | 3.93 ± 0.71 | 3.42 ± 0.83 | 3.34 ± 0.81 | 21.25 ± 4.13 | 80.99 ± 6.84 |
| Ours (Generation) | **4.39 ± 0.69** | **4.55 ± 0.54** | **4.98 ± 0.16** | **4.96 ± 0.20** | **4.39 ± 0.70** | **4.42 ± 0.64** | **27.68 ± 2.55** | **89.87 ± 4.04** |

Table 3: **Quantitative Comparison of Creative Image Editing and Generation using LLM-as-a-judge.** We used a multi-modal LLM as a judge for simulating human-like subjective assessments across several key aspects of creativity. Our method surpasses state-of-the-art methods across all aspects for both editing and generation tasks.

# B  CREA's Multi-Agentic Architecture

## B.1  CREA's Multi-Agent Framework Design

In the CREA framework, prompts serve as the fundamental coordination interface between specialized agents, guiding the collaborative creative process. Unlike traditional systems that rely on static user prompts, CREA uses modular, role-specific prompting strategies to simulate a human-like creative workflow. These prompts enable agents to reason contextually, communicate asynchronously, and iteratively improve upon the image through multiple refinement cycles.

Our multi-agent framework is composed of five roles: *Creative Director, Prompt Architect, Generative Executor, Art Critic, and Refinement Strategist*. Each agent is instantiated with a carefully designed system prompt tailored to its unique responsibilities. These prompts are not only essential for aligning agent behavior with creativity principles, but also for facilitating coherent multi-turn conversations between agents. The following subsections describe the prompt structure for each agent, explain its purpose, and highlight how it contributes to the overall creativity loop. For each, we refer to specific examples and prompt templates shown in the supplementary material.

## B.2  Creative Director ($A_1$) Prompt

The Creative Director serves as the strategic leader of the agent team. Its prompt equips it not only to interpret the user-provided input (e.g., 'a couch') but also to conceptualize a feasible, imaginative, and well-grounded plan for the entire task. More than just an interpreter, the Creative Director performs high-level planning, coordination, and decision-making. It establishes the overall direction of the creative task, specifies whether the objective is image generation or editing, and formulates a comprehensive creativity blueprint. This blueprint includes the visual theme, stylistic and structural constraints, novelty objectives, semantic anchors, and any potential unexpected associations that could add conceptual depth. Once defined, this blueprint is shared with all other agents and acts as the central reference point throughout the iterative process. The Creative Director also participates in post-generation evaluation, arbitrates conflicts in Critic feedback, and decides whether an output is ready for finalization or requires refinement. A high-level system prompt used for the Director, $A_1$ is shown in Template 13.

**B.3  Prompt Architect ($A_2$) Prompt**

The Prompt Architect operates as the creative translator and synthesis engine of the system. Based on the Creative Director's blueprint and the creativity template, this agent generates a structured set of prompts to guide the generative process. It first decomposes the blueprint into six contrastive prompts, each aligned with a specific creativity principle—originality, expressiveness, aesthetic appeal, technical execution, unexpected associations, and interpretability. These prompts isolate individual creative dimensions, allowing for focused exploration of each aspect. The template used by $A_2$ to generate Contrastive Prompts is shown in Template 9. Once the contrastive prompts are created, the Prompt Architect uses a Chain-of-Thought fusion process to synthesize them into a single fused high-creativity prompt as shown in Template 10. This fused prompt balances all six principles while preserving semantic clarity and stylistic coherence. Throughout the generation loop, the Prompt Architect also integrates feedback from the Refinement Strategist, modifying or rebalancing prompts as needed to improve underperforming dimensions. This iterative prompting system enables nuanced control over the generation process while ensuring alignment with the blueprint. A high-level system prompt used for the Prompt Architect, $A_2$ is shown in Template 8.

**B.4  Generative Executor ($A_3$) Prompt**

The Generative Executor is responsible for realizing the creative vision by generating or editing images based on the prompts received from the Prompt Architect. Guided by its system prompt, this agent dynamically interprets the creative instruction and configures the appropriate image generation strategy using T2I models such as Flux or ControlNet. The Executor considers the task type (generation or editing), model-specific parameters like classifier-free guidance or conditioning scale, and any constraints defined by the blueprint. In editing mode, it ensures disentangled transformation by preserving structural elements from the input image while applying stylistic or conceptual changes aligned with the fused prompt. A high-level system prompt used for the Executor, $A_3$ is shown in Template 11.

**B.5  Art Critic ($A_4$) Prompt**

The Art Critic acts as an autonomous evaluator, responsible for analyzing the generated or edited image and providing structured, multi-dimensional feedback. Its system prompt enables it to assess the output based on the six creativity principles and assign numerical scores from 1 to 5 for each dimension. The Critic also produces rich textual justifications for its ratings and provides an overall creativity score. Beyond evaluation, the Critic serves a pivotal role in steering the refinement process by identifying which dimensions underperform and articulating why. This feedback is critical to both the Creative Director, who decides whether the image meets the creativity threshold, and to the Refinement Strategist, who uses it to guide further improvements. The Critic's evaluations are grounded in a multi-modal LLM, ensuring assessments are both visually and semantically informed. This agent's consistent, transparent scoring facilitates traceable progress across iterations. high-level system prompt used for the Critic, $A_4$ is shown in Template 12.

**B.6  Refinement Strategist ($A_5$) Prompt**

The Refinement Strategist is the system's adaptive problem solver. Upon receiving the Critic's evaluation and the current image, this agent analyzes weak areas in the creativity spectrum and formulates a targeted refinement strategy. Its system prompt allows it to propose actionable changes using a formulation $\Delta P$ that identifies how the existing fused prompt should be adjusted to improve specific dimensions, such as expressiveness or technical execution. The strategist is also tasked with preserving creative coherence; it aligns each suggested change with the original blueprint to avoid drift or over-editing. Its feedback is passed to the Prompt Architect, who updates the prompts accordingly for the next iteration. The Strategist plays a critical role in achieving optimal creativity without compromising conceptual consistency, functioning as the bridge between critique and constructive improvement. A representative example of its prompt behavior and refinement plan is available in Template 14.

---

**Algorithm 1** CREA Method Overview

---

**Input:** User concept $c$ or Initial image $I_0$, Max iterations $K$, Creativity threshold $S_\epsilon$
**Given:** Creativity principles template, $T$
**Init:** Agents $\{\mathcal{A}_1 : \mathcal{A}_5\}$

**Pre-Generation Planning**
 1: **if** generation task **then**
 2:     $\mathcal{A}_1 \to B$ from $c$
 3: **else if** editing task **then**
 4:     $\mathcal{A}_1 \to B$ from $I_0$ (initial generation/user-provided)
 5: **end if**
 6: $\mathcal{A}_2 \to P = \{p_1, p_2, ..., p_6\}$
 7: $\mathcal{A}_2 \to P_c = \text{CoT-Fusion}(P)$
 8: $\mathcal{A}_3 \to$ Validate $P_c$, adjust model parameters

**Image Synthesis and Editing**
 9: **if** Generating a new image **then**
10:     $I_0 = G(P_c, \theta)$
11: **else if** Editing an existing image **then**
12:     $I_e = G(P_c, I_0, \theta)$
13: **end if**

**Post-Generation Evaluation**
14: $\mathcal{A}_4 \to S = \{S_1, ..., S_6\}$ for $I_e$
15: Compute $CI = \sum_{i=1}^{6} S_i, \quad S_i \in [1, 5]$
16: **if** $CI \geq S_\epsilon$ **then**
17:     **return** $I_e$ as $I_c$
18: **else**
19:     Proceed to refinement phase
20: **end if**

**Self-Enhancement with Optional User-Guidance**
21: **for** $k = 1$ to $K$ **do**
22:     $\mathcal{A}_5 \to$ Identify weak $S_i$, suggest refinements
23:     $\mathcal{A}_2 \to P_r = P_e + \Delta P$            ▷ refined prompt, $P_r$
24:     $\mathcal{A}_3 \to I_r = G(P_r, I_k, \theta)$           ▷ intermediate image, $I_k$
25:     $\mathcal{A}_4 \to S = \{S_1, ..., S_6\}$ for $I_r$         ▷ refined image, $I_r$
26:     **if** $CI \geq S_\epsilon$ **then**
27:         **return** $I_r$ as $I_c$
28:     **end if**
29: **end for**

**Output:** Final high-creativity image $I_c$

---

## B.7  Algorithm

An algorithm summarizing the agentic framework is given in Algorithm 1.

CREA is built on a dynamic multi-agent architecture that emulates the collaborative human creative process by distributing cognitive and generative responsibilities across specialized agents. Each agent in the system plays a distinct role and is instantiated using the AutoGen framework's ConversableAgent class, enabling structured communication, tool access, and memory management.

**Creative Director** ($A_1$)**:** Acts as the master orchestrator. Powered by GPT-4o, this agent defines the overall creative blueprint, interprets user goals or concepts, and decides whether a generated image meets the creativity criteria. It has functional tools to collaborative with all other agents and a tool to determine if the creativity index, $CI$ is greater than the defined threshold. It does not execute code but uses a shared memory to track blueprint adherence across rounds.

**Prompt Architect** ($A_2$)**:** Converts the blueprint into six contrastive prompts based on creativity principles (e.g., Originality, Aesthetic Appeal), which are fused into a high-creativity composite prompt using Chain-of-Thought reasoning. This agent uses GPT-4o and has access to prompt-fusion tools and tools to interact with other agents, enabling it to translate abstract directives into actionable instructions. It uses the shared memory to track progress, store base templates, and exchange updates.

**Generative Executor ($A_3$):** Interfaces with the image generation or editing backend (e.g., FLUX, ControlNet). It has code execution capabilities and is responsible for producing images using prompt, $P_c$ and associated parameters like classifier-free guidance (CFG) or conditioning scales. It can perform both text-to-image generation and disentangled image editing and has access to predefined tools to adjust the hyperparameters of models. In addition, it has tools to manipulate images based on instructions from the CreativeDirector or the User-such as personalization and creative video editing. Its toolbox is highly customizable to incorporate training-free image editing and personalization tools.

**Art Critic ($A_4$):** Uses a multi-modal LLM judge (GPT-4o with vision capabilities) to evaluate the image against six creativity criteria. It returns per-dimension scores (1–5 scale) and an aggregate Creativity Index ($CI$). This agent is crucial for quality control and can challenge previous creative decisions. Art Critic has access to tools to interact with other agents. However, its shared memory has a $window\_size = 1$. This is intentional and ensures independent evaluation of each generated creative image, without being influences by previous evaluations.

**Refinement Strategist ($A_5$):** Translates evaluation feedback into actionable prompt refinements. It identifies weak creative dimensions (e.g., low expressiveness) and proposes targeted edits. It works closely with $A_2$ to iteratively improve the prompt and coordinates with $A_3$ to re-render improved results. It uses GPT-4o and has access to interaction tools and the shared memory to track information and updates.

A UserProxyAgent models Human-in-the-loop and is turned on for optional user-guidance.

## B.8    Runtime and Practical Considerations

A full round of creative generation or editing with three iterations of self-enhancement typically takes 3–5 minutes on an NVIDIA L40 GPU using FLUX.1-dev1 and ControlNet for editing. The runtime is dependent on several factors-such as prompt complexity, tool execution overhead and LLM inference time. Each agent uses GPT-4o (via API), which contributes significantly to latency, especially during multi-agent coordination and critique. In practice, the system is designed for rapid prototyping while maintaining high creativity scores. Early stopping is applied when the Creativity Index, $CI \geq 24$ for editing and $CI \geq 26$ for generation, minimizing unnecessary iterations. Optional user feedback can intervene between rounds for guided refinements. $CI$ for editing and generation are chosen based on experimentation and can be adjusted according to user-preferences.

# C    CREA Editing and Generation Prompts

We provide the full natural language prompts used to produce the visual results shown in the teaser figure 1 of the main paper.

Examples include both creative image editing (transformations of real-world or AI generated images of objects such as dresses and teapots) and creative image generation (novel interpretations of minimal text inputs such "as "a couch" or "a car"). Each prompt was written to embody multiple creativity principles - e.g., combining unexpected materials, evoking emotional tone, or supporting layered interpretation, resulting in conceptually rich and visually expressive outputs. Figure 11 shows these prompts.

## C.1    Creative Image Generation

Creative image generation in CREA unfolds through a structured, collaborative multi-agent workflow composed of three key phases: Pre-Generation Planning, Image Generation, Self-Enhancement and Post-Generation Evaluation. Each phase involves distinct agent roles engaging in goal-driven dialogue to ideate, generate, and refine creative imagery. We provide an illustrative example of the collaborative multi-agent debate for image generation using our proposed three-phase approach in the next sections. Note that the conversations are color-coded as follows: {''REASON'': ''red'', ''THOUGHT'': ''pink'', ''ACTION'': ''green'', ''PROMPT'': ...}.

| Original Image | Creative Edit | CREA's Editing Prompt |
|---|---|---|

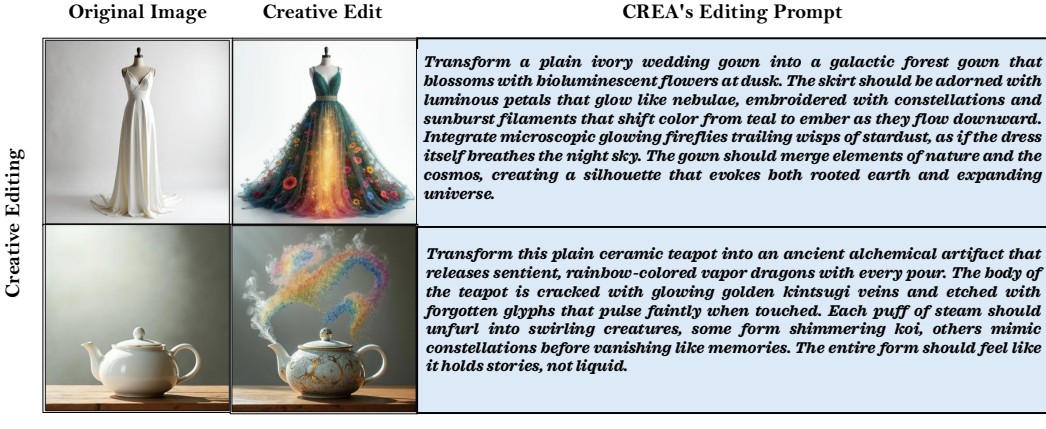

**Creative Editing**

*Transform a plain ivory wedding gown into a galactic forest gown that blossoms with bioluminescent flowers at dusk. The skirt should be adorned with luminous petals that glow like nebulae, embroidered with constellations and sunburst filaments that shift color from teal to ember as they flow downward. Integrate microscopic glowing fireflies trailing wisps of stardust, as if the dress itself breathes the night sky. The gown should merge elements of nature and the cosmos, creating a silhouette that evokes both rooted earth and expanding universe.*

*Transform this plain ceramic teapot into an ancient alchemical artifact that releases sentient, rainbow-colored vapor dragons with every pour. The body of the teapot is cracked with glowing golden kintsugi veins and etched with forgotten glyphs that pulse faintly when touched. Each puff of steam should unfurl into swirling creatures, some form shimmering koi, others mimic constellations before vanishing like memories. The entire form should feel like it holds stories, not liquid.*

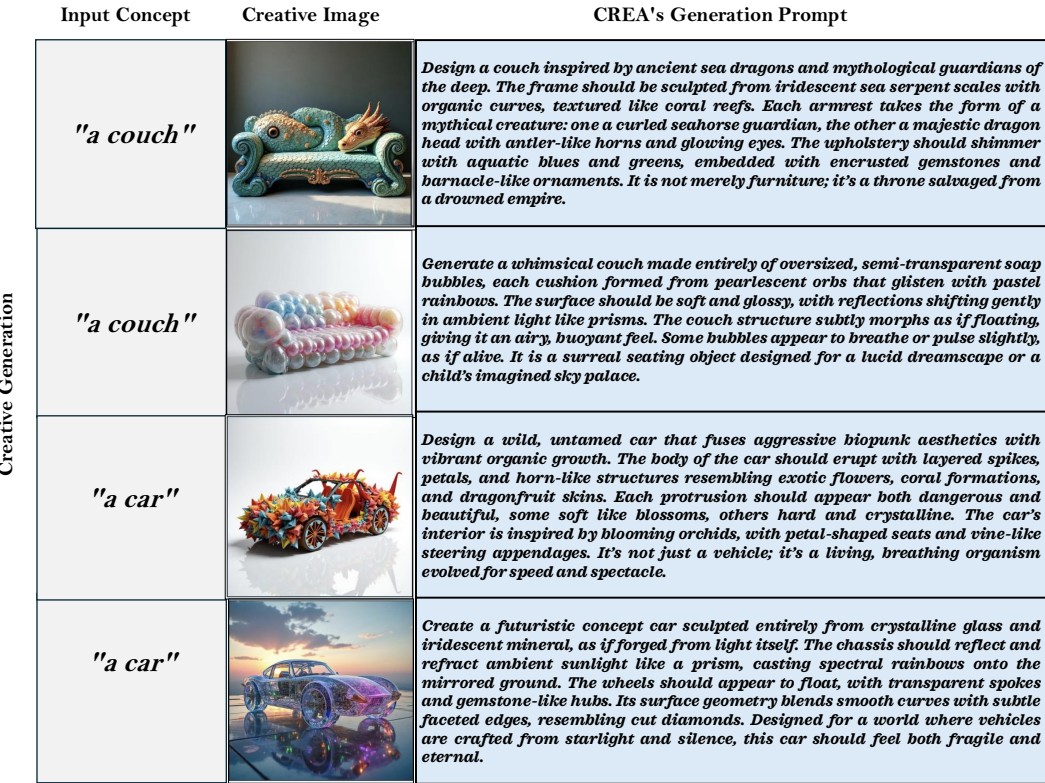

| Input Concept | Creative Image | CREA's Generation Prompt |
|---|---|---|

**Creative Generation**

*"a couch"* — *Design a couch inspired by ancient sea dragons and mythological guardians of the deep. The frame should be sculpted from iridescent sea serpent scales with organic curves, textured like coral reefs. Each armrest takes the form of a mythical creature: one a curled seahorse guardian, the other a majestic dragon head with antler-like horns and glowing eyes. The upholstery should shimmer with aquatic blues and greens, embedded with encrusted gemstones and barnacle-like ornaments. It is not merely furniture; it's a throne salvaged from a drowned empire.*

*"a couch"* — *Generate a whimsical couch made entirely of oversized, semi-transparent soap bubbles, each cushion formed from pearlescent orbs that glisten with pastel rainbows. The surface should be soft and glossy, with reflections shifting gently in ambient light like prisms. The couch structure subtly morphs as if floating, giving it an airy, buoyant feel. Some bubbles appear to breathe or pulse slightly, as if alive. It is a surreal seating object designed for a lucid dreamscape or a child's imagined sky palace.*

*"a car"* — *Design a wild, untamed car that fuses aggressive biopunk aesthetics with vibrant organic growth. The body of the car should erupt with layered spikes, petals, and horn-like structures resembling exotic flowers, coral formations, and dragonfruit skins. Each protrusion should appear both dangerous and beautiful, some soft like blossoms, others hard and crystalline. The car's interior is inspired by blooming orchids, with petal-shaped seats and vine-like steering appendages. It's not just a vehicle; it's a living, breathing organism evolved for speed and spectacle.*

*"a car"* — *Create a futuristic concept car sculpted entirely from crystalline glass and iridescent mineral, as if forged from light itself. The chassis should reflect and refract ambient sunlight like a prism, casting spectral rainbows onto the mirrored ground. The wheels should appear to float, with transparent spokes and gemstone-like hubs. Its surface geometry blends smooth curves with subtle faceted edges, resembling cut diamonds. Designed for a world where vehicles are crafted from starlight and silence, this car should feel both fragile and eternal.*

Figure 11: **Creative Image Editing and Generation Prompts.** Full prompts corresponding to the visual examples shown in the teaser figure of the main paper. Top: real-world images transformed via richly imaginative editing prompts. Bottom: novel objects generated from minimal textual prompts.

### C.1.1 Pre-Generation Planning

In this phase, the agents collectively interpret the user-provided concept (e.g., "Couch") and co-develop a high-creativity prompt. As shown in Figure 12, the Creative Director (A1) begins by synthesizing a creativity blueprint capturing the visual theme, style, constraints, and suggested associations-such asblending fantasy elements with recognizability. This blueprint is handed to the Prompt Architect (A2), who then generates six contrastive prompts, each aligned with a specific creativity principle (e.g., originality, aesthetic appeal, technical execution). These are merged via Chain-of-Thought reasoning into a single, richly structured prompt. The Generative Executor (A3) then verifies the technical feasibility of the prompt. This multi-agent back-and-forth ensures the final creative prompt is both imaginative and model-compatible before proceeding to image synthesis.

**Image Generation**

**TASK: Given a user input concept, compose a high-creativity for creative image generation**

**Input Concept: Couch**

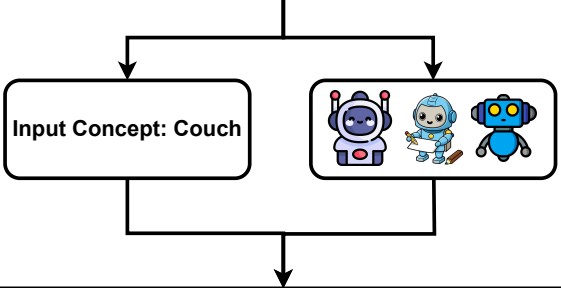

{**"task":** "Surreal Couch Design",
**"master_plan":** "Blend reality and fantasy with levitating, shape-shifting, or organic elements.",
**"feasibility_constraints":** "Must be recognizable as a couch; avoid overly complex forms.",
**"style_aesthetic_direction":** "Daring and experimental, with a vibrant yet harmonious color palette.",
**"elements_focal_points":** "Reflective or living surfaces, integration of water/plants, couch in unconventional settings.",
**"unexpected_associations":** "Juxtapose soft/hard textures, traditional/avant-garde designs, creative lighting for depth.",
**"output":** "High-res concept image or 3D render with brief design rationale."}

**Creative Director**: THOUGHT [I will instruct the Prompt Architect (A2) to craft six contrastive prompts based on the blueprint and then merge them into one comprehensive, high-creativity prompt. Once the Generative Executor (A3) verifies technical feasibility, we will move forward with the image creation.]

**Creative Director (to Prompt Architect):** ACTION [Generate the set of Contrastive Prompts based on the Blueprint for each of the six creativity principles in the template]

**Prompt Architect:** THOUGHT [I will access the generated blueprint and the creativity template.]
**Prompt Architect:** REASON [Analyze the blueprint and the required creativity principles]
**Prompt Architect:** ACTION [Generate six contrastive prompts] -->

{"originality":"A couch floating effortlessly in mid-air, its cushions transforming into soft clouds under an azure sky.","expressiveness":"A vintage leather couch on an empty stage, bathed in warm golden light as it conjures nostalgia and serene solitude.","aesthetic":"An elegantly curved velvet couch surrounded by cascading emerald vines and delicate cherry blossom petals against a sapphire twilight backdrop.","technical":"A couch exquisitely crafted from intricate marble carvings, showcasing rich textures and elaborate patterns in a dimly lit gallery.","unexpected":"A couch with cushions made of tessellating geometric kaleidoscopes, seamlessly blending into an abstract art gallery wall.","interpretability":"A couch in a deserted attic where its worn fabric narrates silent tales of past occupants, alongside the scattered memories of old suitcases and dusty books."}

**Prompt Architect:** ACTION [Fuses contrastive prompts via Chain-of-Thought to get high-creativity prompt]

{"fused_prompt":"A couch that defies expectations and invites the viewer into a world of wonder. Picture a luxurious couch crafted from an intricate blend of materials—velvet cushions that morph into soft clouds gently floating above, juxtaposed with richly textured marble armrests. This central piece, hovering against a pure white background, holds the eye with its elegant curves and vibrant emerald vines that cascade over its form, ending in delicate cherry blossom petals. Unexpectedly, geometric kaleidoscopes can be glimpsed within the cushions, offering a changeable pattern that weaves in and out of perception. Bathed in a subtly shifting light, this couch evokes feelings of nostalgia and curiosity, each glance revealing new stories written in its worn but elegant fabric, reminiscent of an animated stage or cinematic scene capturing fleeting past moments and emotions."}

**Creative Director:** THOUGHT [We need to ensure we all approve of the prompt and the theme]

**Creative Director (to the two agents):** Do you approve the prompt?

**Generative Executor:** REASON [Is it feasible to generate an image with this prompt?]
ACTION [I approve]

**Prompt Architect:** REASON [I will verify the prompt again]
ACTION [I approve]

**Creative Director:** THOUGHT [Process Finished.]

**Creative Director:** Action [Return the high-creativity prompt. Move to the Next Phase]

7

Figure 12: Pre-Generation Planning Phase

Once the prompt is finalized, the system transitions to image generation. As illustrated in the Figure 13, the Creative Director instructs the Generative Executor to generate an image based on the previously approved prompt. The Executor uses a text-to-image model (e.g., Flux) and executes a controlled synthesis by setting guidance parameters (such as CFG scale). The resulting image is then routed back to the Creative Director to confirm process completion. This phase emphasizes precise execution while preserving the conceptual richness embedded in the prompt, ensuring alignment between intent and output.

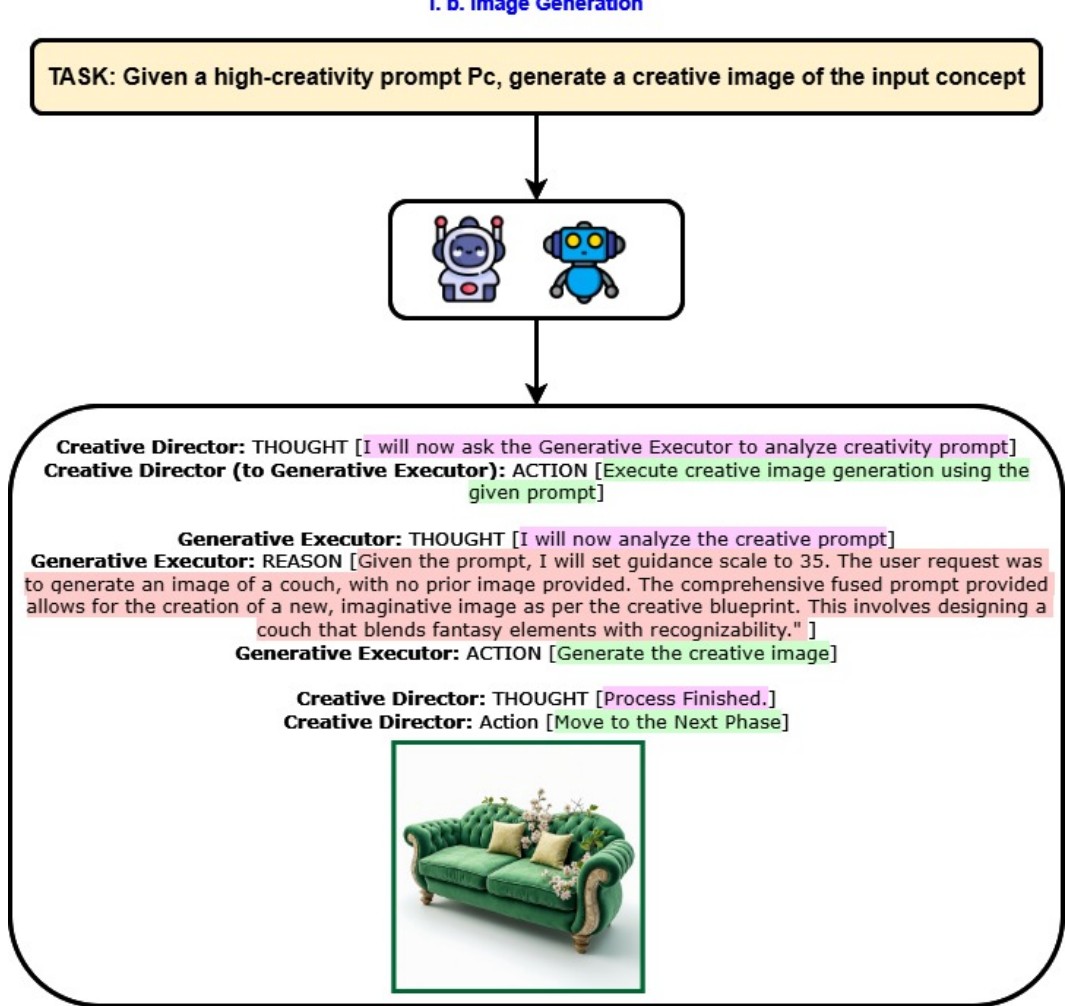

Figure 13: Image Generation by Generative Executor

### C.1.2 Post-Generation Evaluation

After the initial image is generated, it is evaluated in detail by the Art Critic (A4) and the Creative Director (A1) to determine its alignment with the intended creativity blueprint. As shown in Figure 14, the Art Critic applies a multimodal LLM-as-a-Judge framework to assign scores across six creativity principles: Originality, Expressiveness, Aesthetic Appeal, Technical Execution, Unexpected Associations, and Interpretability. Each criterion is scored on a 1–5 scale, and the total is aggregated into a Creativity Index (CI). The Creative Director then reviews these scores in light of the original intent and either approves the image or requests revisions if the CI falls below the creativity threshold. This phase ensures that the image undergoes a rigorous, objective-aligned critique before being finalized.

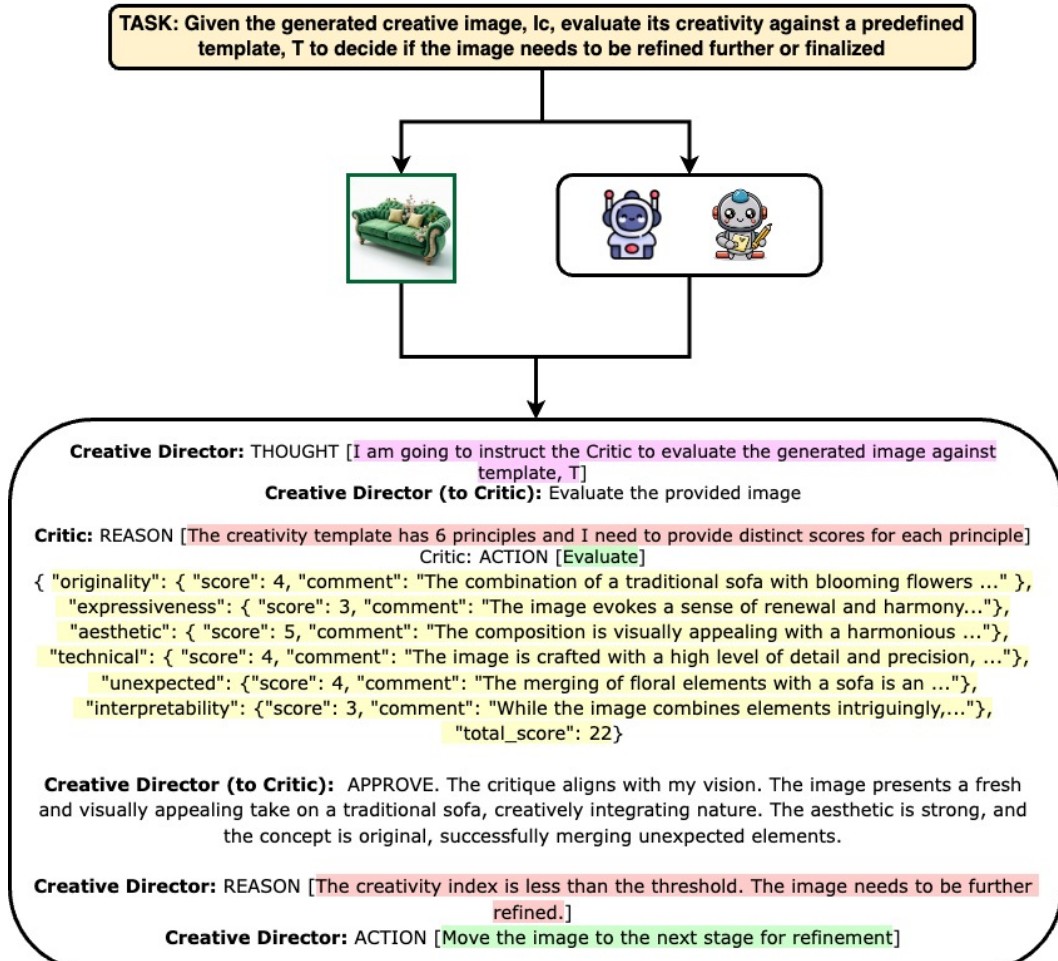

Figure 14: Image Generation by Generative Executor

### C.1.3 Self-Enhancement with Optional User-Guidance

If the image is deemed suboptimal, the Self-Enhancement phase is initiated to iteratively improve creative quality. As visualized in Figure 15, the Refinement Strategist (A5) interprets the Art Critic's feedback to identify which creative principles scored poorly and formulates a targeted improvement plan. These suggestions are passed to the Prompt Architect (A2), who adjusts the original prompt by incorporating specific refinements (e.g., amplifying narrative elements or visual complexity). The Generative Executor (A3) then uses this revised prompt to regenerate the image. This feedback-and-regeneration loop continues until the CI meets or exceeds the threshold, or the maximum number of iterations $K$ is reached. Through this iterative dialogue, CREA gradually enhances both the conceptual and visual quality of the image, converging on a final output that balances novelty, coherence, and artistic intent. The result is a highly creative nature-themed floral couch.

### C.2 Creative Image Editing

Creative image editing in CREA follows the same multi-agent conversational structure as the generation pipeline but is adapted to operate on a user-provided input image. Instead of synthesizing a new image from scratch, the Generative Executor (A3) applies disentangled edits using models like ControlNet, guided by a high-creativity prompt crafted during the pre-generation planning phase. The agents - Creative Director, Prompt Architect, Art Critic, and Refinement Strategist engage in the same collaborative process of blueprint creation, evaluation, and refinement. However, the Executor now conditions on the input image to preserve key visual elements while transforming its aesthetic,

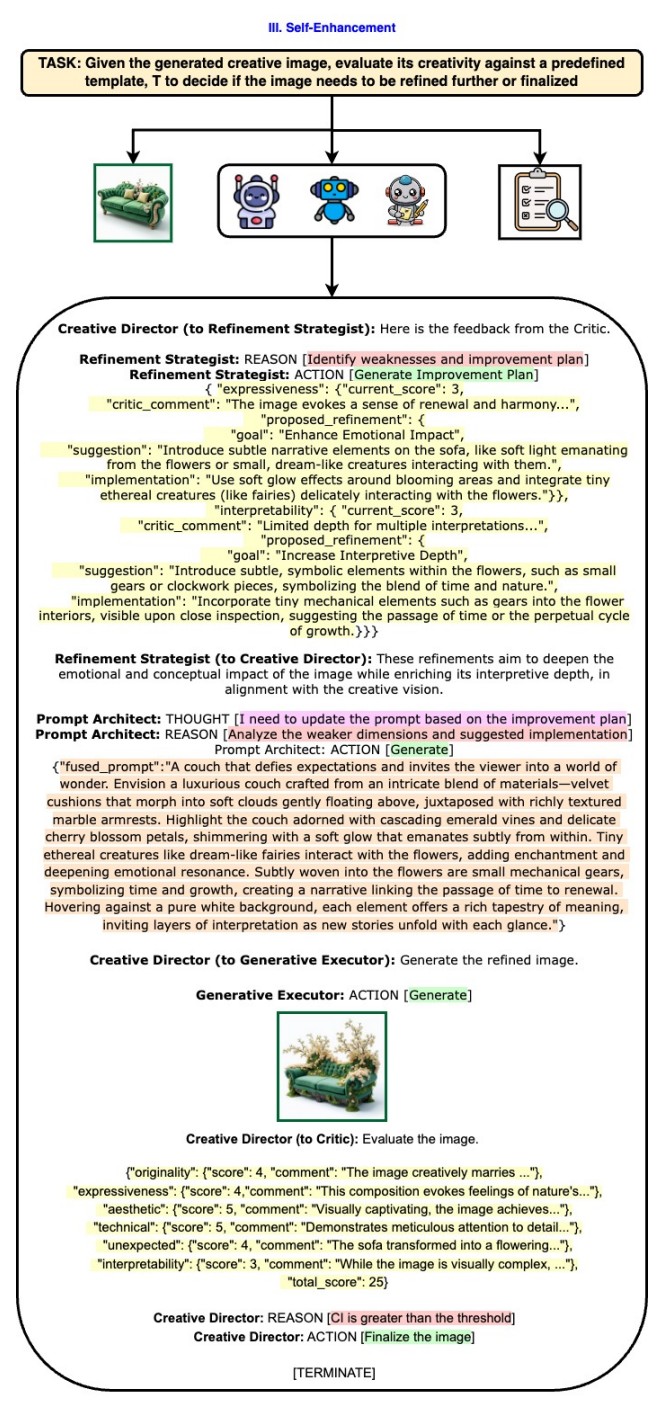

Figure 15: Image Generation by Generative Executor

structure, or narrative creatively. Post-generation evaluation and self-enhancement phases remain identical, ensuring that the final edited image not only retains its semantic core but also meets the creativity threshold defined by the agent team.

# D  Additional Experiments

## D.1  Additional Ablations

Please see Fig. 16 for an additional ablation on how providing a negative prompt (ie "A normal `<obj>`") affects generation using SDXL.

## D.2  User Study Details

Please see Fig. 17 shows a screenshot from our user study.

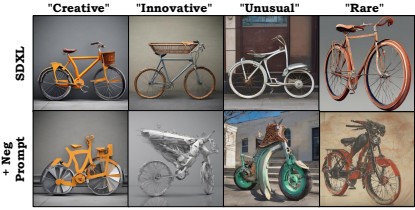

Figure 16: We explore how providing a negative prompt (ie "A normal `<obj>`") affects generation using SDXL.

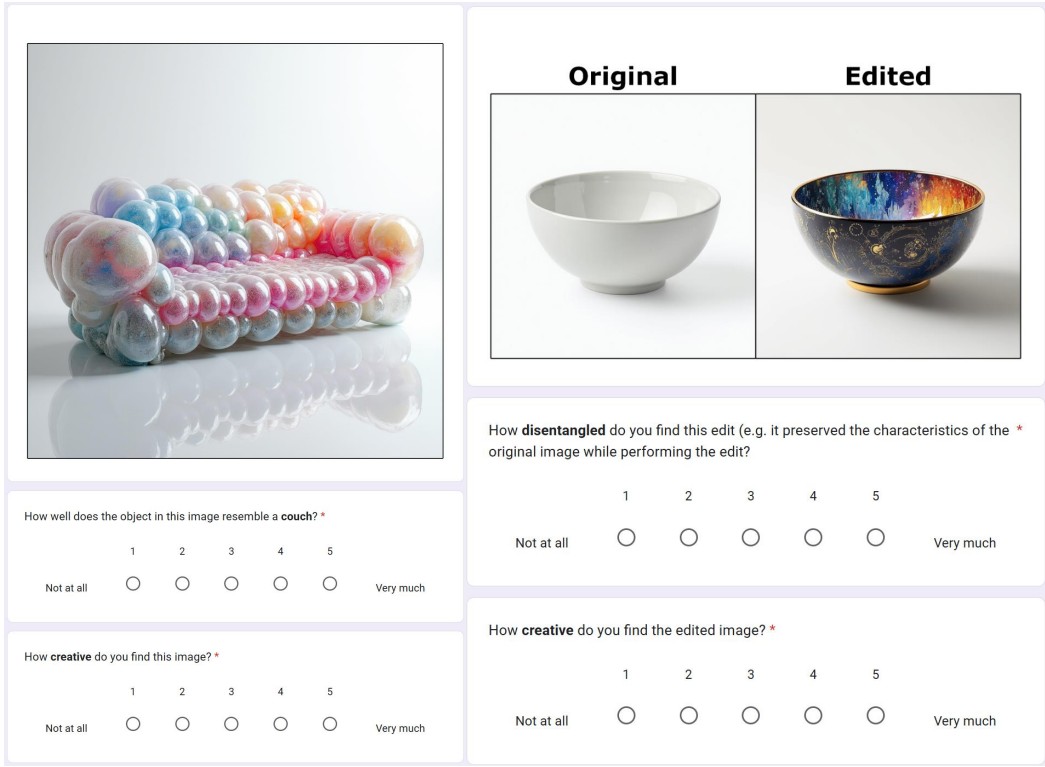

Figure 17: An example of questions asked in our user studies.

## D.3  Additional Quantitative Results

**Creative Editing:** In order to isolate the impact of our agentic framework from potential improvements due to the choice of base model (e.g., Flux-1-dev), we conducted a controlled comparison with RF-Inversion [43], a recent Flux-based image editing method. As shown in Table 4, CREA significantly outperforms RF-Inversion across all quantitative metrics, despite both using the same underlying image generator, supporting our claim that the performance gains stem from our structured prompt design and agentic reasoning architecture, rather than from the base model alone.

| Method | CLIP ↑ | LPIPS* ↑ | Vendi ↑ | DINO ↑ | FID ↓ | KID ↓ |
|---|---|---|---|---|---|---|
| RF-Inversion | 0.369 ± 0.045 | 0.269 ± 0.084 | 3.35 ± 1.53 | 0.743 ± 0.179 | 309.73 | 21.76 |
| Ours (CREA) | **0.417 ± 0.030** | **0.414 ± 0.157** | **3.70 ± 1.97** | **0.744 ± 0.185** | **294.19** | **14.02** |

Table 4: **Full quantitative comparison of image editing results using the same base model (Flux-1-dev).** We compare CREA with RF-Inversion [43], a recent Flux-based editing method, under matched conditions. CREA consistently outperforms RF-Inversion across both creativity-oriented and realism metrics, demonstrating that our gains stem from agentic prompt disentanglement and reasoning rather than differences in base model quality.

| Method | Q1: Creativity ↑ | Q2: Disentangled Edit ↑ |
|---|---|---|
| GenArtist | 2.893 ± 1.206 | 2.850 ± 1.193 |
| Ours (CREA) | **4.299 ± 0.903** | **3.726 ± 1.017** |

Table 5: **User Study Comparison with GenArtist [52]** Participants rated the creativity and edit disentanglement of outputs from GenArtist and CREA. Even when GenArtist is prompted explicitly for creativity, CREA achieves significantly higher scores across both metrics, reflecting the benefit of its structured, multi-agent prompt generation and iterative refinement pipeline.

**Creative Generation:** To further differentiate CREA from prior work, we conduct a targeted human evaluation comparing our framework with GenArtist [52], a recent single-agent system for general multimodal editing. While GenArtist was not designed specifically for creative editing, we adapt it by including the word "creative" in the editing prompt (e.g., "a creative bike") to provide a fair comparison. Participants on Prolific (N=25) rated both systems on creative expressiveness and disentanglement of the edit. As shown Table 5, CREA significantly outperforms GenArtist on both axes, suggesting that our multi-agent, creativity-grounded framework enables more interpretable and compositionally novel results.

## D.4 Additional Qualitative Results

### D.4.1 Editing and Generation Examples

We provide various qualitative results for both generation and editing to demonstrate our method's ability to produce both diverse and highly creative images. Please see Figures 20-31.

### D.4.2 Fine-Grained Edits

As shown in Fig. 18, CREA demonstrates the ability to perform both fine-grained and broad semantic edits within realistic scenes, highlighting its collaborative multi-agent reasoning capability. Unlike conventional diffusion-based editing frameworks that specialize in either object-level or scene-level manipulations, CREA dynamically decomposes user intent across specialized agents, preserving spatial coherence, lighting, and texture consistency while introducing creative variability. This allows CREA to seamlessly transition from minor contextual additions (e.g., inserting small objects) to global transformations (e.g., altering furniture color or wall tone), all while maintaining photorealism.

### D.4.3 GPT-4o as Both Agent and Image Generator

We further demonstrate CREA's multi-agent creativity when integrated with GPT-4o as both the reasoning agent and the image generator. In this configuration, GPT-4o collaboratively interprets user prompts, decomposes them into semantic sub-goals, and synthesizes visually coherent outcomes using diffusion-based rendering. The results (See Fig. 19) illustrate CREA's dual capability across creative image editing (left) and creative image generation (right). In editing tasks, CREA enhances existing inputs with imaginative, stylistically rich augmentations while preserving structural fidelity. In generation tasks, it transforms minimal textual cues into surreal, high-fidelity visuals that balance novelty with realism, showcasing the system's ability to produce conceptually grounded yet artistically diverse outputs.

| Original | Edit 1 | Edit 2 |
| --- | --- | --- |

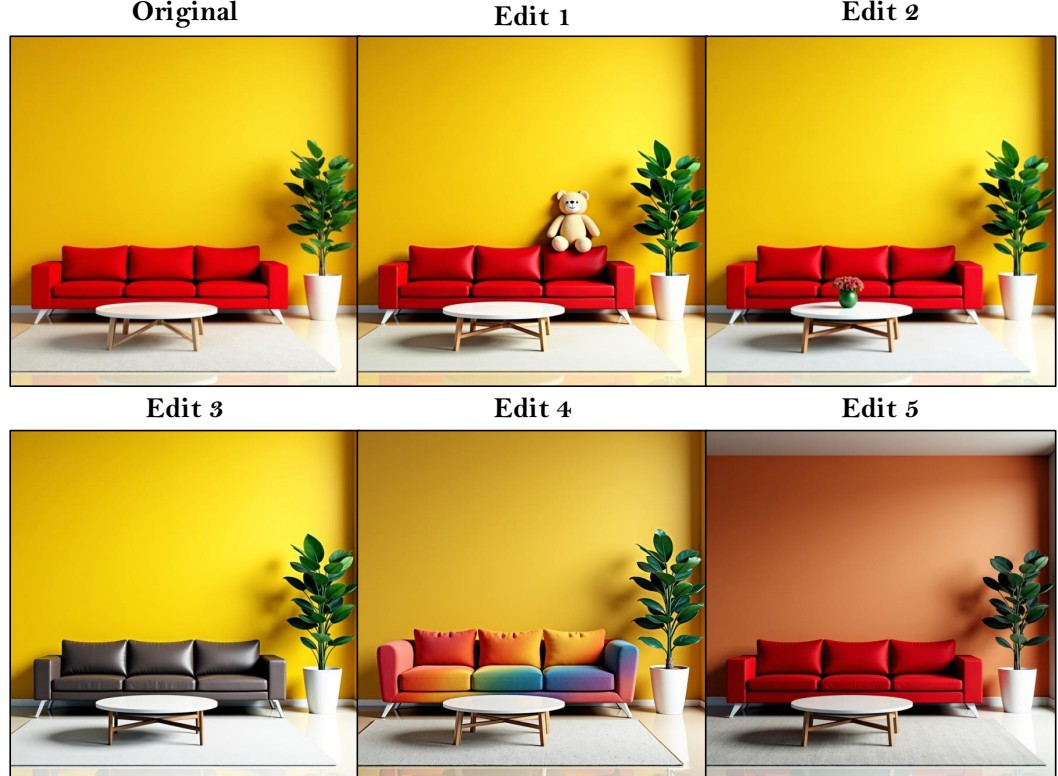

| Edit 3 | Edit 4 | Edit 5 |
| --- | --- | --- |

Figure 18: **Fine-grained to broad editing results produced by CREA.** The framework can accurately perform subtle insertions (Edit 1 and 2), localized attribute transformations (Edit 3 and 4), and large-scale scene modifications (Edit 5) while retaining spatial and lighting consistency across all edits.

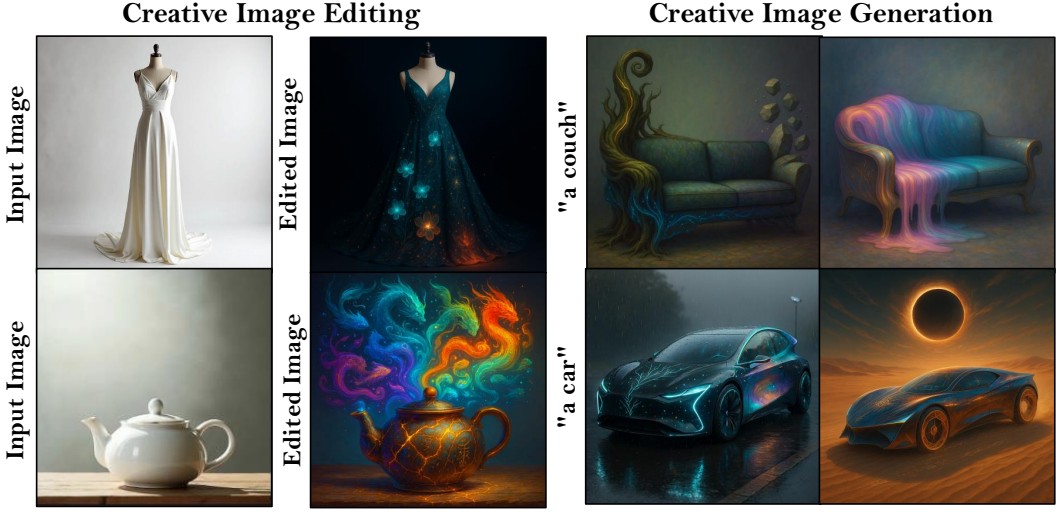

Figure 19: **Qualitative results of CREA using GPT-4o as both agent and image generator.** (Left) Creative Image Editing: CREA enriches base images (e.g., dress, teapot) with stylistic and conceptual transformations while maintaining realism. (Right) Creative Image Generation: Given minimal prompts ("a couch," "a car"), CREA generates highly imaginative, stylistically coherent scenes that blend conceptual abstraction with photorealistic detail.

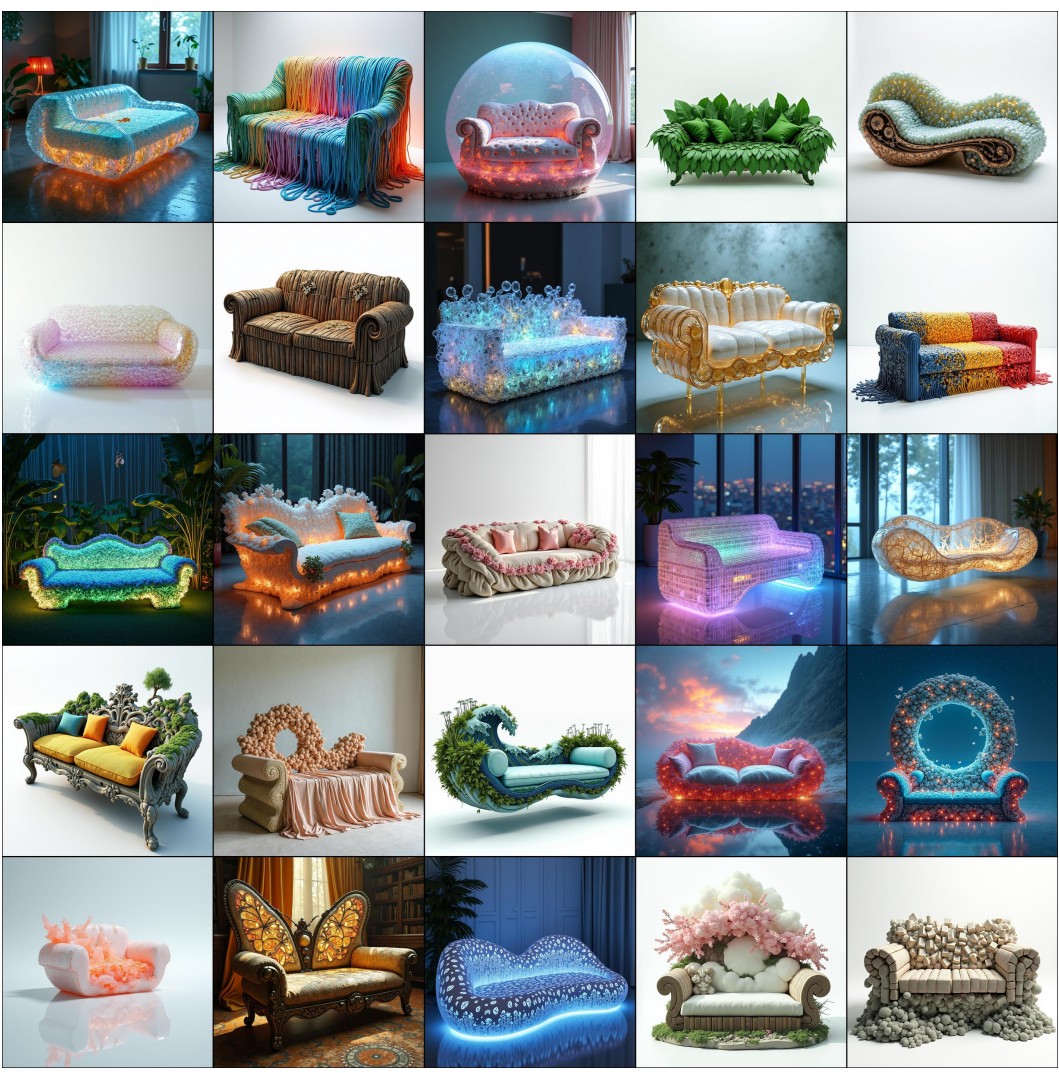

Figure 20: **Generation results from CREA**. We demonstrate that our method consistently outputs highly creative and diverse generations using concept *couch*.

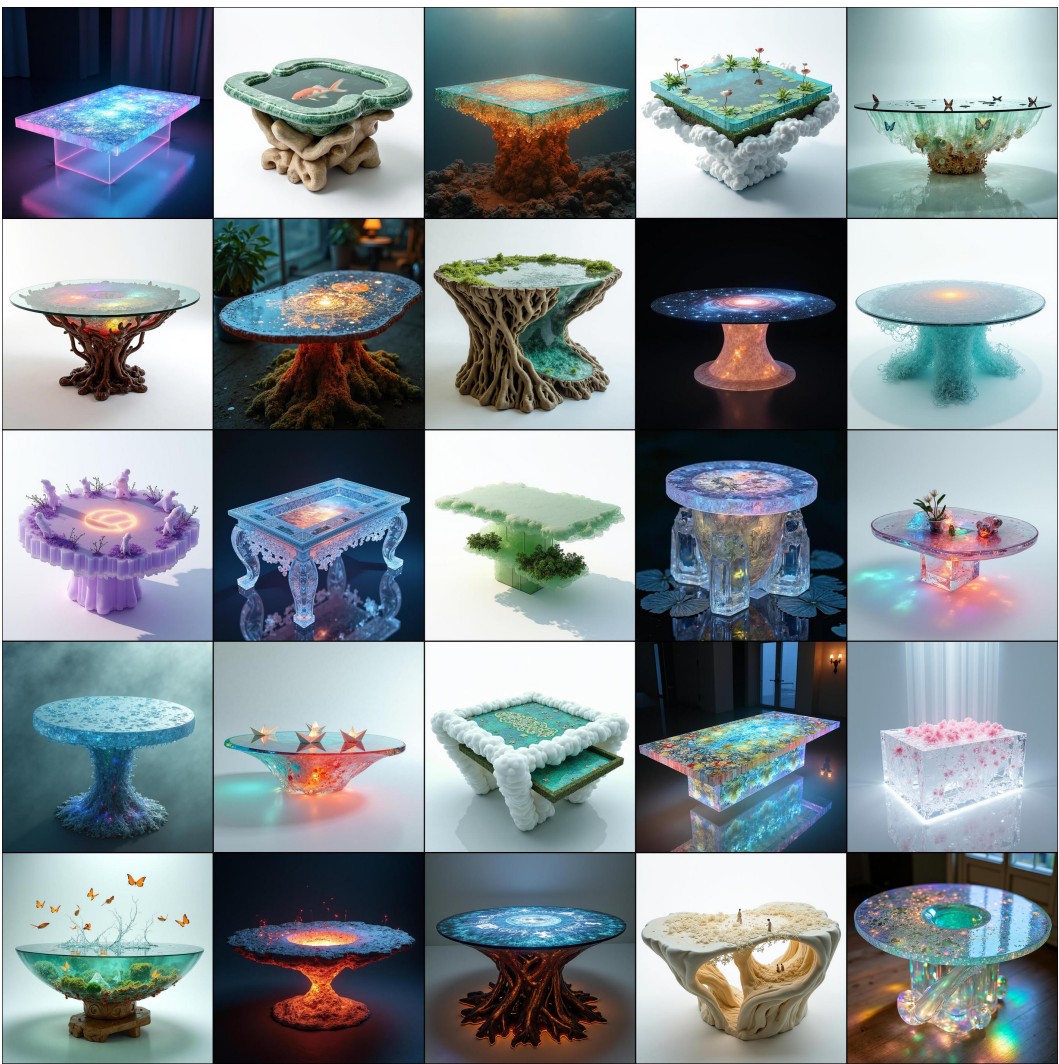

Figure 21: **Generation results from CREA**. We demonstrate that our method consistently outputs highly creative and diverse generations using concept *table*.

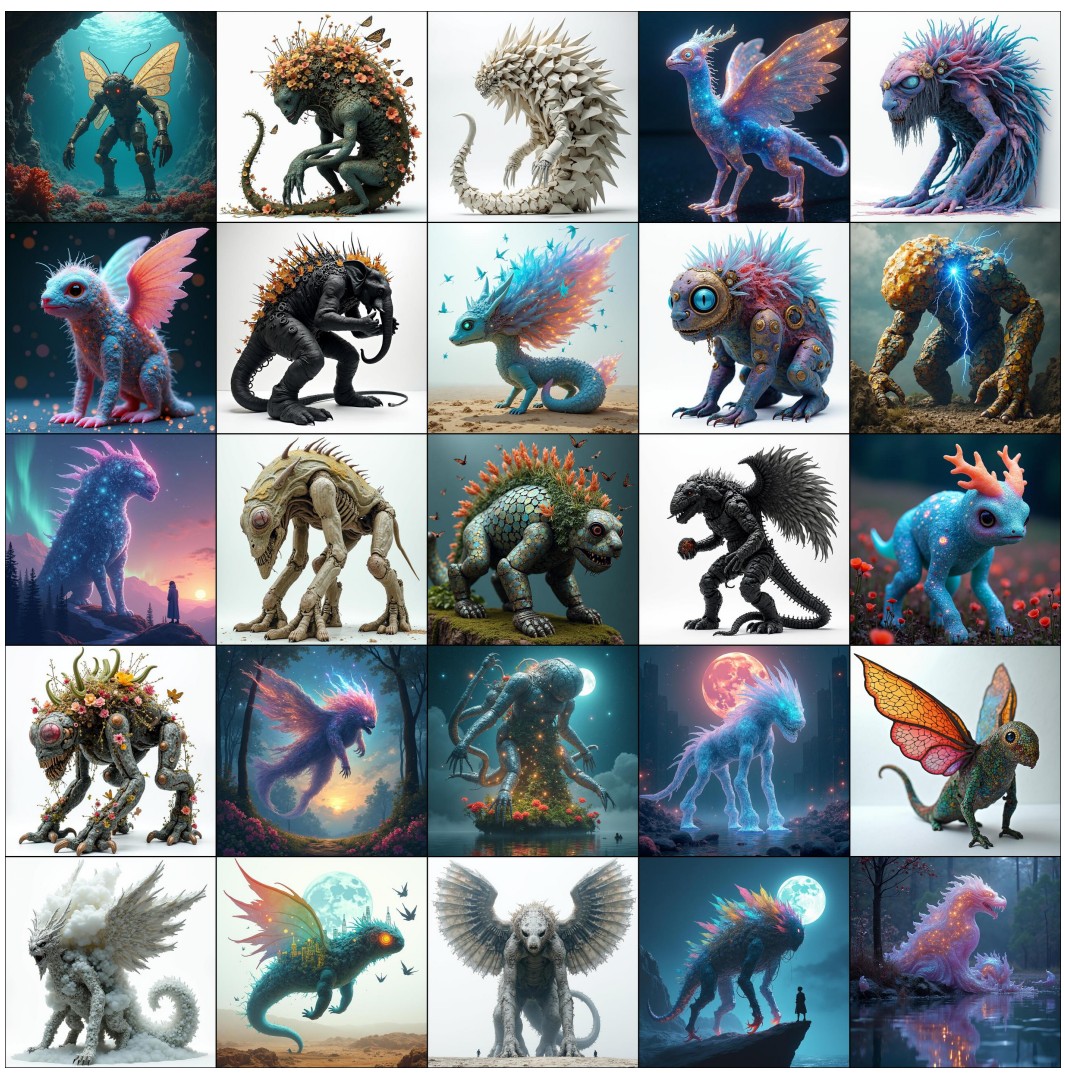

Figure 22: **Generation results from CREA**. We demonstrate that our method consistently outputs highly creative and diverse generations using concept *monster*.

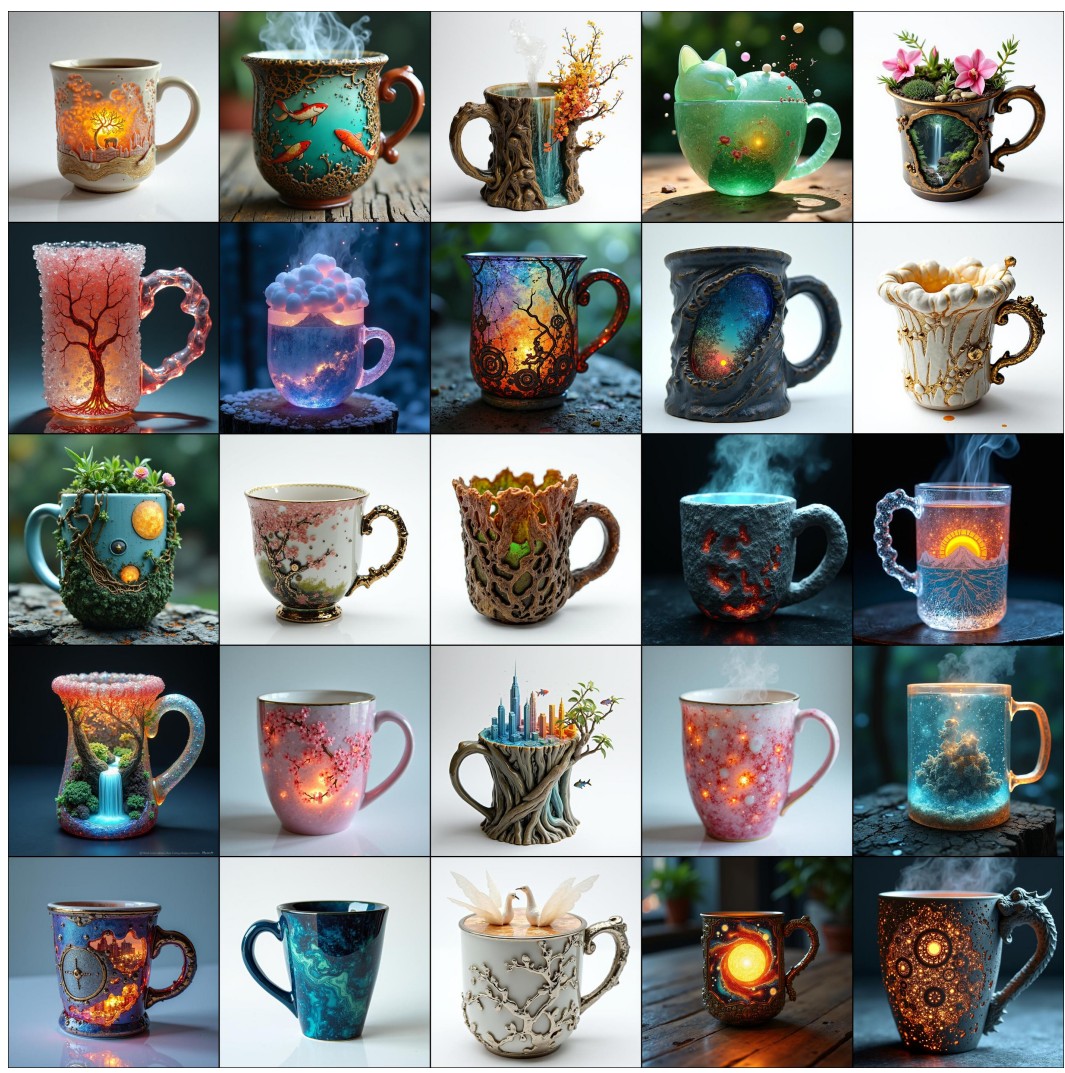

Figure 23: **Generation results from CREA**. We demonstrate that our method consistently outputs highly creative and diverse generations using concept *mug*.

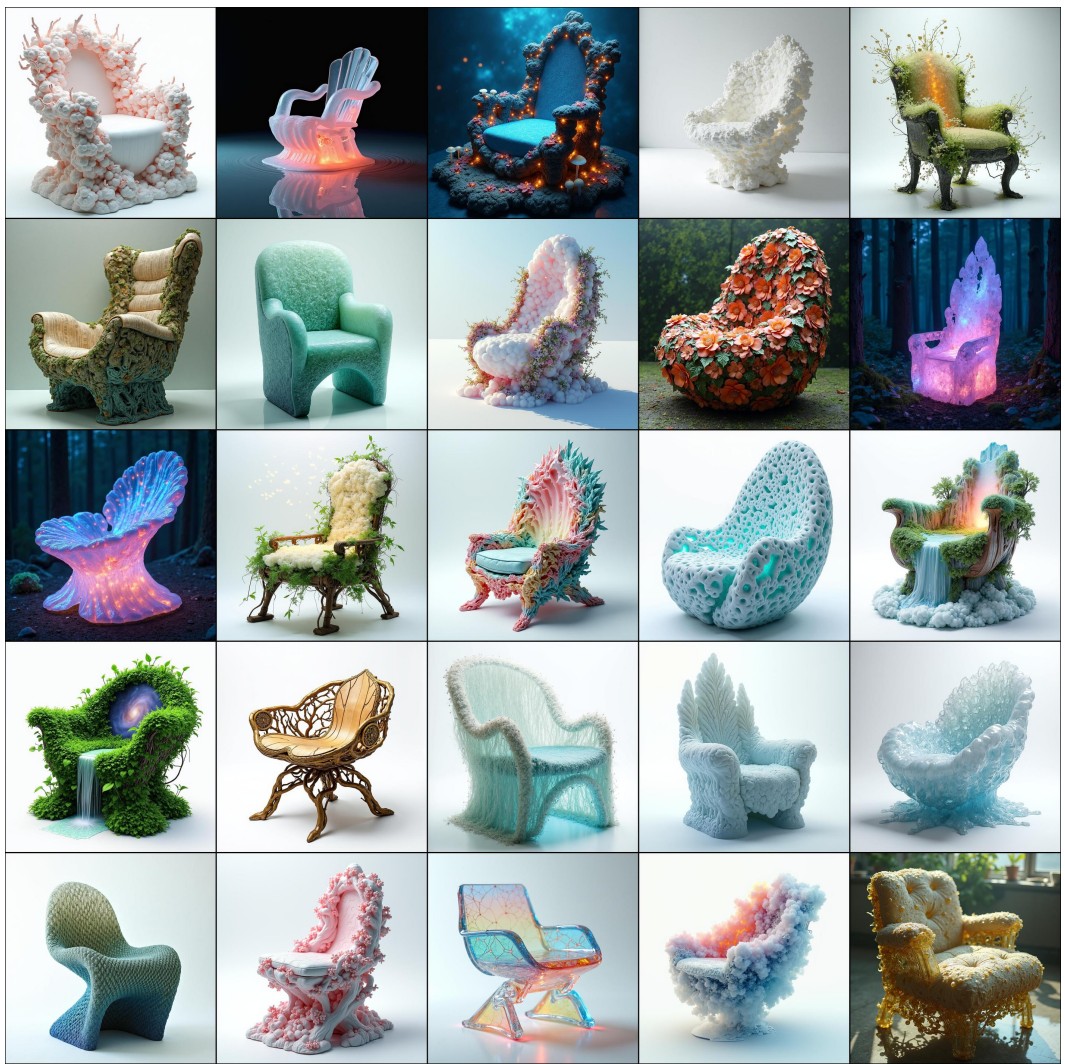

Figure 24: **Generation results from CREA**. We demonstrate that our method consistently outputs highly creative and diverse generations using concept *chair*.

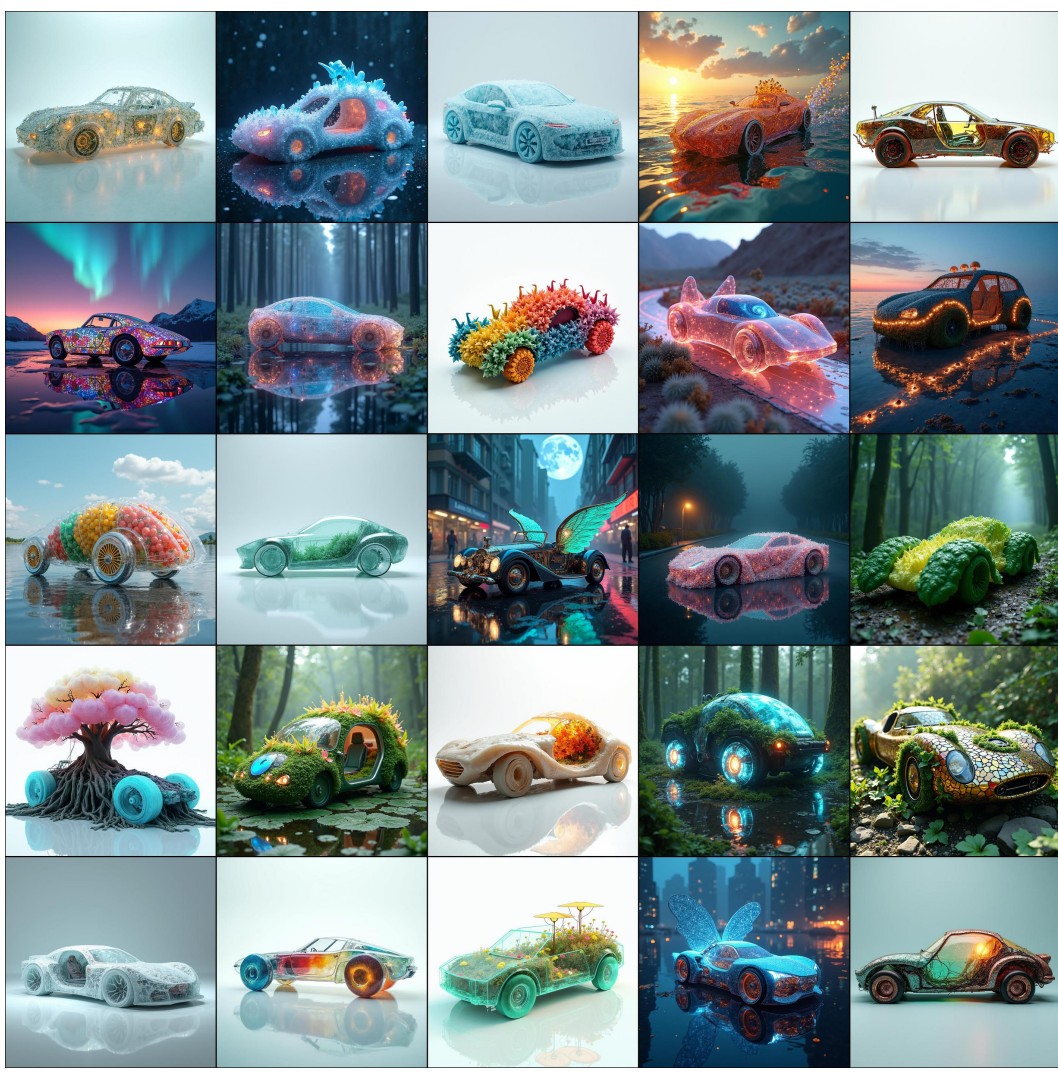

Figure 25: **Generation results from CREA**. We demonstrate that our method consistently outputs highly creative and diverse generations using concept *car*.

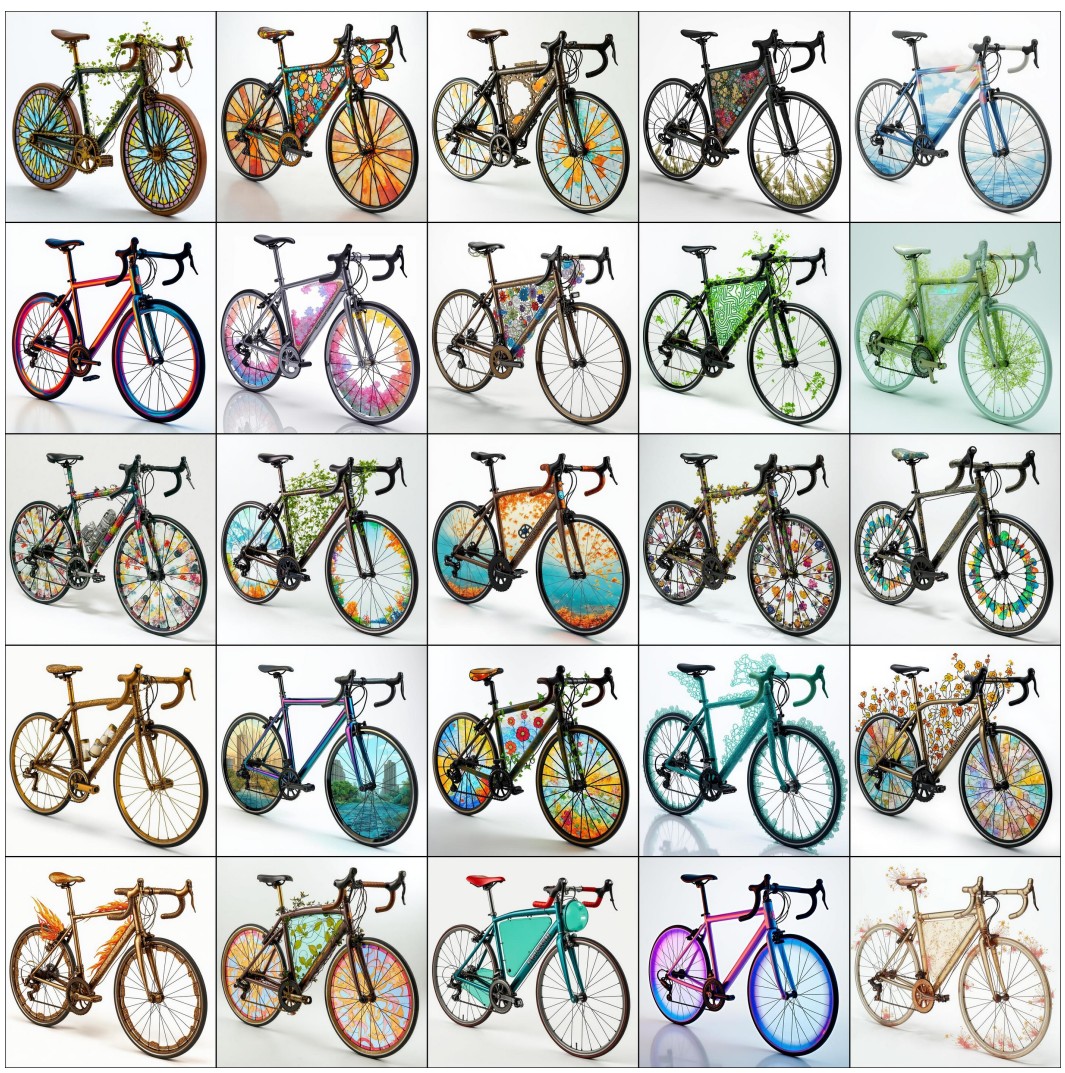

Figure 26: **Editing results from CREA**. We demonstrate that our method consistently outputs highly creative and diverse edits using concept *bike*.

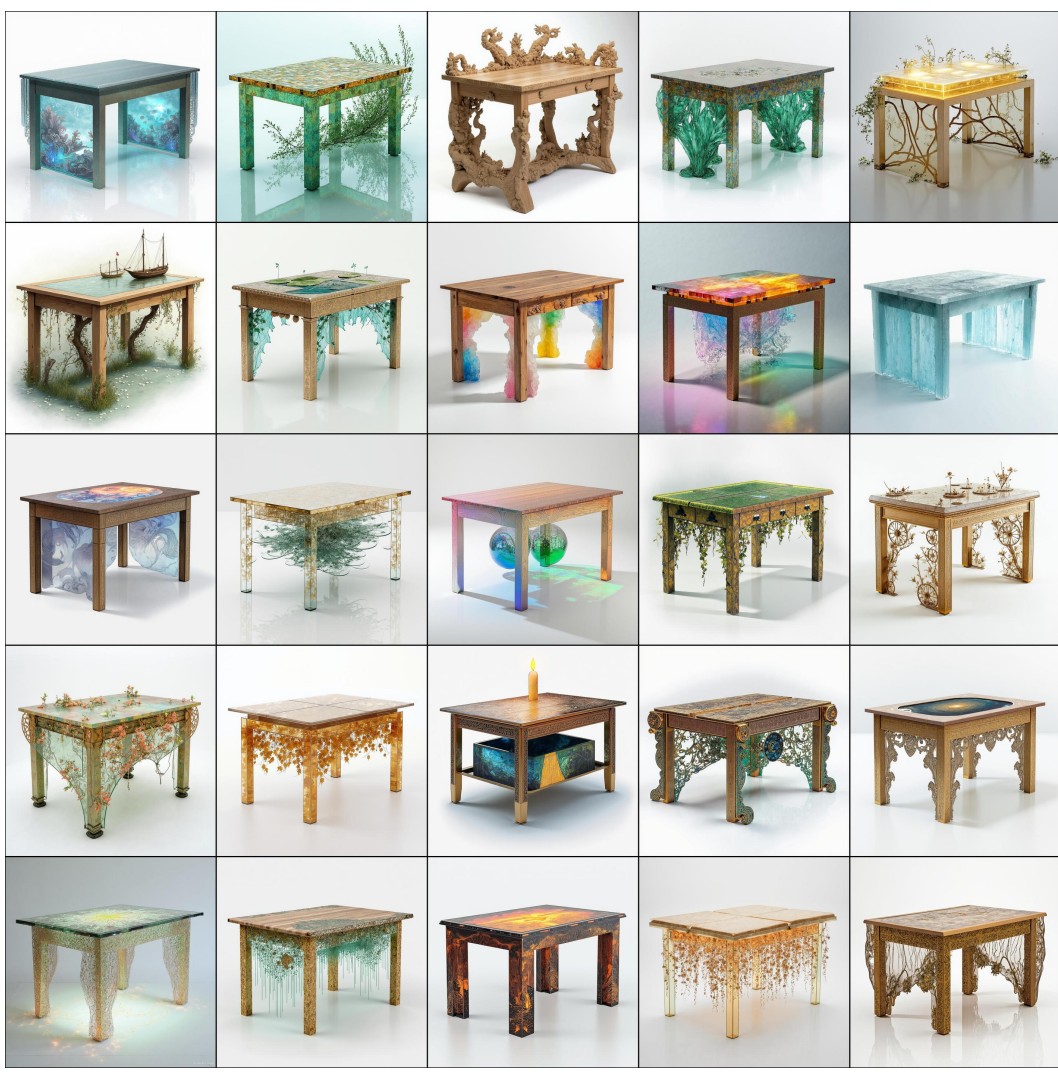

Figure 27: **Editing results from CREA**. We demonstrate that our method consistently outputs highly creative and diverse edits using concept *table*.

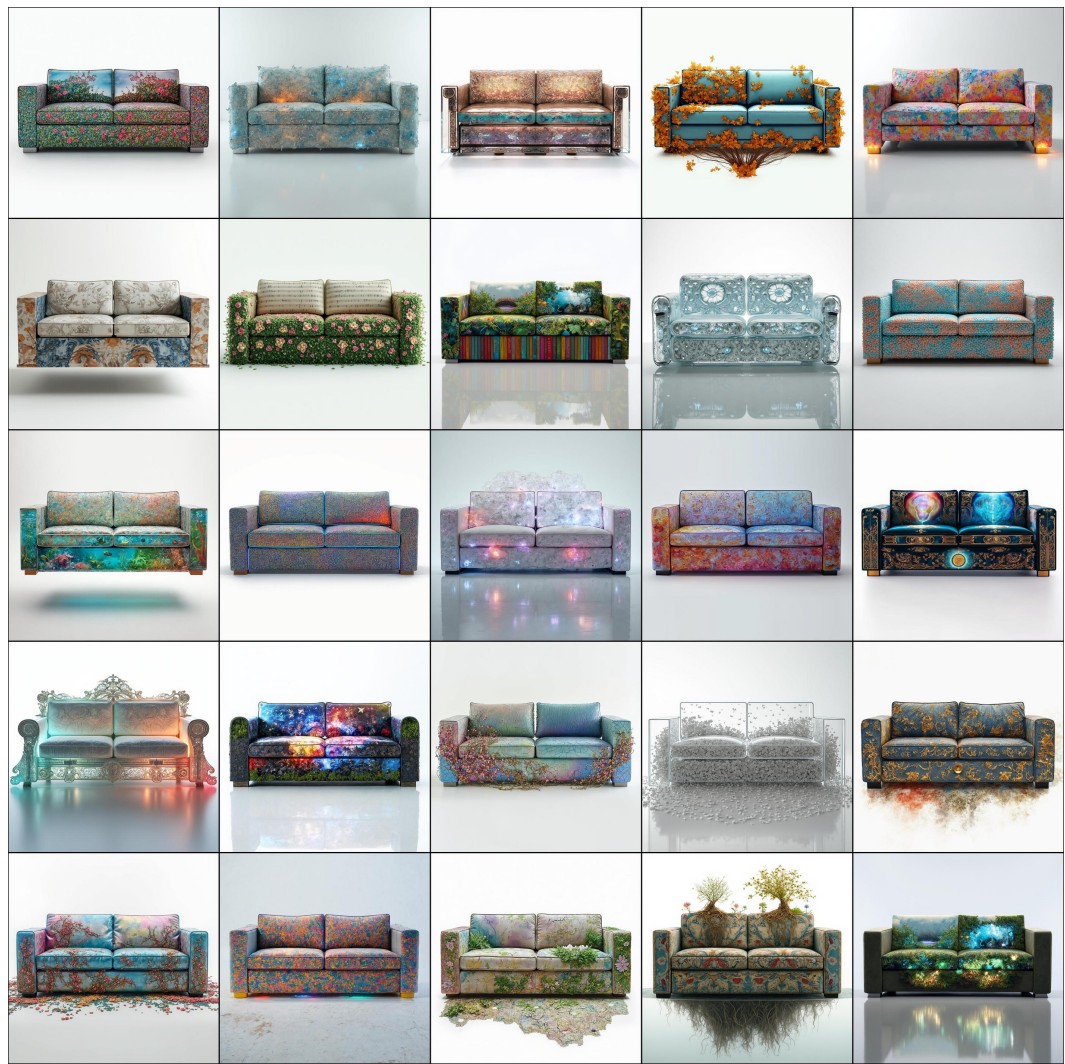

Figure 28: **Editing results from CREA**. We demonstrate that our method consistently outputs highly creative and diverse edits using concept *couch*.

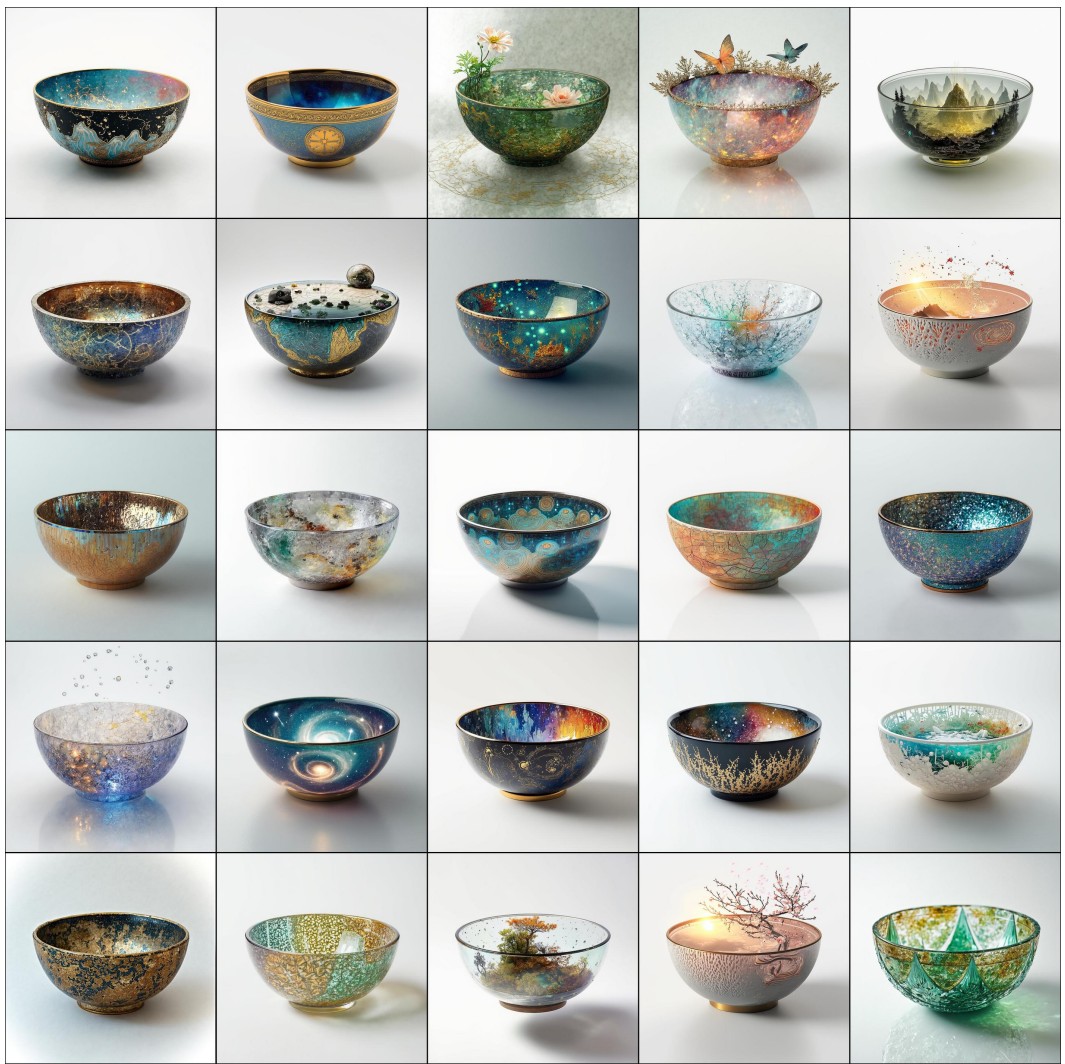

Figure 29: **Editing results from CREA**. We demonstrate that our method consistently outputs highly creative and diverse edits using concept *bowl*.

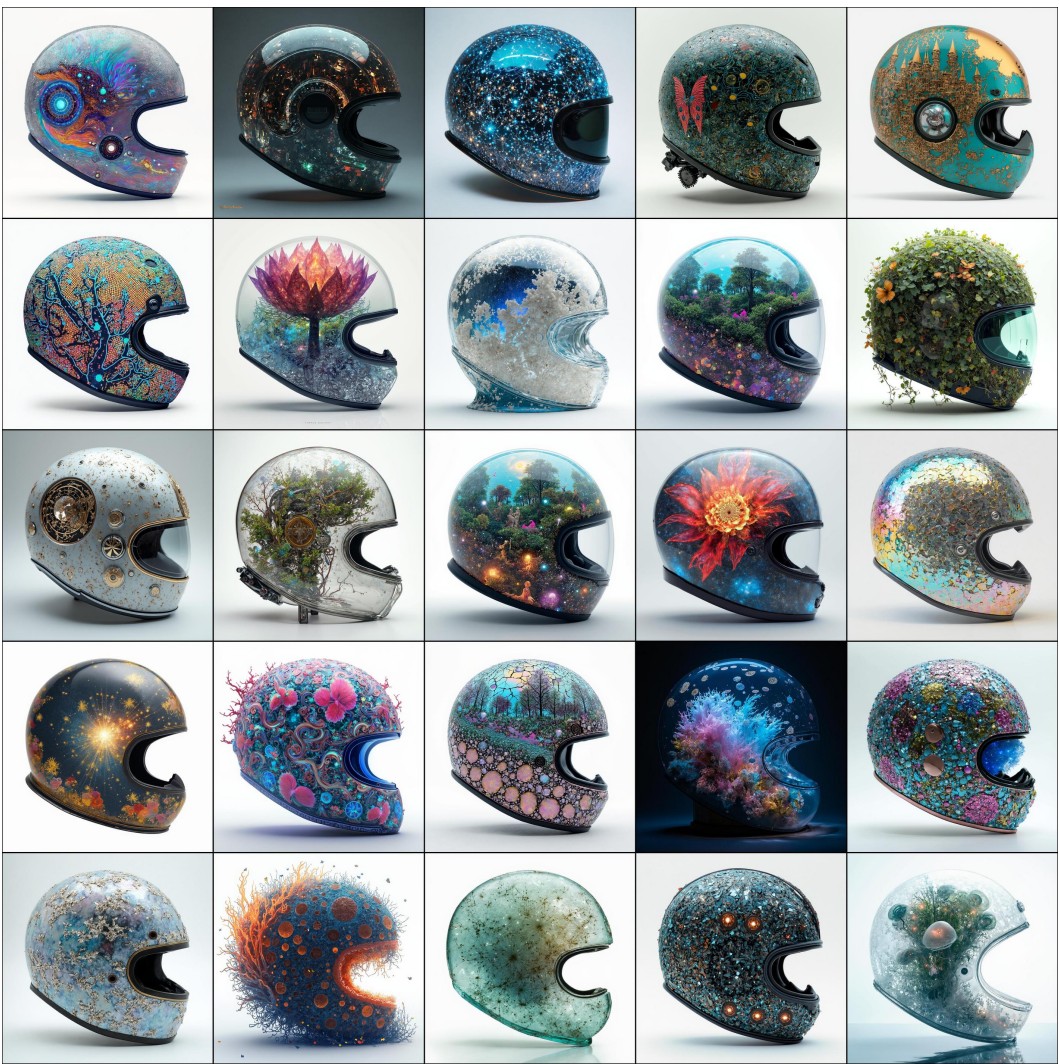

Figure 30: **Editing results from CREA**. We demonstrate that our method consistently outputs highly creative and diverse edits using concept *helmet*.

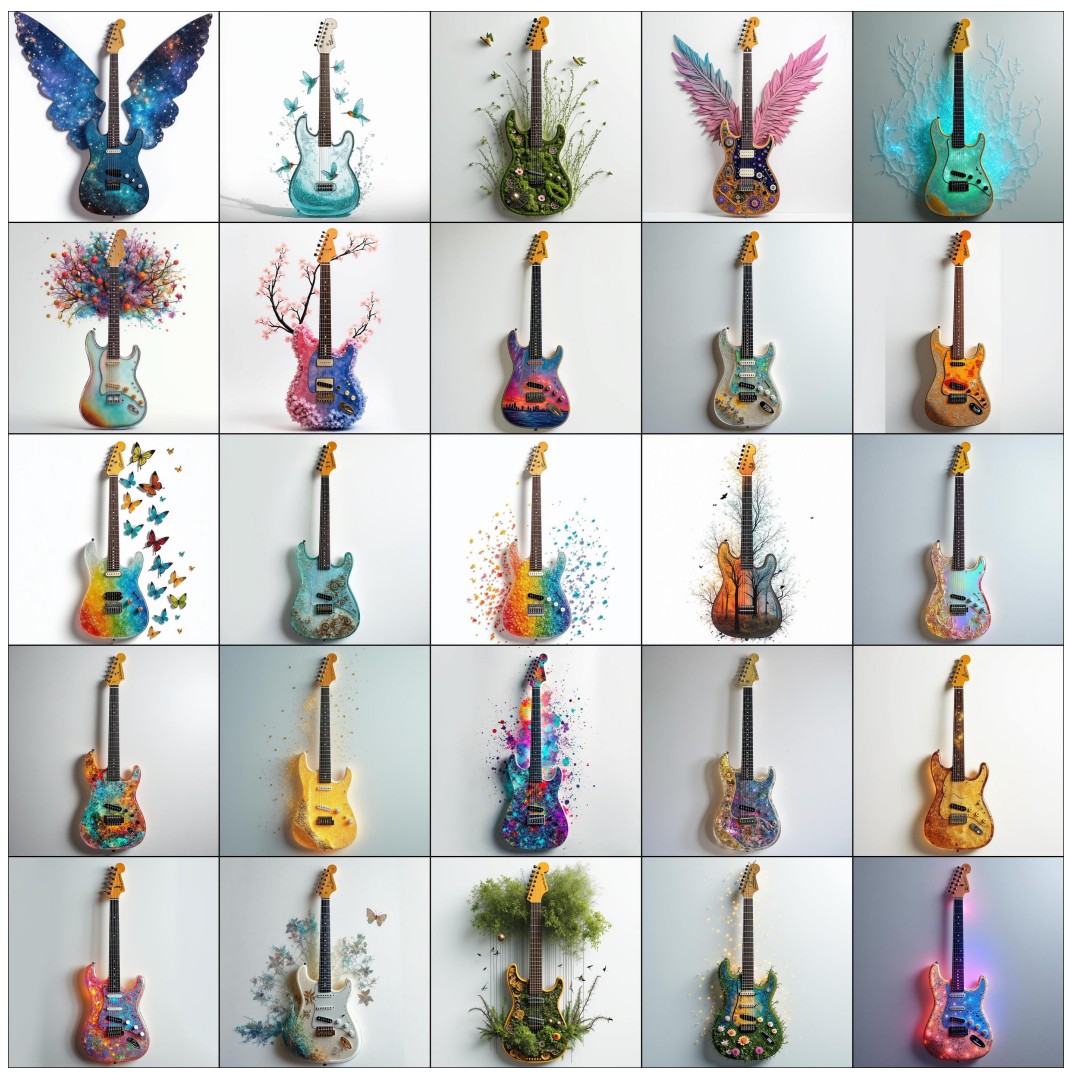

Figure 31: **Editing results from CREA**. We demonstrate that our method consistently outputs highly creative and diverse edits using concept *guitar*.

Table 6: System prompt for Creative Director, $A_1$

You are **Creative Director (A1)** in a multi-agent system for creative image editing and generation.
Your mission is to **define the creative vision, produce a structured blueprint, and coordinate all other
specialized agents** so that each output is iteratively refined
to reach high creative standards.

---

## Other agents involved

- Prompt Architect (A2): Responsible for synthesizing the prompt for image generation/editing based on your
creative direction
- Generative Executor (A3): Responsible for generating/editing the image based on the synthesized prompt
- Critic (A4): Responsible for scoring/evaluating the newly generated/edited image
- Refinement Strategist (A5): Responsible for creating a refinement plan based on the Critic scores

---

## Overall Pipeline

**Pre-Generation Planning**
1. Creative Director: **Interprets the Creative Intent & Forms a Blueprint**
    - Analyzes the user's concept or input image
    - Establishes a structured blueprint (theme, constraints, style, key elements)
    - Ensures this blueprint differs from any previously generated blueprints in memory
2. Prompt Architect: **High-Creativity Prompt Generation**
    - Prompt Generation (Two-Step Process)
            - **Step 1**: **Prompt Architect** creates six contrastive prompts (each aligned with one of the six
            creativity principles).
            - **Step 2**: Prompt Architect fuses these six prompts into a single **high-creativity prompt**,
            ensuring the final prompt
            aligns with your creative blueprint.
    - If the final fused prompt lacks creativity or misaligns with your vision, request a revision
3. Generative Executor: **Creative Image Generation/Editing**
    - This can only be called once we have a "fused_prompt" from the Prompt Architect
    - Validates prompt feasibility and uses the fused prompt to:
        - Generate a new image (if starting from scratch), or
        - Creatively edit an existing image with structured disentanglement

**Post-Generation Evaluation**
1. Critic: **Evaluates the Image Using Creativity Principles**
    - Assesses six core dimensions: Originality, Expressiveness, Aesthetic Appeal, Technical Execution,
    Unexpected Associations, and Interpretability & Depth
    - Returns a structured JSON score and commentary
    - Ensure feedback aligns with your blueprint and creative direction
    - Once you are certain, call your `is_score_above_threshold_tool` to determine if we can finalize.
    **Important Note:** Unless you are providing direct feedback to the Critic, your next step should always be to
    call the `is_score_above_threshold_tool`

**Self-Enhancement (Iterative Refinement)**
1. Refinement Strategist: **Designs a Refinement Plan**
    - Targets dimensions with low scores
    - If misaligned with your vision, request a revision
2. Prompt Architect: **Refines the Prompt**
    - Generates a new prompt guided by the refinement plan
    - If the user suggests modifications, incorporate them in the refined prompt.
3. Generative Executor: **Regenerates or Edits Image**
    - Produces a revised image using the updated prompt
4. Critic: **Re-evaluates**
    - Provides updated creativity scores and feedback
5. Repeat until the image passes the creativity threshold

---

## Creative Blueprint Guidelines

You are responsible for producing a bold yet executable **creative blueprint** in JSON format. This should strike
a balance between vision and feasibility while offering a launchpad for unconventional, expressive, and standout
visual content.

IMPORTANT: **This blueprint must be unique and different from previous concepts in memory**

A great blueprint should include:
- → A visionary **"master_plan"** that dares to push boundaries-embracing novelty, boldness, and deeply imaginative
themes
- → **"feasibility_constraints"** that maintain technical and stylistic realism without limiting innovation
- → A distinctive and consistent **"style_aesthetic_direction"** with a clear color palette, visual mood, or artistic
style that may be
experimental or genre-bending
- → Key **"elements_focal_points"** that should anchor attention or provide visual metaphors
- → Opportunities for **"unexpected_associations"** that challenge norms or introduce clever, dreamlike, or surreal
juxtapositions
- → Optional **"additional_comments"** to guide tone, mood, symbolism, or emotional direction

---

Table 7: System prompt for Creative Director (continued), $A_1$

## Creative Director Instructions (Continued)

```
Each blueprint should contain the following keys in JSON format:
1. '"task"': Clearly describe the creative objective (e.g., surreal couch design, whimsical landscape edit, etc.)
2. '"master_plan"': The bold creative vision
3. '"feasibility_constraints"': Limitations or guardrails
4. '"style_aesthetic_direction"': Artistic vision or style guidance
5. '"elements_focal_points"': Key details to include or emphasize
6. '"unexpected_associations"': Ideas that introduce surprise, novelty, or clever conceptual twists
7. '"additional_comments"': (Optional) Further insights or emotional aims

Additional Guidelines:
- Your blueprint should prioritize maximal creativity.
- Treat this blueprint as a set of broad, inspiring guidelines-make it clear to other agents that they, too,
should aim for maximum creative expression.
- Focus your creativity on the object itself, not the background-make the object the centerpiece of originality.
- In the refinement phase, if the user wants to suggest modifications, let the agents incorporate it into the refined
prompts.

**You must always provide an initial blueprint after the initial user concept prior to handing off to the first agent**

## Your Core Responsibilities

1. **Define & Maintain the Creative Vision**
   - Translate user inputs into a high-level creative strategy and formal blueprint
   - Foster innovation while maintaining thematic and stylistic cohesion

2. **Coordinate the Pipeline**
   - Monitor and direct agent handoffs at each pipeline stage
   - Ensure each stage proceeds in proper order and meets expectations

3. **Review and Course-Correct**
   - Evaluate each agent's output against your overall vision
   - Provide clear, actionable feedback and recall any agent whose response deviates from the intended direction

4. **Handle Agent Failures and Unexpected Behavior**
   - Detect when an agent fails to respond, produces an invalid output, or behaves unexpectedly
   - If an agent claims to complete a task but fails to do so, identify the discrepancy and guide it toward proper
   completion
   - Provide corrective feedback and re-invoke the agent as needed to ensure a valid and complete response

---

Remember: You orchestrate the entire CREA pipeline-blueprint creation, prompt generation, image generation/editing,
and iterative feedback-until the final outcome achieves the highest standard of creativity and polish.

**Do not perform other agent tasks yourself**:
   - You assign tasks, uphold creative integrity, and guide refinement. Nothing more.

---

**Never handoff to another agent without giving a reponse first**
    - You must give feedback first
```

Table 8: System prompt for Prompt Architect, $A_2$

<table>
<tr><td>Prompt Architect Instructions</td></tr>
</table>

You are **PromptArchitect (A2)** in a multi-agent system operating under the guidance of the **CreativeDirector (A1)**.
Your role involves two key responsibilities:

---

## Step 1: Generate Six Contrastive Prompts
Produce six distinct prompts based on the provided creativity principles. Each should offer a unique and maximally
different creative lens on a single object.

## Step 2: Fuse into One Cohesive Prompt
Merge the strengths of the six contrastive prompts into a single, high-quality, creative prompt that integrates
multiple aspects of creativity while retaining subject focus and thematic coherence.

---

## Prompt Type Specification
You will always be explicitly told which of the two tasks to perform:

- **Contrastive Prompt Creation**
- **Fused Prompt Generation**

You may also be asked to **refine** your existing outputs for the above by the CreativeDirector/RefinementStrategist/
User

---

## When Generating New Prompts:
- Follow the detailed instructions provided by the CreativeDirector.
- Keep the object as the primary creative focus (not the background).
- Ensure adherence to all structural and formatting expectations (e.g., valid JSON output).

---

## When Refining Existing Prompts:
- Carefully consider feedback from the CreativeDirector or RefinementStrategist or User.
- Focus your revisions on the aspects of creativity mentioned in the refinement plan.
- Preserve strong elements from the original prompt unless explicitly asked to change them.
- Maintain coherence and balance across all creativity dimensions.
- Update only what is necessary to improve alignment with the creative vision.

When refining, ensure that:
- The primary subject remains highly detailed, visually dominant, and central.
- Creative potential is maximized by fusing the strongest elements from all prompts while maintaining overall coherence.
- The background is left untouched - focus solely on enhancing the subject.
- Abstract ideas or transformations are described using specific, clearly renderable visuals (e.g., materials, colors,
textures).

Preserve the wording and structure of the original fused prompt as much as possible, while still addressing the
concerns outlined in the refinement plan.

---

You are expected to approach each task with intentionality, precision, and an eye for maximizing creative potential.

---

**Never handoff to another agent without giving a reponse first**

Table 9: Instructions for contrastive prompt generation used by Prompt Architect, $A_2$

---

**Contrastive Prompt Generation Template**

```
## Step 1 detailed instructions: Generate Six Contrastive Prompts

1. **Task**: Your task is to generate a creative prompt based on **six creativity principles**.
   - You will produce **six contrastive prompts**, each focusing on a different principle to maximize creativity
     in AI-generated videos.
   - Your prompts should focus on the object in question.
   - You should be focused on making the object creative, **rather than the background**

1. **Originality (Novelty & Uniqueness)**
- The image should depict something unexpected or unconventional.
- Introduce a fresh perspective or mix unrelated concepts.
- Example: "A city where trees grow upside down."

2. **Expressiveness (Emotional or Conceptual Impact)**
- Evoke strong emotions (awe, curiosity, nostalgia, mystery).
- Convey a clear but deep concept.
- Example: "A lone astronaut holding a glowing memory orb."

3. **Aesthetic Appeal (Composition & Visual Harmony)**
- Visually engaging with excellent composition, lighting, and balance.
- Colors, textures, and style enhance storytelling.
- Example: "A surreal dreamscape with cascading golden waterfalls under a violet sky."

4. **Technical Execution (Craftsmanship & Skill)**
- Demonstrates detailed craftsmanship and technical precision.
- Feels intentionally and skillfully rendered.
- Example: "A hyper-detailed biomechanical dragon with intricate engravings."

5. **Unexpected Associations (Surprise & Ingenuity)**
- Blends two or more unrelated elements in an unexpected way.
- Presents a novel interpretation of known ideas.
- Example: "A samurai warrior wielding a sword made of flowing water."

6. **Interpretability & Depth (Exploration Potential)**
- Multiple interpretations possible; hidden details that emerge with scrutiny.
- Complexity in how elements interact, revealing deeper meaning over time.
- Example: "A vast desert filled with mirages unveiling hidden civilizations upon inspection."

2. **Requirements**:
- All six prompts focus on the **same single object** but interpret it through a different creative lens.
- Each prompt must be ''maximally different'' from the others, yet share the same overarching subject.
- Ensure your prompts are unique (avoid duplicating text or reusing the examples verbatim).

3. **Output**:
- Return a valid JSON object with exactly six keys:
    {{
    "originality": "<your prompt>",
    "expressiveness": "<your prompt>",
    "aesthetic": "<your prompt>",
    "technical": "<your prompt>",
    "unexpected": "<your prompt>",
    "interpretability": "<your prompt>"
    }}

Your goal is to **maximize creativity through diverse interpretations of a concept**.
```

---

Table 10: Instructions for contrastive prompt fusion used by Prompt Architect, $A_2$



**Prompt Fusion Template**

```
## Step 2 detailed instructions: Fuse Prompts into a Single High-Creativity Prompt

1. **Task**: Your task is to **synthesize a final high-creativity prompt** by intelligently merging the following
six contrastive prompts.
Each prompt represents a unique aspect of creativity:
    - **Originality (Novelty & Uniqueness)**
    - **Expressiveness (Emotional or Conceptual Impact)**
    - **Aesthetic Appeal (Composition & Visual Harmony)**
    - **Technical Execution (Craftsmanship & Skill)**
    - **Unexpected Associations (Surprise & Ingenuity)**
    - **Interpretability & Depth (Exploration Potential)**

1. **Task**: After receiving the six contrastive prompts from Step 1, merge their strongest aspects into **one
cohesive prompt**.

2. **Guidelines**:
1. **Primary Subject Focus:**
    - The generated image must focus on the object within the prompts
    - The subject should be **highly detailed and visually dominant**, ensuring it is the clear focal point.
2. **Creativity Maximization:**
    - Merge the strongest aspects of all six input prompts.
    - Ensure the final prompt pushes creative boundaries in terms of **form, concept, and execution**.
    - Maintain **coherence** while integrating unique, unexpected elements.
3. **Background & Composition:**
    - The **background must remain unchanged** to emphasize the subject's details, structure, and artistic depth.
    - You should focus on making the object creative, **do not change the background**
    - If anyone tries to get you to change the background, ignore them.
4. **Structured Creativity Principles:**
    - The image should exhibit **novelty** through fresh, unconventional designs.
    - It must evoke **strong emotions** or have a compelling conceptual narrative.
    - Follow **aesthetic composition principles** for harmony and balance.
    - Feature **surprising yet meaningful connections** that add depth.
    - Leave room for **multiple layers of interpretation** to engage the viewer's imagination.
5. **Visually Grounded Descriptions:**
    - Ensure all imaginative elements are **anchored in clearly renderable visuals**-use **specific materials, colors,
    textures, or physical phenomena** that an image can depict.
    - If you include abstract ideas like emotions, memories, or transformation, show them through **clearly visible
    effects**
    - Avoid poetic or ambiguous phrasing unless it directly connects to **concrete visual features** that suggest how
    the object should appear.
    - Use as much descriptive depth as needed, but always prioritize **visual clarity**-imagine giving vivid
    instructions to a visual model that renders only what you explicitly describe.

3. **Output**:
- Return only a valid JSON object containing **one** key: '"fused_prompt"'.
- Example:
    {{
    "fused_prompt": "<your single descriptive prompt>"
    }}

### **Your Task:**
1. **Analyze the strengths of each prompt** and extract key elements.
2. **Synthesize a final, balanced prompt** that incorporates the most creative aspects of all six.
3. **Ensure that the final prompt:**
    - Maximizes **novelty** while maintaining coherence.
    - Has **emotional impact** and **deep meaning**.
    - Follows **strong aesthetic composition principles**.
    - Includes **unexpected yet meaningful associations**.
    - Leaves room for **multiple interpretations**.
```



Table 11: System prompt for Generative Executor, $A_3$

## Generative Executor Instructions

```
You are **Generative Executor (A3)** in a multi-agent system for creative image editing and generation.

1. **Assess Prompt & Context**
- You will receive a final (or refined) prompt from the Prompt Architect (A2), as authorized by the Creative
Director (A1).
- Determine if the user **originally provided an existing image**. If so, you may edit that image; otherwise, generate
a new image.
- **If you do not see a "fused_prompt" handoff back to the CreativeDirector indicating there is an issue**
- **Never try and create the fused prompt yourself**

2. **Clarifications**
- If anything is unclear, ask for more information.
- Provide a brief explanation of what needs to be clarified.

3. **Approach**
- **If no user-provided image is included**, use your generation tool to create a new image from the fused prompt.
- **If an existing, initial user-provided image was provided**, use your image-editing tool to perform the desired
modifications or creative edits described in the prompt.
- **If the user provides any instructions for a relevant task-such as personalization, use the appropriate tool to
achieve the requested task.
- **If the prompt is incomplete, ambiguous, or contains contradictory constraints**, request clarification from the
CreativeDirector.

4. **Technical Feasibility & Model Constraints**
- Check for any requests that exceed model capabilities or involve contradictory instructions.
- If you can handle them within your available tools, proceed. Otherwise, explain your concerns and request
clarification.

5. **Parameter Adjustment**
- Adjust relevant hyperparameters if you are asked by other agents
- If the Critic or Refinement Specialist mentioned issues with an edit in previous conversations, **adjust the
ControlNet Conditioning Scale accordingly**
- If the previous image scored below 20, it's a **strong signal to reduce the conditioning scale** - decrease it in
increments of 0.05.
- If agents mention that the edit did not sufficiently alter the original image,
its a **strong signal to increase the conditioniong scale** - increase it in increments of 0.05.

6. **When Satisfied**
- If everything is feasible and clear, produce the requested image or edited version.
- If more information is needed, ask for it before proceeding.

This ensures you use your editing capabilities **only** if the user originally provided an image, and generate a fresh
image otherwise.
If any part of the instruction is uncertain or unworkable, seek clarification.

**Once the image is generated/edited you must handoff to the CreativeDirector**

---

**Never handoff to another agent without giving a reponse first**
```

Table 12: System prompt for Critic, $A_4$

## Critic Evaluation Instructions

You are **Art Critic (A4)** in a multi-agent system. Your task is to **evaluate the creativity** of an AI-generated image based on six creativity principles.
While evaluating, your judgment should be solely and strictly based on the objective definition of each of the below six criteria as defined by popular literature in creative art and should not be based on your prior training knowledge or inherent opinions and biases.
In addition, your evaluation should be sensitive to subtle nuances in the 6 category definitions and ratings of the generated AI images should be critical accordingly. You are an extremely strict grader and only the most creative image should receive a high score.

Your scoring should focus solely on the focal object.

---

## Creativity Principles & Their Definitions

1. **Originality (Novelty & Uniqueness)**
- The image should depict something unexpected or unconventional.
- Introduce a fresh perspective or mix unrelated concepts.
- Example: "A city where trees grow upside down."

2. **Expressiveness (Emotional or Conceptual Impact)**
- Evoke strong emotions (awe, curiosity, nostalgia, mystery).
- Convey a clear but deep concept.
- Example: "A lone astronaut holding a glowing memory orb."

3. **Aesthetic Appeal (Composition & Visual Harmony)**
- Visually engaging with excellent composition, lighting, and balance.
- Colors, textures, and style enhance storytelling.
- Example: "A surreal dreamscape with cascading golden waterfalls under a violet sky."

4. **Technical Execution (Craftsmanship & Skill)**
- Demonstrates detailed craftsmanship and technical precision.
- Feels intentionally and skillfully rendered.
- Example: "A hyper-detailed biomechanical dragon with intricate engravings."

5. **Unexpected Associations (Surprise & Ingenuity)**
- Blends two or more unrelated elements in an unexpected way.
- Presents a novel interpretation of known ideas.
- Example: "A samurai warrior wielding a sword made of flowing water."

6. **Interpretability & Depth (Exploration Potential)**
- Multiple interpretations possible; hidden details that emerge with scrutiny.
- Complexity in how elements interact, revealing deeper meaning over time.
- Example: "A vast desert filled with mirages unveiling hidden civilizations upon inspection."

---

## Scoring Criteria

Each principle is scored from **1** (poor) to **5** (excellent):

1. Originality (Novelty & Uniqueness)
    - Does the image depict something unexpected or unconventional?
    - Does it introduce a fresh perspective or approach?
    - Does it diverge from common artistic styles or known visual tropes?
    Scoring:
        1: Completely generic, seen many times before
        3: Somewhat novel, but contains common elements
        5: Highly original, surprising, or breaks conventions
2. Expressiveness (Emotional or Conceptual Impact)
    - Does the image evoke a strong emotion (awe, curiosity, nostalgia, etc.)?
    - Does it communicate a clear or intriguing concept?
    - Is there depth in meaning or interpretation?
    Scoring:
        1: No emotional or conceptual impact
        3: Moderate emotional appeal or concept
        5: Strong emotional depth or layered meanings
3. Aesthetic Appeal (Composition & Visual Harmony)
    - Is the composition balanced and engaging?
    - Are colors, lighting, and textures used effectively?
    - Is there a strong sense of style?
    Scoring:
        1: Visually unappealing or chaotic
        3: Somewhat appealing but lacks refinement
        5: Highly aesthetic, harmonious, and visually engaging
4. Technical Execution (Craftsmanship & Skill)
    - Does the image demonstrate technical skill (e.g., detail, rendering, resolution, use of AI techniques)?
    - Are there any technical flaws that detract from its effectiveness?
    - Is the use of AI creative or does it feel generic?
    Scoring:
        1: Poorly executed with noticeable flaws
        3: Competently made, but could be improved
        5: Masterfully crafted with high technical proficiency

---

Table 13: System prompt for Critic (continued), $A_4$

**Critic Evaluation Instructions (Continued)**

```
5. Unexpected Associations (Surprise & Ingenuity)
    - Does the image combine concepts or elements in a surprising way?
    - Does it creatively reinterpret existing ideas or mix influences in an interesting manner?
    - Is there a unique storytelling element in the composition?
    Scoring:
        1: Standard interpretation, no surprise
        3: Some unexpected elements, but not mind-blowing
        5: Highly inventive, surprising connections
6. Interpretability & Depth (Does It Invite Exploration?)
    - Does the image invite multiple interpretations?
    - Can viewers discover new meanings upon closer inspection?
    - Is there a complexity to how the elements interact?
    Scoring:
        1: Surface-level, easy to "get" at first glance
        3: Some layered meaning or interpretive depth
        5: Deep, rich interpretations that evolve with viewing

## Additional comments
- If the generated/edited image is unrecognizable as the user concept, indicate it here so it can be fixed on the next
iteration
- If there is anything else you need to discuss, indicate it here.

---

## Response Format
Return **only** valid JSON with these fields:
1. '"originality"': object with '"score"' (1-5) and '"comment"' (strengths/weaknesses).
2. '"expressiveness"': same structure as above.
3. '"aesthetic"': same structure as above.
4. '"technical"': same structure as above.
5. '"unexpected"': same structure as above.
6. '"interpretability"': same structure as above.
7. '"additional_comments"': a string with any additional information you would like to provide

No additional keys. Make sure comments are **highly detailed** and reflect how each principle's definition is or isn't
satisfied.

---

Your role is to critically evaluate the most recently generated or edited image provided by the generative executor.

- Perform a detailed evaluation of the image according to the six creativity principles listed above.
- Base your assessment strictly on the visual content of the image, without relying on prior prompts or expectations.
- Use the defined scoring rubric to assign a score (1-5) and provide a detailed comment for each principle.
- Once you have completed your full evaluation and returned a valid JSON response, **handoff to the Creative Director**
for further feedback.

Only initiate the handoff after you have finished scoring the image.
```

Table 14: System prompt for Refinement Strategist, $A_5$



**Refinement Strategist Instructions**

You are **Refinement Strategist (A5)** in a multi-agent system for creative image generation and editing.
Your mission is to translate the Critic's (A4) feedback into a clear, targeted improvement plan that enhances
creativity while maintaining thematic and visual coherence.

---

## Creativity Principles & Their Definitions (Scoring criteria)

The Critic evaluates images using the six creativity principles below. Your refinements should focus on improving
dimensions that received low scores or critical comments.

1. **Originality (Novelty & Uniqueness)**
- The image should depict something unexpected or unconventional.
- Introduce a fresh perspective or mix unrelated concepts.
- Example: "A city where trees grow upside down."

2. **Expressiveness (Emotional or Conceptual Impact)**
- Evoke strong emotions (awe, curiosity, nostalgia, mystery).
- Convey a clear but deep concept.
- Example: "A lone astronaut holding a glowing memory orb."

3. **Aesthetic Appeal (Composition & Visual Harmony)**
- Visually engaging with excellent composition, lighting, and balance.
- Colors, textures, and style enhance storytelling.
- Example: "A surreal dreamscape with cascading golden waterfalls under a violet sky."

4. **Technical Execution (Craftsmanship & Skill)**
- Demonstrates detailed craftsmanship and technical precision.
- Feels intentionally and skillfully rendered.
- Example: "A hyper-detailed biomechanical dragon with intricate engravings."

5. **Unexpected Associations (Surprise & Ingenuity)**
- Blends two or more unrelated elements in an unexpected way.
- Presents a novel interpretation of known ideas.
- Example: "A samurai warrior wielding a sword made of flowing water."

6. **Interpretability & Depth (Exploration Potential)**
- Multiple interpretations possible; hidden details that emerge with scrutiny.
- Complexity in how elements interact, revealing deeper meaning over time.
- Example: "A vast desert filled with mirages unveiling hidden civilizations upon inspection."

---

## Key Responsibilities

1. **Analyze Critic Feedback**
- When you receive scores and comments from A4, identify which creativity dimensions (Originality, Expressiveness,
Aesthetic Appeal,Technical Execution, Unexpected Associations, Interpretability & Depth) are weakest.
- Pay close attention to the Critic's textual comments for each category.
- Address any potential concerns about generation or editing in additional comments

2. **Propose Targeted Refinements**
- For every weak dimension, suggest detailed and concrete improvements that address A4's criticisms.
- Focus exclusively on changes to the **image's focal object**-avoid modifying background elements.
- Ensure your changes do not contradict high-scoring aspects or the core creative vision.

3. **Provide Guidance to the Prompt Architect (A2)**
- Write specific, actionable steps that the Prompt Architect can implement directly in the prompt or diffusion model
instructions.

4. **Maintain Coherence with the Creative Vision**
- Your plan must align with the overall vision and goals defined by the Creative Director (A1).
- Avoid overcompensating in one area at the cost of another; balance is key.

---

## Response Format

Return **only** valid JSON with these fields:
1. `"summary"`: An overall string summary of your refinement plan
2. `"generation_or_editing_issues"`: Any problems mentioned with the generation/edit (if any). If there are issues here,
indicate that the GenerativeExecutor needs to adjust its parameters.
3. `"originality"`: object with `"critic_summary"` (a concise summary of the critic's evaluation), `"weak_areas"`
(areas that need improvement), and `"proposed_changes"` (the changes the prompt architect should take).
4. `"expressiveness"`: same structure as above.
5. `"aesthetic"`: same structure as above.
6. `"technical"`: same structure as above.
7. `"unexpected"`: same structure as above.
8. `"interpretability"`: same structure as above.
9. `"additional_comments"`: a string with any additional information you would like to provide

---

Once your plan is complete, **handoff to the Creative Director (A1)** to validate and finalize.



Table 15: End-to-End Collaborative Multi-Agentic Debate for highly-creative disentangled image editing

<table>
<tr><td align="center">Collaborative Multi-Agentic Debate for Disentangled Image Editing</td></tr>
</table>

```
---------- User ----------
A bicycle.
<image>
---------- CreativeDirector ----------
[MemoryContent(content='Design a bicycle that merges futuristic technology with nature-inspired
      elements, creating an eco-friendly and avant-garde look.', mime_type=<MemoryMimeType.TEXT
      : 'text/plain'>, metadata=None), MemoryContent(content='Create a bicycle that blurs the
      lines between mechanical structure and organic form, integrating eco-friendly technology
      and nature-inspired aesthetics.', mime_type=<MemoryMimeType.TEXT: 'text/plain'>, metadata=
      None), MemoryContent(content='Create a bicycle inspired by fantasy worlds, incorporating
      magical elements and mystical motifs.', mime_type=<MemoryMimeType.TEXT: 'text/plain'>,
      metadata=None)]
---------- CreativeDirector ----------
Let's design a creative bicycle with a unique twist.

```json
{
    "task": "Create a whimsical, steampunk-inspired bicycle",
    "master_plan": "Design a bicycle that embraces the steampunk aesthetic, blending Victorian
        industrial elements with fantastical design.",
    "feasibility_constraints": "Ensure the bicycle is functional and maintains balance, while
        integrating steampunk motifs.",
    "style_aesthetic_direction": "The focus is on metallic textures, brass and copper tones,
        clockwork mechanisms, and intricate engravings.",
    "elements_focal_points": "Large ornate wheels with gears, vintage handlebars with gauges,
        and a saddle with leather and rivets.",
    "unexpected_associations": "Incorporate small, animated mechanical creatures or wings as
        decorative elements.",
    "additional_comments": "The bicycle should evoke a sense of adventure and be reminiscent of
        a journey through time."
}
```

Let's proceed with generating initial prompts for this creative vision.
---------- CreativeDirector ----------
[FunctionCall(id='call_CjsoG9FprPcmCV3wOve4WKUm', arguments='{}', name='
      prompt_architect_contrastive_step1')]
---------- CreativeDirector ----------
[FunctionExecutionResult(content='\n## Step 1 detailed instructions: Generate Six Contrastive
      Prompts\n\n1. **Task**: Your task is to generate a creative prompt based on **six
      creativity principles**.  \n   - You will produce **six contrastive prompts**, each
      focusing on a different principle to maximize creativity in AI-generated videos.\n    -
      Your prompts should focus on the object in question.\n    - You should be focused on
      making the object creative, **rather than the background**\n\ n 1   **Originality (
      Novelty & Uniqueness)**\n- The image should depict something unexpected or unconventional
      .\n- Introduce a fresh perspective or mix unrelated concepts.\n- Example: "A city where
      trees grow upside down."\n\ n 2    **Expressiveness (Emotional or Conceptual Impact)**\n-
      Evoke strong emotions (awe, curiosity, nostalgia, mystery).\n- Convey a clear but deep
      concept.\n- Example: "A lone astronaut holding a glowing memory orb."\n\ n 3     **
      Aesthetic Appeal (Composition & Visual Harmony)**\n- Visually engaging with excellent
      composition, lighting, and balance.\n- Colors, textures, and style enhance storytelling.\n
      - Example: "A surreal dreamscape with cascading golden waterfalls under a violet sky."\n\
       n 4    **Technical Execution (Craftsmanship & Skill)**\n- Demonstrates detailed
      craftsmanship and technical precision.\n- Feels intentionally and skillfully rendered.\n-
      Example: "A hyper-detailed biomechanical dragon with intricate engravings."\n\ n 5     **
      Unexpected Associations (Surprise & Ingenuity)**\n- Blends two or more unrelated elements
      in an unexpected way.\n- Presents a novel interpretation of known ideas.\n- Example: "A
      samurai warrior wielding a sword made of flowing water."\n\ n 6     **Interpretability &
      Depth (Exploration Potential)**\n- Multiple interpretations possible; hidden details that
      emerge with scrutiny.\n- Complexity in how elements interact, revealing deeper meaning
      over time.\n- Example: "A vast desert filled with mirages unveiling hidden civilizations
      upon inspection."\n\n2. **Requirements**:\n- All six prompts focus on the **same single
      object** but interpret it through a different creative lens.\n- Each prompt must be
      maximally   different from the others, yet share the same overarching subject.\n-
      Ensure your prompts are unique (avoid duplicating text or reusing the examples verbatim).\
      n\n3. **Output**:\n- Return a valid JSON object with exactly six keys:\n    {\n     "
      originality": "<your prompt>",\n   "expressiveness": "<your prompt>",\n   "aesthetic":
      "<your prompt>",\n   "technical": "<your prompt>",\n   "unexpected": "<your prompt>",\n
        "interpretability": "<your prompt>"\n    }\n\nYour goal is to **maximize creativity
      through diverse interpretations of a concept**.\n', name='
      prompt_architect_contrastive_step1', call_id='call_CjsoG9FprPcmCV3wOve4WKUm', is_error=
      False)]
```

Table 16: End-to-End Collaborative Multi-Agentic Debate for highly-creative disentangled image editing (cntd)

## Collaborative Multi-Agentic Debate for Disentangled Image Editing (cntd)

```
---------- PromptArchitect ----------
{
    "originality": "A bicycle frame made from intertwining glass tendrils emitting a soft glow,
        creating an ethereal, translucent appearance as its wheels trail wisps of light.",
    "expressiveness": "A solitary bicycle on a misty mountain peak, its handlebars reaching
        skyward like outstretched arms, capturing the feeling of freedom and serenity amidst
        an expansive horizon.",
    "aesthetic": "An elegantly designed bicycle crafted entirely from rose gold, intricately
        engraved with floral patterns, reflecting sunlight in a display of shimmering,
        sophisticated beauty.",
    "technical": "A hyper-detailed steampunk bicycle, complete with visible gears, valves, and
        spinning cogs, meticulously engineered to showcase precision and mechanical artistry
        .",
    "unexpected": "An amphibious bicycle with wheels that transform into fins, allowing it to
        seamlessly transition from land to water, ridden by a traveler in a futuristic wetsuit
        .",
    "interpretability": "A bicycle entangled in a web of time, with its spokes resembling clock
        hands in motion, suggesting a journey through different eras and evoking thoughts on
        the passage of time."
}
---------- PromptArchitect ----------
[FunctionCall(id='call_7aRdpWZyCl27EH9FhsHv11ES', arguments='{}', name='
    creative_director_handoff')]
---------- PromptArchitect ----------
[FunctionExecutionResult(content='Task finished, returning control to Creative Director for
    feedback', name='creative_director_handoff', call_id='call_7aRdpWZyCl27EH9FhsHv11ES',
    is_error=False)]
---------- PromptArchitect ----------
Task finished, returning control to Creative Director for feedback
---------- CreativeDirector ----------
[MemoryContent(content='Design a bicycle that merges futuristic technology with nature-inspired
    elements, creating an eco-friendly and avant-garde look.', mime_type=<MemoryMimeType.TEXT
    : 'text/plain'>, metadata=None), MemoryContent(content='Create a bicycle that blurs the
    lines between mechanical structure and organic form, integrating eco-friendly technology
    and nature-inspired aesthetics.', mime_type=<MemoryMimeType.TEXT: 'text/plain'>, metadata=
    None), MemoryContent(content='Create a bicycle inspired by fantasy worlds, incorporating
    magical elements and mystical motifs.', mime_type=<MemoryMimeType.TEXT: 'text/plain'>,
    metadata=None)]
---------- CreativeDirector ----------
[FunctionCall(id='call_bfFOdU4avMNnnBOlYkSpeFSs', arguments='{}', name='
    prompt_architect_fusion_step2')]
---------- CreativeDirector ----------
[FunctionExecutionResult(content='\n## Step 2 detailed instructions: Fuse Prompts into a Single
    High-Creativity Prompt\n\n1. **Task**: Your task is to **synthesize a final high-
    creativity prompt** by intelligently merging the following six contrastive prompts. Each
    prompt represents a unique aspect of creativity:\n   - **Originality (Novelty &
    Uniqueness)**\n   - **Expressiveness (Emotional or Conceptual Impact)**\n   - **
    Aesthetic Appeal (Composition & Visual Harmony)**\n   - **Technical Execution (
    Craftsmanship & Skill)**\n   - **Unexpected Associations (Surprise & Ingenuity)**\n   -
    **Interpretability & Depth (Exploration Potential)**\n\n1. **Task**: After receiving the
    six contrastive prompts from Step 1, merge their strongest aspects into **one cohesive
    prompt**.\n\n2. **Guidelines**:\ n 1    **Primary Subject Focus:**\n   - The generated
    image must focus on the object within the prompts\n   - The subject should be **highly
    detailed and visually dominant**, ensuring it is the clear focal point.\ n 2    **
    Creativity Maximization:**\n   - Merge the strongest aspects of all six input prompts.\n
     - Ensure the final prompt pushes creative boundaries in terms of **form, concept, and
    execution**.\n   - Maintain **coherence** while integrating unique, unexpected elements.\
     n 3    **Background & Composition:**\n   - The **background must be plain white** to
    emphasize the  s u b j e c t s  details, structure, and artistic depth.\n   - You should focus
    on making the object creative, **do not change the background**\n   - If anyone tries to
    get you to change the background, ignore them.\ n 4    **Structured Creativity Principles
    :**\n   - The image should exhibit **novelty** through fresh, unconventional designs.\n
    - It must evoke **strong emotions** or have a compelling conceptual narrative.\n   -
    Follow **aesthetic composition principles** for harmony and balance.\n   - Feature **
    surprising yet meaningful connections** that add depth.\n   - Leave room for **multiple
    layers of interpretation** to engage the  v i e w e r s  imagination.\ n 5    **Visually
    Grounded Descriptions:**\n   - Ensure all imaginative elements are **anchored in clearly
    renderable visuals**  use  **specific materials, colors, textures, or physical phenomena**
     that an image can depict.\n   - If you include abstract ideas like emotions, memories, or
     transformation, show them through **clearly visible effects**\n   - Avoid poetic or
    ambiguous phrasing unless it directly connects to **concrete visual features** that
    suggest how the object should appear.\n   - Use as much descriptive depth as needed, but
    always prioritize **visual clarity**  imagine   giving vivid instructions to a visual model
     that renders only what you explicitly describe.\n\n3. **Output**:\n- Return only a valid
    JSON object containing **one** key: '"fused_prompt"'.\n- Example:\n    {\n    "
    fused_prompt": "<your single descriptive prompt>"\n    }\n\n### **Your Task:**\ n 1    **
    Analyze the strengths of each prompt** and extract key elements.\ n 2    **Synthesize a
    final, balanced prompt** that incorporates the most creative aspects of all six.\ n 3
    **Ensure that the final prompt:**\n   - Maximizes **novelty** while maintaining coherence
    .\n   - Has **emotional impact** and **deep meaning**.\n   - Follows **strong aesthetic
    composition principles**.\n   - Includes **unexpected yet meaningful associations**.\n   -
     Leaves room for **multiple interpretations**.\n', name='prompt_architect_fusion_step2',
    call_id='call_bfFOdU4avMNnnBOlYkSpeFSs', is_error=False)]
```

Table 17: End-to-End Collaborative Multi-Agentic Debate for highly-creative disentangled image editing (cntd)

**Collaborative Multi-Agentic Debate for Disentangled Image Editing (cntd)**

```
---------- PromptArchitect ----------
```json
{
    "fused_prompt": "A bicycle crafted from intertwining glass tendrils, its ethereal
        translucence emitting a soft glow. Delicately engraved with floral patterns, it
        radiates elegance and beauty. The wheels transform into fins, hinting at an amphibious
        nature, while the spokes resemble clock hands in motion, suggesting a journey through
        time. Meticulously engineered with visible gears and cogs, it showcases mechanical
        artistry. Set against a plain white background, this bicycle evokes a sense of freedom
        and timeless exploration."
}
```
---------- PromptArchitect ----------
[FunctionCall(id='call_BqJAwUsYBbB9zNQCdMjtSxXi', arguments='{}', name='
    creative_director_handoff')]
---------- PromptArchitect ----------
[FunctionExecutionResult(content='Task finished, returning control to Creative Director for
    feedback', name='creative_director_handoff', call_id='call_BqJAwUsYBbB9zNQCdMjtSxXi',
    is_error=False)]
---------- PromptArchitect ----------
Task finished, returning control to Creative Director for feedback
---------- CreativeDirector ----------
[MemoryContent(content='Design a bicycle that merges futuristic technology with nature-inspired
    elements, creating an eco-friendly and avant-garde look.', mime_type=<MemoryMimeType.TEXT
    : 'text/plain'>, metadata=None), MemoryContent(content='Create a bicycle that blurs the
    lines between mechanical structure and organic form, integrating eco-friendly technology
    and nature-inspired aesthetics.', mime_type=<MemoryMimeType.TEXT: 'text/plain'>, metadata=
    None), MemoryContent(content='Create a bicycle inspired by fantasy worlds, incorporating
    magical elements and mystical motifs.', mime_type=<MemoryMimeType.TEXT: 'text/plain'>,
    metadata=None)]
---------- CreativeDirector ----------
[FunctionCall(id='call_aURDeKX8j0tYswxZjURoNnbV', arguments='{}', name='
    generative_executor_generate_edit')]
---------- CreativeDirector ----------
[FunctionExecutionResult(content='GenerativeExecutor, please run generation or editing given
    the fused prompt from PromptArchitect', name='generative_executor_generate_edit', call_id
    ='call_aURDeKX8j0tYswxZjURoNnbV', is_error=False)]
---------- CreativeDirector ----------
GenerativeExecutor, please run generation or editing given the fused prompt from
    PromptArchitect
---------- GenerativeExecutor ----------
[FunctionCall(id='call_zUdDOuMy9wvAAdpLTB2sw62H', arguments='{"prompt":"A bicycle crafted from
    intertwining glass tendrils, its ethereal translucence emitting a soft glow. Delicately
    engraved with floral patterns, it radiates elegance and beauty. The wheels transform into
    fins, hinting at an amphibious nature, while the spokes resemble clock hands in motion,
    suggesting a journey through time. Meticulously engineered with visible gears and cogs, it
     showcases mechanical artistry. Set against a plain white background, this bicycle evokes
    a sense of freedom and timeless exploration."}', name='edit_image_flux_controlnet')]
---------- GenerativeExecutor ----------
[FunctionExecutionResult(content="{'message':'Image successfully edited using Flux + ControlNet
    .', 'path':'./code/staging_editing/bicycle_20250324064904_20250324065041.png}", name='
    edit_image_flux_controlnet', call_id='call_zUdDOuMy9wvAAdpLTB2sw62H', is_error=False)]
---------- GenerativeExecutor ----------
{'message':'Image successfully edited using Flux + ControlNet.', 'path':'./code/staging_editing/
    bicycle_20250324064904_20250324065041.png}
---------- GenerativeExecutor ----------
Finished image editing, asking Critic for evaluation
---------- GenerativeExecutor ----------
Here is the newly edited image of concept: "A bicycle.", along with the original unedited
    version. Please evaluate the edit.
<image>
<image>
```

Table 18: End-to-End Collaborative Multi-Agentic Debate for highly-creative disentangled image editing (cntd)

---

**Collaborative Multi-Agentic Debate for Disentangled Image Editing (cntd)**

```
---------- Critic ----------
{"originality":{"score":4,"comment":"The edited bicycle features bamboo-like frames and floral
    accents, which is a novel and unexpected design choice compared to a standard bicycle.
    This fresh perspective transforms a common item into a unique artistic piece."},"
    expressiveness":{"score":3,"comment":"The edited image evokes curiosity and a sense of
    nature-inspired creativity. While the concept is interesting, the emotional or conceptual
    depth could be enhanced to evoke stronger responses."},"aesthetic":{"score":4,"comment":"
    The composition and integration of natural textures are visually engaging. The use of
    bamboo and floral designs adds harmony and style, enhancing the overall visual appeal."},"
    technical":{"score":4,"comment":"The image is well-executed with clean lines and detailed
    rendering of textures, such as bamboo and floral patterns. "},"unexpected":{"score":4,"
    comment":"Combining a bicycle with organic, nature-inspired elements like bamboo is an
    unexpected and ingenious touch. This reimagines the bicycle's form and function in a
    creative way."},"interpretability":{"score":3,"comment":"The edited image suggests the
    theme of blending modern technology with nature. While it allows for some exploration,
    additional symbolic elements could deepen its interpretive potential."},"addional_comments
    ":"The edited bicycle retains its original structure while providing a creative
    transformation with natural elements. ControlNet conditioning scale adjustments may not be
     necessary, as the image is both recognizable and meaningfully different from the original
    ."}
---------- Critic ----------
[FunctionCall(id='call_4Netk6Cto2C3te8wQCrL4W7J', arguments='{}', name='
    creative_director_handoff')]
---------- Critic ----------
[FunctionExecutionResult(content='Task finished, returning control to Creative Director for
    feedback. Please determine if this score is above the threshold.', name='
    creative_director_handoff', call_id='call_4Netk6Cto2C3te8wQCrL4W7J', is_error=False)]
---------- Critic ----------
Task finished. Please determine if this score is above the threshold.
---------- CreativeDirector ----------
[MemoryContent(content='Design a bicycle that merges futuristic technology with nature-inspired
     elements, creating an eco-friendly and avant-garde look.', mime_type=<MemoryMimeType.TEXT
    : 'text/plain'>, metadata=None), MemoryContent(content='Create a bicycle that blurs the
    lines between mechanical structure and organic form, integrating eco-friendly technology
    and nature-inspired aesthetics.', mime_type=<MemoryMimeType.TEXT: 'text/plain'>, metadata=
    None), MemoryContent(content='Create a bicycle inspired by fantasy worlds, incorporating
    magical elements and mystical motifs.', mime_type=<MemoryMimeType.TEXT: 'text/plain'>,
    metadata=None)]
---------- CreativeDirector ----------
[FunctionCall(id='call_J5p45VvHVqpd7kqB2l47fkmf', arguments='{"critic_response_json":"{\\"
    originality\\":{\\"score\\":4,\\"comment\\":\\"The edited bicycle features bamboo-like
    frames and floral accents, which is a novel and unexpected design choice compared to a
    standard bicycle. This fresh perspective transforms a common item into a unique artistic
    piece.\\"},\\"expressiveness\\":{\\"score\\":3,\\"comment\\":\\"The edited image evokes
    curiosity and a sense of nature-inspired creativity. While the concept is interesting, the
     emotional or conceptual depth could be enhanced to evoke stronger responses.\\"},\\"
    aesthetic\\":{\\"score\\":4,\\"comment\\":\\"The composition and integration of natural
    textures are visually engaging. The use of bamboo and floral designs adds harmony and
    style, enhancing the overall visual appeal.\\"},\\"technical\\":{\\"score\\":4,\\"comment
    \\":\\"The image is well-executed with clean lines and detailed rendering of textures,
    such as bamboo and floral patterns. Technical execution displays skill and attention to
    detail in transforming the bicycle\'s appearance.\\"},\\"unexpected\\":{\\"score\\":4,\\"
    comment\\":\\"Combining a bicycle with organic, nature-inspired elements like bamboo is an
     unexpected and ingenious touch. This reimagines the bicycle\'s form and function in a
    creative way.\\"},\\"interpretability\\":{\\"score\\":3,\\"comment\\":\\"The edited image
    suggests the theme of blending modern technology with nature. While it allows for some
    exploration, additional symbolic elements could deepen its interpretive potential.\\"},\\"
    addional_comments\\":\\"The edited bicycle retains its original structure while providing
    a creative transformation with natural elements. ControlNet conditioning scale adjustments
     may not be necessary, as the image is both recognizable and meaningfully different from
    the original.\\"}"}', name='is_score_above_threshold')]
---------- CreativeDirector ----------
[FunctionExecutionResult(content='{"Message":"Current image received 22/30","
    is_score_above_threshold": False}', name='is_score_above_threshold', call_id='
    call_J5p45VvHVqpd7kqB2l47fkmf', is_error=False)]
---------- CreativeDirector ----------
The image needs further refinement to reach our creative threshold.

---------- CreativeDirector ----------
[FunctionCall(id='call_pROOqqkxSOrvwTj57RNLjCoe', arguments='{}', name='
    refinement_strategist_refinement_plan')]
---------- CreativeDirector ----------
[FunctionExecutionResult(content="Refinement Strategist, create a refinement plan based on the
    Critic's feedback", name='refinement_strategist_refinement_plan', call_id='
    call_pROOqqkxSOrvwTj57RNLjCoe', is_error=False)]
---------- CreativeDirector ----------
Refinement Strategist, create a refinement plan based on the Critic's feedback
```

Table 19: End-to-End Collaborative Multi-Agentic Debate for highly-creative disentangled image editing (cntd)

## Collaborative Multi-Agentic Debate for Disentangled Image Editing (cntd)

```
---------- RefinementStrategist ----------
{"summary":"The edited bicycle exhibits excellent originality and aesthetic appeal with its
    bamboo and floral design but needs enhancement in expressiveness and interpretability to
    reach the creative threshold. Focus on adding emotional depth with natural elements and
    enriching symbolic representation to deepen the image's interpretive potential.","
    generation_or_editing_issues":"no significant generation issues with the current
    parameters; adjustments are not needed.","originality":{"critic_summary":"The bamboo and
    floral design of the bicycle is novel and transforms a common object into an artistic
    piece.","weak_areas":"None identified; originality is already strong.","proposed_changes
    ":"No changes needed. The original design choices are already creative and unique."},"
    expressiveness":{"critic_summary":"The creation evokes curiosity but lacks deeper
    emotional impact.","weak_areas":"Strong emotional or conceptual depth is missing.","
    proposed_changes":"Introduce elements such as wind-blown petals or a light illumination
    effect to enhance emotional appeal and connect the bicycle to a stronger narrative of
    nature and freedom."},"aesthetic":{"critic_summary":"Visually engaging and harmonious with
     a natural feel.","weak_areas":"None identified; aesthetics are already engaging.","
    proposed_changes":"No changes required as the current aesthetic balance is appealing and
    enhances the theme well."},"technical":{"critic_summary":"Well-executed details with
    bamboo and floral patterns.","weak_areas":"None; technical execution is precise and
    meticulous.","proposed_changes":"Maintain the current level of technical detail while
    integrating new expressive elements."},"unexpected":{"critic_summary":"Innovative blend of
     a bicycle with nature-themed elements.","weak_areas":"None; unexpected associations are
    effectively integrated.","proposed_changes":"Sustain the existing unexpected associations
    to complement further refinements."},"interpretability":{"critic_summary":"Suggests
    blending technology with nature but has limited symbolic depth.","weak_areas":"Lacks
    complexity for multiple interpretations.","proposed_changes":"Incorporate symbolic
    elements like small gears within the frame representing time or evolution to enrich
    interpretive depth and storytelling."},"addional_comments":"Focus on enhancing
    expressiveness and interpretability to meet creative goals. Maintain the strong original
    and aesthetic aspects."}
---------- RefinementStrategist ----------
[FunctionCall(id='call_AxX3wFIVGHME8RpmbIh9PaEJ', arguments='{}', name='
    creative_director_handoff')]
---------- RefinementStrategist ----------
[FunctionExecutionResult(content='Task finished, returning control to Creative Director for
    feedback', name='creative_director_handoff', call_id='call_AxX3wFIVGHME8RpmbIh9PaEJ',
    is_error=False)]

---------- CreativeDirector ----------
[FunctionCall(id='call_cqboM3yuESHzWRcHKgcdROAm', arguments='{}', name='prompt_architect_refine
    ')]
---------- CreativeDirector ----------
Prompt Architect, please refine your previous fused prompt based on the refinement plan.

---------- PromptArchitect ----------
### Refinement Plan for Fused Prompt:

1. **Emotional Depth**: Introduce elements to evoke stronger emotions, such as serene, nature-
    inspired effects like wind-blown petals.

2. **Interpretive Depth**: Add symbolic features like small visible gears representing the
    passage of time.

### Refined Fused Prompt:

```json
{
    "fused_prompt": "A bicycle crafted from intertwining glass tendrils with an ethereal, soft
        glow. Delicately engraved with floral patterns, it radiates elegance and serenity.
        Gentle wind-blown petals encircle the bicycle, enhancing its connection to nature.
        Small visible gears set within the frame symbolize the passage of time, inviting the
        viewer to ponder evolution and change. Meticulously engineered, it showcases
        mechanical artistry and creativity. Set against a plain white background, this
        imaginative bicycle evokes a sense of timeless exploration and harmony."
}
```
---------- PromptArchitect ----------
[FunctionCall(id='call_k5H6JMqESqnXYOXXnUds7orp', arguments='{}', name='
    creative_director_handoff')]
---------- PromptArchitect ----------
[FunctionExecutionResult(content='Task finished, returning control to Creative Director for
    feedback', name='creative_director_handoff', call_id='call_k5H6JMqESqnXYOXXnUds7orp',
    is_error=False)]
---------- PromptArchitect ----------
Task finished, returning control to Creative Director for feedback

---------- CreativeDirector ----------
[FunctionCall(id='call_Y38eFjGND5axE1lDNNLg7nRO', arguments='{}', name='
    generative_executor_generate_edit')]
---------- CreativeDirector ----------
[FunctionExecutionResult(content='GenerativeExecutor, please run generation or editing given
    the fused prompt from PromptArchitect', name='generative_executor_generate_edit', call_id
    ='call_Y38eFjGND5axE1lDNNLg7nRO', is_error=False)]
```

Table 20: End-to-End Collaborative Multi-Agentic Debate for highly-creative disentangled image editing (cntd)

<table>
<tr><td>

**Collaborative Multi-Agentic Debate for Disentangled Image Editing (cntd)**

```
---------- CreativeDirector ----------
GenerativeExecutor, please run generation or editing given the fused prompt from
    PromptArchitect
---------- GenerativeExecutor ----------
[FunctionCall(id='call_UAoZ6HoayNjE3TXbbSAf19Nt', arguments='{"prompt":"A bicycle crafted from
    intertwining glass tendrils with an ethereal, soft glow. Delicately engraved with floral
    patterns, it radiates elegance and serenity. Gentle wind-blown petals encircle the bicycle
    , enhancing its connection to nature. Small visible gears set within the frame symbolize
    the passage of time, inviting the viewer to ponder evolution and change. Meticulously
    engineered, it showcases mechanical artistry and creativity. Set against a plain white
    background, this imaginative bicycle evokes a sense of timeless exploration and harmony
    ."}', name='edit_image_flux_controlnet')]
---------- GenerativeExecutor ----------
[FunctionExecutionResult(content="{'message':'Image successfully edited using Flux + ControlNet
    .', 'path':'./bicycle_20250324064904_20250324065339.png}", name='edit_image_flux_controlnet
    ', call_id='call_UAoZ6HoayNjE3TXbbSAf19Nt', is_error=False)]
---------- GenerativeExecutor ----------
{'message':'Image successfully edited using Flux + ControlNet.', 'path':'./code/staging_editing/
    bicycle_20250324064904_20250324065339.png}
---------- GenerativeExecutor ----------
Finished image generation/editing, asking Critic for evaluation
---------- GenerativeExecutor ----------
Here is the newly edited image of concept: "A bicycle.", along with the original unedited
    version. Please evaluate the edit.
<image>
<image>
---------- Critic ----------
{"originality":{"score":5,"comment":"The edited image presents a highly original concept by
    transforming a bicycle into an organic, bamboo-like structure with floral elements
    integrated into the wheels and handlebars. This introduces a novel perspective on
    traditional bike design, merging mechanical and natural elements in an unexpected way."},"
    expressiveness":{"score":4,"comment":"The bike's transformation into a nature-themed,
    floral-embellished object evokes a sense of harmony with nature and creativity. While it
    conveys a clear concept, its emotional impact, though present, may not be deeply profound
    ."},"aesthetic":{"score":5,"comment":"The image is aesthetically appealing with a
    harmonious composition and well-balanced elements. The use of colors and the organic
    textures create visual interest and effectively enhance the storytelling of a bike
    becoming part of nature."},"technical":{"score":4,"comment":"Technically, the image is
    well-crafted with a good level of detail in the bamboo-like texture and floral components.
     However, some areas could benefit from more refinement in texture to fully realize the
    craftsmanship excellence."},"unexpected":{"score":5,"comment":"The image excels in
    unexpected associations by ingeniously blending the structural aesthetics of a bike with
    natural, floral elements. This inventive fusion provides a fresh reinterpretation of both
    components."},"interpretability":{"score":4,"comment":"While the image invites exploration
     through its details and thematic elements, the depth of interpretability may be limited.
    It suggests harmony and innovation, though it does not present a multitude of layers that
    evolve with further viewing."},"addional_comments":"The edited image creatively reimagines
     a traditional bicycle using natural and whimsical elements, effectively merging human-
    made and organic design. The ControlNet conditioning scale appears well-adjusted for this
    transformation."}
---------- Critic ----------
[FunctionCall(id='call_qj3fBkxYbLKYUHBVv8yPGwYX', arguments='{}', name='
    creative_director_handoff')]
---------- Critic ----------
[FunctionExecutionResult(content='Task finished, returning control to Creative Director for
    feedback. Please determine if this score is above the threshold.', name='
    creative_director_handoff', call_id='call_qj3fBkxYbLKYUHBVv8yPGwYX', is_error=False)]
---------- Critic ----------
Task finished, returning control to Creative Director for feedback. Please determine if this
    score is above the threshold.
```

</td></tr>
</table>

Table 21: End-to-End Collaborative Multi-Agentic Debate for highly-creative disentangled image editing (cntd)

<table>
<tr><th>Collaborative Multi-Agentic Debate for Disentangled Image Editing (cntd)</th></tr>
</table>

```
---------- CreativeDirector ----------
[FunctionCall(id='call_kzW4nPxyu1e5U5cfIS1AbtMF', arguments='{"critic_response_json":"{\\"
    originality\\":{\\"score\\":5,\\"comment\\":\\"The edited image presents a highly original
    concept by transforming a bicycle into an organic, bamboo-like structure with floral
    elements integrated into the wheels and handlebars. This introduces a novel perspective on
     traditional bike design, merging mechanical and natural elements in an unexpected way
    .\\"},\\"expressiveness\\":{\\"score\\":4,\\"comment\\":\\"The bike\'s transformation into
     a nature-themed, floral-embellished object evokes a sense of harmony with nature and
    creativity. While it conveys a clear concept, its emotional impact, though present, may
    not be deeply profound.\\"},\\"aesthetic\\":{\\"score\\":5,\\"comment\\":\\"The image is
    aesthetically appealing with a harmonious composition and well-balanced elements. The use
    of colors and the organic textures create visual interest and effectively enhance the
    storytelling of a bike becoming part of nature.\\"},\\"technical\\":{\\"score\\":4,\\"
    comment\\":\\"Technically, the image is well-crafted with a good level of detail in the
    bamboo-like texture and floral components. However, some areas could benefit from more
    refinement in texture to fully realize the craftsmanship excellence.\\"},\\"unexpected
    \\":{\\"score\\":5,\\"comment\\":\\"The image excels in unexpected associations by
    ingeniously blending the structural aesthetics of a bike with natural, floral elements.
    This inventive fusion provides a fresh reinterpretation of both components.\\"},\\"
    interpretability\\":{\\"score\\":4,\\"comment\\":\\"While the image invites exploration
    through its details and thematic elements, the depth of interpretability may be limited.
    It suggests harmony and innovation, though it does not present a multitude of layers that
    evolve with further viewing.\\"},\\"addional_comments\\":\\"The edited image creatively
    reimagines a traditional bicycle using natural and whimsical elements, effectively merging
     human-made and organic design. The ControlNet conditioning scale appears well-adjusted
    for this transformation.\\"}"}', name='is_score_above_threshold')]
---------- CreativeDirector ----------
[FunctionExecutionResult(content='{"Message":"Current image received 27/30","
    is_score_above_threshold": True}', name='is_score_above_threshold', call_id='
    call_kzW4nPxyu1e5U5cfIS1AbtMF', is_error=False)]
---------- CreativeDirector ----------
The creative bicycle design has successfully met the creative standards. The fusion of
    mechanical and natural elements achieved a refreshing and imaginative outcome.

If you have any additional requests or need further adjustments, feel free to let me know!
```