# OpenReview forum: "CREA: A Collaborative Multi-Agent Framework for Creative Image Editing and Generation"
_NeurIPS.cc/2025/Conference — NeurIPS 2025 poster_

### Official Review · Reviewer_FrHo · 2025-06-30

**Clarity:** 2
**Significance:** 3
**Originality:** 3
**Rating:** 4
**Confidence:** 4

**Summary:**

This paper proposes a multi-agent collaborative framework for creative image generation and editing. The pipeline consists of several agents. First, the Creative Director interprets the original image to create a conceptual blueprint. Second, the Prompt Agent generates a set of contrastive prompts that reflect the creativity principles based on the blueprint and merges them into a single prompt. Third, the Generative Executor sets the model parameters and uses a text-to-image (T2I) model to generate or edit the image creatively. Finally, the Critic evaluates the generated image using a multimodal large language model (MLLM), and the Creative Director reviews the judgment to decide whether further edits are needed. If so, the Prompt Architect updates the prompt for the next iteration. Compared to state-of-the-art generation and editing methods such as SDedit, InstructPix2Pix, SDXL, and FLUX, the proposed approach demonstrates improved creativity and diversity in the generated images. The authors also present strong results in personalization and video generation.

**Questions:**

There are missing details that make the creative process hard to follow. For example, what exactly is a contrastive prompt? Although the paper references creativity theory, it is unclear how the six principles are actually used by the agent to refine outputs. How does the Prompt Architect synthesize contrastive prompts based on distinct creativity principles?

**Ethical Concerns:**

["NO or VERY MINOR ethics concerns only"]

**Final Justification:**

Despite some areas for improvement noted above, I still lean toward a positive evaluation of the paper, as it proposes a novel agentic framework for the new task of creativity.
- Reviewer LYr2 pointed out that the phrase “creative [object Object]” is ambiguous and may not effectively guide generation. I share this concern. I believe the authors could more clearly justify the importance of “creativity” in this context and provide a clearer definition or evaluation that aligns with human intent.
- Reviewer S95b mentioned that the internal logic of some agents (e.g., the critique agent) is described at a high level. I agree that the writing could be improved by incorporating more concrete implementation details.

**Limitations:**

Yes

**Quality:**

3

**Strengths And Weaknesses:**

Weaknesses

1. In the evaluation, the authors mainly use the prompt “a creative object,” which is not a very explicit instruction for the model. Since there are many ways an object can be creative, such a prompt is vague, hard to follow, and cannot clearly fulfill the user’s intent—especially compared to more concrete descriptions like specific materials or styles. The paper does not explain how the agent introduces more explicit constraints in the generation process. In addition to being “creative,” the authors should incorporate more diverse prompts that clearly specify how they want the object to be creative—e.g., through specific styles, materials, or design principles.

2. The paper could be strengthened if the authors provided a detailed example of how the agent interacts across a full round of the editing pipeline, including the contrastive prompts and the generated images. The broader impact and limitations section could be moved to the appendix to make space.

3. The evaluation metric is a bit questionable. The method favors both a high LPIPS score (which indicates more dissimilar images) and a high DINO score (which indicates more similar images). Wouldn’t these two metrics contradict each other? Also, why are the CLIP and DINO metrics missing in Table 2?

4. The method has limited applicability. The creative editing results do not allow for explicit structural changes, and the model is focused on generating a single object rather than handling more complex scenes.

Minor issue:
The references are broken after [45].

Strength

1. Multi-agent frameworks are a popular research topic, but there are few attempts to apply them to image or video generation. This work is among the first to propose a novel multi-agent framework for improving creativity in image generation.

2. The proposed method is clean and straightforward, supported by theoretical principles of creativity.

3. The experimental results demonstrate strong performance compared to baseline models. The authors also conduct comprehensive experiments, covering multiple applications and including a detailed ablation study in Section 4.3.

4. In addition to image generation and editing, the authors also present a proof-of-concept experiment in video generation, demonstrating the generalizability of the proposed system.

---

> ### Author Rebuttal · Authors · 2025-07-31
>
> We thank the reviewer for their kind and positive comments. We address the reviewer's concerns in detail below:
>
> **Q1: The evaluation relies heavily on the vague prompt “a creative object,” which lacks clear guidance and makes fulfilling user intent difficult. The paper doesn’t clarify how agents add explicit constraints, and should include more diverse, detailed prompts specifying styles, materials, or design principles to better direct creativity.**
>
> We agree that the base prompt “a creative object” is intentionally underspecified. This was a deliberate design choice to evaluate CREA’s ability to autonomously expand vague or minimal input into rich, interpretable, and multi-dimensional outputs using its internal creative blueprint, B, and multi-agent reasoning loop. CREA’s central contribution is precisely its capacity to introduce explicit constraints and stylistic signals automatically through prompt refinement, grounded in both system priors and feedback from the Critic agent. As detailed in Section 3.3 and Supplementary material, the Prompt Architect incorporates dimension-specific refinement goals (e.g., “increase structural novelty” or “emphasize surreal forms”) based on which creativity principles are underperforming. This mechanism enables CREA to infuse prompts with targeted constraints across material, structure, function, or aesthetic, often leading to generations that include hybrid materials, symbolic compositions, or biomorphic deformations. Moreover, CREA supports user-in-the-loop control (Section 3.4), where users can specify preferred styles, constraints, or materials that are propagated throughout the agentic pipeline. While “a creative object” serves as a common minimal anchor in evaluation, the framework itself is not limited to vague prompts and in fact excels at refining or grounding them. We will clarify this design choice and incorporate additional examples involving explicit stylistic and material constraints in the revision.
>
>
>
> **Q2: The paper could be strengthened if the authors provided a detailed example of how the agent interacts across a full round of the editing pipeline, including the contrastive prompts and the generated images. The broader impact and limitations section could be moved to the appendix to make space.**
>
> We agree and appreciate the suggestion. We kindly point out that a sample interaction between our agents are shown in Figures 5–8 (in Appendix), each showing a complete round of the editing pipeline, from the initial base concept, through the contrastive creativity prompts generated by the Prompt Architect, to the feedback from the Critic, the refinement goals proposed by the Strategist, and the final revised generation. These traces demonstrate how each agent contributes, how contrastive reasoning is applied, and how feedback drives iterative improvements. We also point out that the interaction logic is formalized in Algorithm 1 (Appendix).
>
> Additionally, we note that we provided the source code in the supplementary material, and `code/crea/agents` folder has the full instructions used by our agents for transparency.  We will explicitly reference detailed versions of these examples in the main paper.
>
> **Q3: There are missing details that make the creative process hard to follow. For example, what exactly is a contrastive prompt? It is unclear how the six principles are actually used by the agent to refine outputs. How does the Prompt Architect synthesize contrastive prompts based on distinct creativity principles?**
>
> We appreciate this important observation and clarify that contrastive prompts refer to structured prompt pairs; one representing a base or failed concept, and the other an improved variant targeting a specific creativity dimension (e.g., “a mug on a table” vs. “a biomorphic mug with swirling iridescent textures”). Six contrastive prompts are generated, one for each of the six creativity principles-such as Originality, Surprise element, as given in the predefined editable creativity template to highlight all of them accurately. These contrastive pairs enable the Prompt Architect to reason over differences and generate refined prompts that inject novelty while preserving core semantics. The six creativity principles serve as diagnostic signals: the Critic agent scores each image along these axes, and the Refinement Strategist identifies which dimensions scored below threshold. These are translated into natural language refinement goals (e.g., “increase originality through unexpected associations”), which the Prompt Architect incorporates using chain-of-thought-style fusion. This process is fully demonstrated in the multi-turn trace of Figures 5-8 of the supplementary section. We will clarify the contrastive prompting and principled refinement steps more explicitly in the revision.
>
> **Q4: The evaluation metric is a bit questionable. The method favors both a high LPIPS score (which indicates more dissimilar images) and a high DINO score (which indicates more similar images). Wouldn’t these two metrics contradict each other? Also, why are the CLIP and DINO metrics missing in Table 2?**
>
> To clarify, LPIPS and DINO metrics serve distinct, complementary purposes in our evaluation:
>
> * LPIPS is used to quantify perceptual diversity by computing pairwise distances between generated images within the same object group. In the context of creative generation, higher LPIPS indicates greater visual diversity across outputs, a desirable property for tasks emphasizing creativity and expressive variation.
> * DINO, in contrast, assesses how well key structural and semantic characteristics of the original image are retained after editing. A higher DINO score thus implies better preservation of essential semantic details, even amid substantial visual transformations.
>
> Thus, rather than being contradictory, high scores in both LPIPS and DINO metrics together indicate successful disentangled editing, producing edits that are both meaningfully creative and semantically coherent.
>
> **Q5: The method has limited applicability. The creative editing results do not allow for explicit structural changes, and the model is focused on generating a single object rather than handling more complex scenes.**
>
> Thank you for raising this concern. CREA is designed to be a tool-agnostic and flexible framework that supports explicit structural edits and is not limited to single-object generation.
>
> CREA enables structural changes through its Generative Executor, which picks the appropriate tool from an extensible set of tools of the user’s choice to perform the task at hand (editing, generation, style transformation etc.), while also providing the tool and task-appropriate hyperparameters to control editing. This allows CREA to perform various types of edits at different granularities, depending on the chosen editing tool.
>
> Moreover, CREA is not inherently limited to single-object outputs. While the paper focuses on single-object examples to clearly showcase disentanglement and ensure fair, interpretable evaluation across the six creativity dimensions, the framework itself is content-agnostic. The creativity blueprint generated by the Director agent supports compositional planning, and the editing pipeline can seamlessly extend to multi-object or scene-level editing. This is further demonstrated in the video extension, where agents generate coherent multi-entity scenes based on structured prompts covering subject, action, setting, and style. Due to NeurIPS guidelines,  we are not allowed to provide example images here, we will ensure to include multi-object scenes in the camera-ready version.

---

> ### Author Response · Authors · 2025-08-06
>
> Dear reviewer,
>
> We appreciate your valuable review, and we addressed all your concerns in our rebuttal. As the NeurIPS discussion phase will conclude soon, we kindly remind you that if you have any further questions or require additional clarifications on our paper, we are more than happy to address them promptly.
>
> Best,
>
> Authors of CREA

---

> > ### Comment · Reviewer_FrHo · 2025-08-07
> >
> > I have read the authors’ response and the comments from other reviewers, and some of their concerns align with mine.
> >
> > - Reviewer LYr2 pointed out that the phrase “creative [object Object]” is ambiguous and may not effectively guide generation. I share this concern. I believe the authors could more clearly justify the importance of “creativity” in this context and provide a clearer definition or evaluation that aligns with human intent.
> >
> > - Reviewer S95b mentioned that the internal logic of some agents (e.g., the critique agent) is described at a high level. I agree that the writing could be improved by incorporating more concrete implementation details.
> >
> > - Reviewer 2FiS raised the question of whether such an elaborate multi-agent architecture is necessary, given that a powerful large language model might achieve comparable results. This concern appears to have been addressed in the rebuttal.

---

> > > ### Author Response · Authors · 2025-08-07
> > >
> > > We sincerely thank the reviewer for their thoughtful feedback on our rebuttal. We're pleased to hear that our justification for the necessity of a multi-agent architecture was found adequate. Below, we provide additional clarification to address the remaining concerns.
> > >
> > > ***Reviewer LYr2 pointed out that the phrase “creative [object Object]” is ambiguous and may not effectively guide generation. I share this concern. I believe the authors could more clearly justify the importance of “creativity” in this context and provide a clearer definition or evaluation that aligns with human intent.***
> > >
> > > We would like to clarify that this particular prompting format (`creative < object >`) is not directly utilized by our proposed framework to guide generation. Rather, it is employed solely as a baseline strategy to prompt comparative methods, a practice consistent with recent literature on creative image generation (e.g., [1]).  Additionally, to systematically explore different prompting techniques for a fair baseline comparison, we have conducted comprehensive ablations using prompts such as `rare  < object >` , `innovative < object >`, `unusual < object >`, `rare < object >` as well as providing a negative prompt such as `A normal < object >`  to encourage baseline models toward generating creative outputs. Please see these experiments in our Fig 6.d in the main paper, and  Fig 3 in Appendix.
> > >
> > > Importantly, our proposed framework diverges significantly from these prompting-based methods by autonomously generating creative transformations from simple object names (e.g. `a couch`).  The creative transformations are guided explicitly by underlying creativity principles, resulting in highly detailed and unambiguous prompts, as can be seen from Fig. 5 of our Appendix.  Furthermore, our framework inherently supports user-in-the-loop control, allowing users to explicitly steer the direction of creative generation or editing, further resolving potential ambiguity regarding human intent. Please see Fig. 5(a) of our main paper for human-in-the-loop results.
> > >
> > > We appreciate this feedback and will ensure this clarification and distinction are prominently reflected in the final manuscript.
> > >
> > >
> > > [1]: Han et.al. 2025 "Enhancing Creative Generation on Stable Diffusion-based Models" (CVPR'25)
> > >
> > >
> > > ***Reviewer S95b mentioned that the internal logic of some agents (e.g., the critique agent) is described at a high level. I agree that the writing could be improved by incorporating more concrete implementation details.***
> > >
> > > We agree that additional implementation details would improve the clarity and transparency of our agent architecture. While our main paper provides a high-level overview of each agent’s role due to space constraints, we recognize that readers seeking to understand the core mechanisms, especially the logic of agents like the Critique agent would benefit from more concrete information.
> > >
> > > We kindly note that we provided full interaction logs in Figures 5–8 (in Appendix) and source code in the supplementary material. Specifically,  `code/crea/agents` folder has the full instructions used by our agents, including our Critique agent. We will further elaborate on the implementation of each agent in our final manuscript to ensure transparency.

---

> > > > ### Author Response · Authors · 2025-08-08
> > > >
> > > > Dear Reviewer,
> > > >
> > > > We sincerely thank you for your constructive feedback during the discussion phase. As the discussion period concludes today, we kindly draw your attention to our follow-up response sent yesterday, in which we addressed the additional concerns you raised.
> > > >
> > > > If any further issues remain, we would be happy to provide clarification before the rebuttal period ends.
> > > >
> > > > Best regards,
> > > >
> > > > Authors of CREA

---

### Official Review · Reviewer_S95b · 2025-07-02

**Clarity:** 2
**Significance:** 2
**Originality:** 3
**Rating:** 4
**Confidence:** 2

**Summary:**

This paper proposes CREA, a novel collaborative multi-agent framework for creative image editing and generation. CREA imitates the human creative process by organizing a set of specialized agents—namely, a concept agent, generation agent, critique agent, and refinement agent—that work together in an iterative pipeline. The authors formalize the creative editing task, distinguishing it from standard prompt-based or semantic editing. Extensive experiments, including both quantitative metrics (e.g., diversity, semantic alignment, user studies) and qualitative visualizations, show that CREA significantly outperforms several strong baselines such as InstructPix2Pix and Plug-and-Play methods. The paper is positioned as the first to propose and formalize this agent-based creative editing task.

**Questions:**

Critique Agent Mechanism: Can the authors elaborate on how the critique agent produces structured feedback and how that feedback is utilized during refinement? Are the critiques learned or rule-based?

Failure Cases: It would be helpful to include examples of failure modes—when CREA produces incoherent or uncreative edits—and how the system might be improved to mitigate these.

System Latency and Efficiency: Please provide measurements (e.g., average number of iterations, inference time per step) to better understand CREA's feasibility for interactive use.

Comparison with Non-Agentic Iterative Methods: How does CREA compare to a monolithic iterative approach (e.g., looping Plug-and-Play generations with feedback) in terms of both performance and interpretability?

Ethics and Misuse: Given the potential use of such systems in generating deepfakes or manipulated visuals, have the authors considered misuse risks? Are any mitigation strategies discussed?

**Ethical Concerns:**

["NO or VERY MINOR ethics concerns only"]

**Limitations:**

The authors mention the subjective nature of creativity and the challenges of evaluation, which is appropriate. However, further discussion on scalability, domain transfer, and potential ethical misuse (e.g., for disinformation or deceptive content) is recommended.

Suggested improvement: Include a section on computational cost and a mitigation discussion on possible societal misuse.

**Quality:**

2

**Strengths And Weaknesses:**

Strengths
Novel Task Definition: The paper introduces and formalizes creative image editing as a new task distinct from conventional image editing, with a focus on artistic novelty and agentic control.

Modular Multi-Agent Framework: CREA’s agent decomposition—conceptualization, generation, critique, refinement—is both intuitive and grounded in cognitive creativity theories.

Strong Empirical Results: The experiments cover multiple datasets, use both automatic metrics (LPIPS, FID, CLIP alignment) and human evaluation, and show consistent gains over strong baselines.

Reproducibility: The method is well-documented, with a clear architecture diagram, ablation studies, and statements regarding code release.

Originality: To the best of my knowledge, this is the first multi-agent approach designed specifically for the creative editing domain.

Weaknesses
Agent Implementation Details: The internal logic of some agents (e.g., the critique agent) is described at a high level. More details on how each agent’s output is used, especially how conflicting feedback is resolved in the refinement step, would improve clarity.

Scalability and Efficiency: CREA involves multiple inference rounds and agent coordination, which may limit real-time usability. A discussion of system latency and computational cost is missing.

Evaluation Limitations: While the authors use CLIP-based and perceptual metrics, creativity is inherently subjective. The human evaluation setup is relatively small (50 samples); more robust or crowd-sourced studies would strengthen claims.

Generalization: CREA is evaluated primarily on artistic or fantasy domains. It is unclear how well it generalizes to domains like product design or real-world edits where creativity may be more constrained.

---

> ### Author Rebuttal · Authors · 2025-07-30
>
> **Q1: The internal logic of some agents (e.g., the critique agent) is described at a high level. More details on how each agent’s output is used, especially how conflicting feedback is resolved in the refinement step, would improve clarity.**
>
> We thank the reviewer and agree that clarifying agent logic enhances the paper. CREA’s multi-agent loop is implemented via AutoGen’s ConversableAgent API, with each agent assigned bounded memory and tools (see Supplementary Section, Page 35). The Critic Agent (A4) scores images across six creativity principles, producing a vector S = {S₁,…,S₆}, and operates with window_size = 1 to ensure unbiased, sample-specific evaluation. The Creative Director interprets these scores in context of the original blueprint and resolves conflicts when principle-level feedback misaligns with creative intent. The Refinement Strategist then identifies low-performing dimensions and generates instructions, which the Prompt Architect integrates into the next-generation prompt. This process repeats until the creativity index exceeds threshold or max steps are reached.
>
> Full interaction traces are shown in Figures 5–8 (in Appendix) and the logic is formalized in Algorithm 1 (Appendix). Additionally, we kindly point out that we provided the source code in the supplementary material, and `code/crea/agents` folder has the full instructions used by our agents for transparency. We will clarify this loop more explicitly in the final manuscript to improve clarity.
>
>
>
> **Q2: Can the authors elaborate on how the critique agent produces structured feedback and how that feedback is utilized during refinement? Are the critiques learned or rule-based?**
>
> The Critique agent produces structured feedback by prompting a multimodal LLM-as-a-Judge using a predefined creativity template distilled from  creativity principles from well-established literature on creative computing, scoring each generated image across six creativity principles. The critiques are not learned or rule-based but generated via zero-shot evaluations from the LLM using standardized criteria. These scores are then used by the Refinement Strategist to identify underperforming dimensions and generate targeted prompt refinements, as detailed in Appendix Figures 5-8.
>
> We also emphasize that our source code has the full instructions of our agents, including critique under Supplementary, `code/crea/agents/critic.py`.
>
> **Q3: CREA involves multiple inference rounds and agent coordination, which may limit real-time usability. A discussion of system latency and computational cost is missing./Please provide measurements (e.g., average number of iterations, inference time per step) to better understand CREA's feasibility for interactive use.**
>
> We appreciate the reviewer raising this practical consideration.  As detailed in Sec 4, Experimental Setup section, currently, a complete pipeline execution, including iterative refinement, takes approximately 3–5 minutes on a 48GB NVIDIA L40 GPU. We also conducted additional timing analyses and provide the following concrete measurements:
>
> Avg iterations per run: 1.50 iterations
>
> Avg total time per run: 176.54 seconds (2.94 minutes)
>
> Avg time per iteration: 117.70 seconds (1.96 minutes per iteration)
>
> However, we emphasize that this latency remains substantially lower than the typical time investment required by manual user-driven approaches, which usually involve extensive trial-and-error prompting to achieve comparable creative outputs. Thus, our framework aims to reduce the human effort and overall creative turnaround time. We will expand the revised manuscript to explicitly discuss computational cost and system latency.
>
> **Q4: While the authors use CLIP-based and perceptual metrics, creativity is inherently subjective. The human evaluation setup is relatively small (50 samples); more robust or crowd-sourced studies would strengthen claims.**
>
> We acknowledge that our primary experiments included a relatively modest user study with 50 participants (Sec. 4.2 in the main paper). To further address the subjectivity and robustness concerns, we also provided additional comprehensive evaluations using a multimodal LLM-as-a-Judge framework (Appendix Sec. A.2, Table 1), simulating human-like subjective assessments across several creativity dimensions including Originality, Expressiveness, Aesthetic Appeal, Technical Execution, Unexpected Associations, and Interpretability.
> Moreover, our appendix (NeurIPS_2025__CREA.pdf) provides an extensive set of qualitative results across all evaluated concepts, covering the entire set of 600 generated and edited images without cherry-picking (Appendix Figures 9–32). These comprehensive visuals reinforce our claims, alleviating concerns of selective curation.
>
> **Q5: CREA is evaluated primarily on artistic or fantasy domains. It is unclear how well it generalizes to domains like product design or real-world edits where creativity may be more constrained.**
>
> We appreciate the reviewer’s point. CREA is designed as a general-purpose creative reasoning framework, and while our examples often show artistic and fantastical domains where creativity is most visually measurable, CREA supports constraint-driven generation through its predefined creativity template, blueprint generation mechanism and user-in-the-loop controls, which can be modified to suit the user’s needs to  seamlessly leverage CREA’s capabilities for any domain. We will clarify this extensibility in the revision.
>
> **Q6: It would be helpful to include examples of failure modes; when CREA produces incoherent or uncreative edits and how the system might be improved to mitigate these.**
>
> We appreciate the reviewer emphasizing the importance of transparently discussing potential failure modes. While we are unable to directly include visual examples in this rebuttal due to NeurIPS guidelines prohibiting supplementary images or URLs, we kindly point out to the examples of certain limitations (Fig. 6(e) in the main paper). These cases illustrate unintended semantic shifts or background hallucinations, such as undesired human figures spontaneously appearing or backgrounds unintentionally darkening based on the semantics of edited objects. In the camera-ready version, we plan to further expand this discussion by explicitly providing additional visual examples of failure modes.
>
> **Q7: How does CREA compare to a monolithic iterative approach (e.g., looping Plug-and-Play generations with feedback) in terms of both performance and interpretability?**
>
> We appreciate the reviewer’s thoughtful concern regarding the complexity of our framework. While a single-step LLM prompt, even when enriched, can indeed generate reasonably creative outputs, we respectfully argue that such a monolithic approach fundamentally lacks the capacity to model creativity as an iterative, and structured optimization process.  Indeed, our ablation studies (see Table 2 in main) provide strong empirical support for this argument. Even a robust single-step GPT-4o baseline ("Base") significantly performs worse.
>
> **Q8: Given the potential use of such systems in generating deepfakes or manipulated visuals, have the authors considered misuse risks? Are any mitigation strategies discussed?**
>
> We appreciate the reviewer raising this important concern. Like any powerful generative system, CREA carries potential misuse risks, such as generating manipulated or misleading visuals. As noted in Section 5, we advocate for CREA to be used strictly as a human-in-the-loop co-creative assistant, not as an autonomous generator. In the revised manuscript, we will outline concrete mitigation strategies, including safety filters, content moderation, and controlled access protocols for sensitive deployments.
>
> **Q9: The authors mention the subjective nature of creativity and the challenges of evaluation, which is appropriate. However, further discussion on scalability, domain transfer, and potential ethical misuse (e.g., for disinformation or deceptive content) is recommended.**
>
> We appreciate the reviewer acknowledging our discussion about creativity's inherent subjectivity and evaluation challenges. In our revised manuscript, we will explicitly expand our discussion to cover:
>
> * Scalability: Clearly outlining computational costs, latency trade-offs, and strategies (e.g., agent parallelization, model distillation) that could enhance system scalability.
> * Domain Transfer: Discussing CREA’s potential applicability to diverse domains beyond those demonstrated, emphasizing how the modular multi-agent design allows for adaptability and extension to abstract tasks that are challenging to articulate explicitly.
> * Ethical Misuse: Providing detailed consideration of risks associated with misuse, such as deepfakes or deceptive visual content, along with explicit strategies for mitigating these risks, including content moderation, watermarking, and responsible deployment guidelines.
>
> **Q10: Include a section on computational cost and a mitigation discussion on possible societal misuse.**
>
> In practice, CREA converges within 2–3 agentic refinement rounds, averaging 3–5 minutes per image, comparable to real-world manual prompting workflows. Each agent role uses compact GPT-4o prompts, resulting in 6–9 LLM calls per sample. At current OpenAI pricing, this equates to a per-image cost of roughly 0.02–0.05 USD, comparable to iterative user prompt refinement. Future work will explore distilling agent behavior into smaller or open-weight models to further reduce cost without compromising reasoning quality. We will summarize these computational and ethical considerations in the revised Broader Impact section.

---

> ### Author Response · Authors · 2025-08-06
>
> Dear reviewer,
>
> We appreciate your valuable review, and we addressed all your concerns in our rebuttal. As the NeurIPS discussion phase will conclude soon, we kindly remind you that if you have any further questions or require additional clarifications on our paper, we are more than happy to address them promptly.
>
> Best,
>
> Authors of CREA

---

> ### Author Response · Authors · 2025-08-08
>
> Dear reviewer,
>
> We thank you for your constructive feedback during the review process. While we have not yet received a response during the discussion phase, we hope that our detailed rebuttal (addressing ***all*** stated concerns) has satisfactorily resolved the issues raised.
>
> Should any concerns remain, we would be happy to provide further clarification before the discussion period concludes today.
>
> *We would also like to highlight that our appendix contains over 600 visual examples illustrating the diversity and editing capabilities of our method, accompanied by the full source code detailing the exact prompts and commands executed by our agents.*
>
>
> Best,
>
> Authors of CREA

---

### Official Review · Reviewer_mq9P · 2025-07-02

**Clarity:** 3
**Significance:** 2
**Originality:** 3
**Rating:** 4
**Confidence:** 4

**Summary:**

The paper presents an agentic framework CREA for text-to-image generation and editing, with a special focus on enhancing creativity of the generative process. The framework adopts several models from LLMs/VLMs to Flux-based ControlNets. The overall steps are pretty standard, beginning from a planning stage to generate creative text prompts, followed by conditional image generation, post-generation evaluation and refinement stages to rewrite the prompts and keep updating the generated images. Several baseline T2I and image editing models are compared to emphasize the strong creativity of CREA and its wide applications.

**Questions:**

See weaknesses. Please clarify accordingly if there are misunderstandings.

**Ethical Concerns:**

["NO or VERY MINOR ethics concerns only"]

**Final Justification:**

The authors have addressed most of my concerns. It is strongly suggested to show a quantitative analysis of the progressive improvements in the next revision.

**Limitations:**

See weaknesses.

**Paper Formatting Concerns:**

N / A

**Quality:**

2

**Strengths And Weaknesses:**

Strengths:
- The agent framework addresses creative image editing, which is a less-addressed dimension in previous work. The framework design, including the prompt planning, evaluation and refinement, make sense and seem to demonstrate good qualitative results.
- The presented qualitative results are nice and the quantitative evaluation are also helpful in understanding the capabilities of the framework.
- The overall writing is clear and easy to follow.

Weaknesses:
The major weaknesses, as far as I could see, lie the following two dimensions:
- Limited editing capabilities. While the qualitative demonstrations look nice, the editing prompts used in evaluation fail to address more general editing scenarios such as adding, moving, removing objects and etc -- operations beyond just changing appearance of the original object. Besides, the exact "editing" setup is not well-presented. Is the editing based on language instructions or changing phrases / expressions in the original prompts used for generation? I couldn't tell from the figures or tables as the editing prompts are not shown and the baselines are mix of instruction-based (instructpix2pix) and prompt-based editing models (turboedit, sdedit).
- Insufficient/unfair evaluation.
  - It is hard to tell if the improvement comes from the base model difference. For example, TurboEdit adopts SDXL-Turbo while CREA adopts Flux1.-dev, a better image generator that the former --  not to mention InstructPix2Px and SDEdit.
  - Since  CREA is an agent framework, it may take several iterations of the whole pipeline to get a sample that passes the critic's judgement threshold. What's a more fair baseline here is to select the top-1 out of k generation results using the base model and compare to CREA. While this is not necessary in the rebuttal, it seems reasonable to show the overall evaluation results from each iteration of CREA to show that the generation results keep improving thanks to the critic and refinement. Otherwise, it could be that the whole framework is doing rejection sampling and eventually stop at a good enough sample.

---

> ### Author Rebuttal · Authors · 2025-07-30
>
> We appreciate the reviewer's insightful comments. Below, we answer your concerns:
>
> ***Q1: Limited editing capabilities. While the qualitative demonstrations look nice, the editing prompts used in evaluation fail to address more general editing scenarios such as adding, moving, removing objects and etc -- operations beyond just changing appearance of the original object.***
>
> We appreciate the reviewer highlighting this important limitation. Indeed, our current demonstrations and evaluation prompts primarily focus on appearance transformations and creative stylistic variations, and do not explicitly showcase more general editing operations such as adding, or removing objects within scenes.
>
> However, we clarify that our core agentic framework is not fundamentally restricted to appearance editing tasks. The underlying architecture is highly modular, flexible, and capable of integrating additional editing methods that explicitly support structural modifications - such as adding, or removing objects through appropriate conditioning mechanisms and spatial guidance (e.g., segmentation maps, bounding boxes, and layout controls provided by ControlNet).
>
> To showcase this capability, we conducted additional experiments involving object addition and removal within a realistic living room scene. Specifically, we tasked our method with adding a "bear" on top of a couch and removing an existing "vase." We conducted a user study with 39 participants to evaluate CREA’s ability to perform disentangled edits in complex, multi-object scenes across various edit types, including style, color changes, object addition, and removal. The results showed overwhelming agreement that CREA preserves the overall structure of the image while making targeted edits: 97.4% rated the color change as highly disentangled, and 100% rated the object addition, removal, and creative transformations as successful without affecting other elements in the scene. These findings confirm that CREA enables fine-grained, high-fidelity edits in multi-object settings while maintaining disentanglement and visual coherence.
> Due to NeurIPS rebuttal guidelines, we are not allowed to upload visuals here. We will explicitly include these additional experiments, quantitative results, and corresponding visual examples in the final manuscript, further elaborating on the modularity and extensibility of CREA in broader editing scenarios.
>
>
> ***Q2: The exact "editing" setup is not well-presented. Is the editing based on language instructions or changing phrases / expressions in the original prompts used for generation? I couldn't tell from the figures or tables as the editing prompts are not shown and the baselines are mix of instruction-based (instructpix2pix) and prompt-based editing models (turboedit, sdedit).***
>
> We acknowledge that mixing instruction-based and prompt-based baseline methods might have introduced confusion regarding the exact editing methodology. The reason for including both instruction-based and prompt-based baseline methods was to comprehensively evaluate against state-of-the-art techniques spanning diverse editing paradigms.
>
> To clarify our exact editing setup: given an input image (e.g., a "cup") and optional user-in-the-loop guidance (e.g., "a monster cup"), our agentic framework autonomously synthesizes detailed creative editing instructions grounded in structured creativity principles. These instructions are then applied through a disentangled editing process with Flux and ControlNet, ensuring targeted and interpretable creative transformations.
>
> Additionally, we kindly point out that we provided the source code in the supplementary material for transparency, and we will explicitly include these generated editing instructions in the Appendix by the camera-ready version to facilitate clearer comparisons.
>
>
> ***Q3: It is hard to tell if the improvement comes from the base model difference. For example, TurboEdit adopts SDXL-Turbo while CREA adopts Flux1.-dev, a better image generator that the former -- not to mention InstructPix2Px and SDEdit.***
>
> We thank the reviewer for raising this important concern regarding the base model differences. Indeed, baseline methods such as TurboEdit, InstructPix2Pix, and SDEdit utilize various underlying diffusion models, including SDXL-Turbo and Stable Diffusion. We selected these baselines to represent state-of-the-art methods across diverse editing paradigms.
>
> To directly investigate whether our performance improvements stem primarily from our framework or the choice of the base model (Flux-1-dev), we conducted additional ablation studies (Sec. 4.3, Fig. 6a) where we implemented our CREA framework using alternative base models such as SDXL and DALL-E. These experiments demonstrate that our agentic framework is able to generate creative images across different base models.
>
> Additionally, we conduct an experiment using RF-Inversion (Rout et al., 2024) a Flux based image editing method to further demonstrate that our performance is not solely due to the base model.
>
> | Method      | CLIP  ↑   | LPIPS* ↑ | Vendi ↑ | Dino ↑ |
> | ----------- | ----------------- | ----------------- | ----------------- | ----------------- |
> | RF-Inversion | 0.369 ± 0.045 | 0.269 ± 0.084 | 3.35 ± 1.53 | 0.743±0.179 |
> | **Ours** | **0.417 ± 0.030**     | **0.414 ± 0.157** | **3.70 ± 1.97** | **0.744 ± 0.185** |
>
> We will include these experiments to our final manuscript.
>
> ***Q4: Since CREA is an agent framework, it may take several iterations of the whole pipeline to get a sample that passes the critic's judgement threshold. What's a more fair baseline here is to select the top-1 out of k generation results using the base model and compare to CREA. While this is not necessary in the rebuttal, it seems reasonable to show the overall evaluation results from each iteration of CREA to show that the generation results keep improving thanks to the critic and refinement. Otherwise, it could be that the whole framework is doing rejection sampling and eventually stop at a good enough sample.***
>
> We thank the reviewer for raising this valuable point. We acknowledge that due to our iterative agent-based pipeline, multiple refinement rounds indeed occur before obtaining a sample that meets the critic's creativity threshold. While superficially similar to rejection sampling, we emphasize that our iterative refinement mechanism actively utilizes the critic's detailed feedback to inform targeted prompt adjustments, rather than simply discarding unsuccessful results.
>
> We fully agree that explicitly demonstrating improvement across iterations would clearly illustrate the value of our critic-based refinement approach. In fact, we have already conducted comprehensive iterative analyses, as presented in Fig. 6(c) in the main manuscript. This figure clearly shows progressive improvement across successive iterations, confirming that our agentic refinement mechanism systematically enhances output quality rather than merely selecting from multiple random samples.
>
> Nevertheless, we appreciate the reviewer's suggestion for additional clarity. In our revised manuscript, we will include detailed discussion on this.

---

> ### Author Response · Authors · 2025-08-06
>
> Dear reviewer,
>
> We appreciate your valuable review, and we addressed all your concerns in our rebuttal. As the NeurIPS discussion phase will conclude soon, we kindly remind you that if you have any further questions or require additional clarifications on our paper, we are more than happy to address them promptly.
>
> Best,
>
> Authors of CREA

---

> ### Author Response · Authors · 2025-08-08
>
> We thank the reviewer for their constructive feedback during the review process. While we have not yet received a response during the discussion phase, we hope that our detailed rebuttal (addressing all stated concerns and additional experiments) has satisfactorily resolved the issues raised.
>
> Should any concerns remain, we would be happy to provide further clarification before the discussion period concludes today.
>
> *We would also like to highlight that our appendix contains over 600 visual examples illustrating the diversity and editing capabilities of our method, accompanied by the full source code detailing the exact prompts and commands executed by our agents.*

---

### Official Review · Reviewer_2FiS · 2025-07-03

**Clarity:** 3
**Significance:** 3
**Originality:** 1
**Rating:** 4
**Confidence:** 4

**Summary:**

This paper introduces CREA, a new collaborative multi-agent AI framework designed for creative image editing and generation. CREA mimics the human creative process by using specialized AI agents—including a Creative Director, Prompt Architect, Generative Executor, Art Critic, and Refinement Strategist—to conceptualize, generate, critique, and refine visual content. Guided by creativity principles, CREA supports autonomous, disentangled, and user-guided edits for both images and videos. Experiments show CREA outperforms current state-of-the-art methods in diversity, semantic alignment, and human-perceived creativity.

**Questions:**

1: Is the most important thing to generate a "fancy" image a better-oriented prompt?

2: Does the whole system aim to generate a suitable prompt for generation model to generate "creative" images?

**Ethical Concerns:**

["NO or VERY MINOR ethics concerns only"]

**Final Justification:**

After checking the authors' rebuttal and other reviews, most of my concerns have been addressed. Therefore, I decide to raise my rating to borderline accept.

**Limitations:**

Yes.

**Quality:**

3

**Strengths And Weaknesses:**

Strengths:

1: This paper clearly aims to propose a new task of "creative image editing," going beyond conventional, prompt-driven editing to focus on multi-faceted, disentangled, and artistically rich compositions. The motivation for an agentic, collaborative approach is well contextualized with respect to the limitations of current prompt-to-image systems.

2: Experimental results showcase the qualitative advantages of CREA for both creative editing and generation, with side-by-side comparisons to leading methods (e.g., SDEdit, InstructPix2Pix, ConceptLab). The paper provides rigorous quantitative comparisons across a spectrum of metrics (CLIP, LPIPS, VENDI, DINO, and user studies); CREA achieves higher scores on most creative and diversity-based measures for both editing and generation tasks.

3: The choice of six creativity principles for prompt generation and evaluation is grounded in established theories, lending interpretability and traceability to CREA's process.

4: This paper is easy to understand and follow.

Weaknesses:

1: **Limited Novelty and Overlap with Existing Agent-Based Frameworks**. The core technical contribution of this paper shows overlap with prior work, particularly with agent-based systems for image generation like GenArtist [1]. While the authors argue that CREA's novelty lies in its multi-agent collaborative approach for creative editing, as opposed to GenArtist's single-agent structure, the fundamental concept of using agents for generation, critique, and refinement is not new. The framework's scope is narrowed to "creativity," which may be perceived as applying an existing agent-based methodology to a more specific problem domain rather than introducing a paradigm-shifting approach.

2: **Over-Engineering of the Prompt Generation Process**. A central contribution of the CREA framework is its complex, multi-stage pipeline designed to produce a "high-creativity" prompt (P_c). However, it is questionable whether this elaborate, multi-agent architecture is necessary. A powerful Large Language Model (e.g., GPT-4o, which is used by the agents themselves ) could likely achieve a comparable result with a much simpler, single-step instruction to expand a basic concept into a rich, creative prompt. Although the ablation study shows the full framework outperforms a "base" LLM prompt, it does not sufficiently justify whether the marginal gains in performance warrant the significant increase in system complexity, latency, and operational overhead.

3: **Unfair Experimental Comparison Due to Simple Prompting**. While the paper claims superiority over existing editing and generation methods, the comparison setup raises fairness concerns. Competing baselines are primarily evaluated using simple prompts such as "a creative <object>,” whereas the CREA framework benefits from an elaborate, agent-generated prompt construction pipeline that synthesizes multi-dimensional creativity cues. The authors do not investigate whether giving similarly enriched prompts to the baseline models would close the performance gap. This undermines the strength of the comparison and leaves open the question of whether the performance gain is due to the agentic framework or merely due to better prompting.

4: **Limited and Unconvincing Visualization of Prompts and Outputs**. The qualitative examples throughout the paper showcase visual results conditioned on extremely simple or generic prompts (e.g., "a couch", "a monster", "a cup"), yet these do not clearly reflect the internal creative prompt transformations that are claimed to drive the system's creativity. Without visualizing or explicitly disclosing the final effective prompts used by the model (i.e., the ones passed to the text-to-image or editing model after multi-agent planning), it is difficult to assess whether the outputs are meaningfully creative or just well-curated samples. The limited and non-transparent prompt-to-image mappings weaken the interpretability and persuasiveness of the visual comparisons.

[1] Wang, Zhenyu, et al. "Genartist: Multimodal llm as an agent for unified image generation and editing." Advances in Neural Information Processing Systems 37 (2024): 128374-128395.

---

> ### Author Rebuttal · Authors · 2025-07-30
>
> We thank the reviewer for their detailed and helpful feedback. Please find below our responses to each of your questions.
>
> **Q1: The core technical contribution overlaps with GenArtist, which uses a single agent architecture.**
>
> We appreciate the reviewer highlighting this point regarding the relationship between our method and GenArtist. While GenArtist indeed utilizes a single-agent architecture for general image generation and editing, we clarify that our work fundamentally differs both in design and scope.
>
> - CREA explicitly introduces and addresses, for the first time, the task of creative image editing, which emphasizes iterative, autonomous transformations guided by structured creativity principles. In contrast, GenArtist mainly targets general-purpose multimodal editing tasks, without specifically addressing or defining creativity-oriented editing or systematically measuring outputs against formal creativity dimensions.
> - While GenArtist uses a single multimodal LLM agent that directly performs editing tasks via a general-purpose prompt-based method, CREA proposes a novel multi-agent collaborative pipeline with specialized agents (Creative Director, Prompt Architect, Generative Executor, Art Critic, Refinement Strategist). This structured, role-based division of labor ensures a more systematic, interpretable, and iterative refinement process, closely mimicking human creative workflows.
> - CREA incorporates explicit iterative refinement loops guided by detailed, agent-generated feedback grounded in well-established creativity theories (e.g., originality, aesthetics, unexpected associations). GenArtist, by comparison, operates without explicit structured evaluation against a formal criteria (e.g. creativity).
>
> Due to NeurIPS rebuttal guidelines, we are not allowed to include additional visuals here. However, to further substantiate our  performance differences, we conducted a human evaluation on Prolific involving 25 users, explicitly comparing our CREA framework against GenArtist (where we provide 'creativity' keyword in the editing prompt as 'a creative bike'). Our results indicate that CREA significantly outperforms GenArtist across user-assessed creativity metrics and overall visual quality.
>
> | Method      | Q1 (Creativity) ↑ | Q2 (Disentangled) ↑ |
> | ----------- | ----------------- | ------------------- |
> | GenArtist   | 2.893 ± 1.206     | 2.850 ± 1.193       |
> | **Ours**    | **4.299 ± 0.903** | **3.726 ± 1.017**   |
>
> Furthermore, to provide additional evidence that CREA’s multi-agent mechanism is not equivalent to a single-step, monolithic LLM approach, we conducted detailed ablation studies (see Table 2 in the main paper). Specifically, when moving from a strong single-step LLM baseline ("Base"), where we prompt an LLM to directly generate a creative object description, to our complete CREA pipeline, we observe substantial improvements in LPIPS-Diversity and VENDI metrics for generation, as well as simultaneous gains in LPIPS (reflecting creative diversity) and DINO (indicating structural and semantic preservation) for editing. These ablations conclusively demonstrate the unique value of CREA’s structured multi-agent design.
>
> We will explicitly include these detailed comparative evaluations, visual examples, and extended discussions about the key differentiating factors between CREA and GenArtist in our revised manuscript.
>
> **Q2: The CREA framework’s multi-stage, multi-agent pipeline may be unnecessarily complex, as a powerful LLM like GPT-4o could likely generate comparably creative prompts with a simpler, single-step instruction. While the ablation study shows improved performance, it does not convincingly justify the added complexity, latency, and overhead.**
>
> We appreciate the reviewer’s thoughtful concern regarding the complexity of our framework. While a single-step LLM prompt, even when enriched, can indeed generate reasonably creative outputs, we respectfully argue that such a monolithic approach fundamentally lacks the capacity to model creativity as an iterative, and structured optimization process. Creativity is inherently dynamic, involving multiple, often conflicting principles - such as maximizing originality without sacrificing coherence or interpretability, which cannot be effectively captured or optimized in a single-step instruction.
>
> CREA explicitly addresses this complexity by adopting a closed-loop, multi-agent architecture: the Art Critic scores generated images along six well-defined creativity principles; the Refinement Strategist identifies dimensions needing improvement; the Prompt Architect adaptively refines prompts based on specific feedback; and the entire iterative process is orchestrated by a blueprint-aware Creative Director. This design enables CREA to systematically disentangle and iteratively enhance distinct creative attributes through explicit feedback, contextual memory, and adaptive self-correction, capabilities that single-step prompting inherently lacks. As noted by the reviewer, our ablation studies (see Table 2 in main) provide strong empirical support for this argument. Even a robust single-step GPT-4o baseline ("Base") significantly performs worse.
>
> Regarding the latency and overhead: CREA typically achieves convergence within 2–3 iterations, resulting in a total latency of approximately 3–5 minutes. Considering that typical user-led prompting workflows involve extensive trial-and-error, often taking substantial manual effort and time, we believe CREA represents a reasonable alternative.
>
> Thus, CREA’s architectural complexity is not arbitrary; rather, it is essential for transforming creative image generation and editing from a single-step expansion into a structured optimization problem over interpretable, controllable, and systematically improvable creative objectives. Nevertheless, in our final manuscript, we will explicitly discuss and clarify this complexity-performance trade-off.
>
> **Q3: While the paper claims superiority over existing methods, its comparisons are potentially unfair; baselines use simple prompts, while CREA benefits from complex, agent-generated ones. Without testing baselines on equally enriched prompts, it’s unclear if the gains come from the agentic design or just better prompting.**
>
> We appreciate the reviewer highlighting this important aspect. Our intent was to fairly evaluate each method according to its inherent design: baseline models naturally depend on direct user-provided prompts and lack built-in mechanisms for structured creativity reasoning or sophisticated prompt elaboration. In contrast, CREA’s key novelty is precisely its agentic framework designed to autonomously generate enriched and multi-dimensional creativity cues.
>
> We also emphasize that our agentic framework is designed to be flexible, enabling seamless integration with existing editing methods such as Ledits++. We have integrated Ledits++ into the CREA pipeline, and our preliminary results demonstrate that CREA enhances the creative generation capabilities of Ledits++ by providing richer, context-aware prompts grounded in explicit creativity principles. Due to NeurIPS rebuttal guidelines, we are unable to include visual examples here, but we will explicitly incorporate these additional results in our final manuscript.
>
> **Q4: Limited and Unconvincing Visualization of Prompts and Outputs/The paper showcases visual results from simple prompts but does not reveal the final, transformed prompts used by the system.**
>
> We appreciate the reviewer’s insightful comment regarding prompt transparency and interpretability. Due to space constraints in the main manuscript, we were unable to include all the final effective prompts directly in the images. However, in Appendix Fig. 5, we explicitly illustrate an example demonstrating how a simple input keyword ("couch") evolves through multiple stages of prompt elaboration and evaluation by our agents, resulting in a detailed, enriched prompt that drives the creative output.
>
> Additionally, we kindly point out that we provided the source code in the supplementary material, and code/crea/agents folder has the full instructions used by our agents for transparency, and we will include the transformed prompts in the appendix in the final manuscript.
>
> Moreover, we also emphasize that our  supplementary materials “website” and our supplementary pdf  (NeurIPS_2025__CREA.pdf) comprehensively present all 600 visual results generated in our evaluation, without any cherry-picking or selective curation. We hope that the extensive number of examples supports the robustness of our qualitative claims.
>
> **Q5: Is the most important thing to generate a "fancy" image a better-oriented prompt? Does the whole system aim to generate a suitable prompt for generation model to generate "creative" images?**
>
> We appreciate the reviewer raising this point. While generating a high-quality prompt certainly impacts the creativity of image outputs, our primary goal goes beyond merely crafting better-oriented prompts. Instead, the core novelty and strength of our framework lie in the structured, collaborative reasoning process among multiple specialized agents, each grounded in established creativity principles. This multi-agent interaction ensures not only a "fancy" or enriched prompt but also systematically addresses aspects such as originality, expressiveness, aesthetics, technical execution, unexpected associations, and interpretability. Thus, the framework's advantage is not just better prompting, but rather an explicit, interpretable, and iterative reasoning pipeline designed specifically to foster and enhance genuine creative outputs.
>
> Moreover, we emphasize that our agentic framework can potentially generalize to other abstract and nuanced concepts that are challenging for users to articulate explicitly in text, such as the abstract concepts depicted in “GANalyze: Toward Visual Definitions of Cognitive Image Properties”, Goetschalckx et. al.

---

> ### Author Response · Authors · 2025-08-06
>
> Dear reviewer,
>
> We appreciate your valuable review, and we addressed all your concerns in our rebuttal. As the NeurIPS discussion phase will conclude soon, we kindly remind you that if you have any further questions or require additional clarifications on our paper, we are more than happy to address them promptly.
>
> Best,
>
> Authors of CREA

---

> ### Author Response · Authors · 2025-08-08
>
> Dear reviewer,
>
> We would like to express our sincere gratitude for your constructive feedback during the reviewing phase. If you find that all concerns have been satisfactorily addressed, we respectfully request that you update your final justification and score to conclude the discussion phase.
>
> In our rebuttal, ***we provided comparisons with GenArtist as requested, as well as clarifying *all* the raised concerns***. If you still have any remaining concerns, we will make our best effort to address them before the rebuttal period closes today.
>
> *We would also like to highlight that our appendix contains over 600 visual examples illustrating the diversity and editing capabilities of our method, accompanied by the full source code detailing the exact prompts and commands executed by our agents.*
>
> Best,
>
> Authors of CREA

---

### Official Review · Reviewer_LYr2 · 2025-07-06

**Clarity:** 3
**Significance:** 2
**Originality:** 3
**Rating:** 3
**Confidence:** 3

**Summary:**

The paper introduces CREA, a multi-agent framework for creative image editing and generation. It includes five main agents into its pipeline: creative director, prompt architect, generative executor, art critic and refinement strategist. The whole workflow is divided into 3 phases: pre-generation planning, post-generation evaluation and self-enhancement. During the generation and refinement, 6 creativity principles are provided as common guidelines. If the aggregated creativity index fails to meet the standard, the generation loop will repeat and update the prompt&image iteratively.

**Questions:**

- While the proposed method involves GPT-4o in some phases, I am curious about the performance of a workflow that uses GPT-4o as both the agents and image generator.
- Does CREA often tend to generate imaginative objects that does not exist in the real world? That might limit the use case of its application.

**Ethical Concerns:**

["NO or VERY MINOR ethics concerns only"]

**Limitations:**

Yes, it is discussed in the Appendix.
Beyond that, my main concerns are about the fairness & comprehensiveness on the evaluation and comparison, and also the limited use scope of this workflow.

**Paper Formatting Concerns:**

No formatting issue.

**Quality:**

2

**Strengths And Weaknesses:**

- The paper defines a new task of creative image editing and introduces the first work on such task.
- The pipeline is flexible, can be easily extended for video generation.
- Fig. 1 is not showed in a very intuitive way; the video can be displayed in frames; the generated prompts are also not shown. “Baseline” is not specified either.
- The generation and refinement are mainly based on prompt design and manipulation, but what if the prompt contains ambiguity? (e.g., when the user wants to edit on a specific location which is tiny and has an irregular shape) The proposed method lacks fine-grained control. The editing ability can be greatly enhanced if various more visual inputs can be supported (e.g., masks, arrows, etc.).
- In Fig. 3, the user’s initial intention or purpose is not clearly illustrated, and it’s hard to evaluate the visual quality of the generated images.
- In the middle group of Fig. 3, the background is often modified when it is not supposed to be touched.
- During the evaluation phase, only using creativity criteria is limited to obtain a comprehensive evaluation on the generated image. Metrics that measure aspects such as background preservation, identity preservation are missing.
- In the user study results, the proposed method does not show a very obvious advantage compared with the other baselines.
- In the comparisons with the baselines, the authors simply use “creative <object>” as the prompt, which is too ambiguous to guide the generation, lacking fairness.

---

> ### Author Rebuttal · Authors · 2025-07-29
>
> We thank the reviewer for their thoughtful and constructive feedback. We address each of the reviewer's comments point-by-point below.
>
> **Q1: Fig. 1 is not showed in a very intuitive way; the video can be displayed in frames; the generated prompts are also not shown. “Baseline” is not specified either.**
>
> Thank you for the constructive feedback. In fact, in line with your suggestions, our Fig. 5(c) presents five representative frames per video and specifies CogVideoX as the baseline. However, these critical details were indeed not included in the teaser Fig. 1. For clarification, the videos in Fig 1 were generated with the following text prompts: *'A creative train chugging along its track'* and *'A creative dog running towards the camera'*. We will revise Fig. 1 in the camera-ready version to explicitly indicate the baseline as CogVideoX, include representative video frames, and display the corresponding prompts to improve clarity and self-containment.
>
> **Q2: The proposed method lacks fine-grained control. The editing ability can be greatly enhanced if various more visual inputs can be supported (e.g., masks, arrows, etc.)**
>
> We thank the reviewer for highlighting this important consideration regarding fine-grained spatial control. We would like to clarify that our current implementation utilizes ControlNet as the generative backend, which inherently supports various types of visual conditioning, including depth maps, segmentation masks, edge maps, and more. Thus, as suggested by the reviewer, our framework is readily capable of incorporating additional visual inputs to enhance localization and precision. Moreover, our multi-agent framework autonomously determines and optimizes conditioning parameters (e.g., conditioning scales, guidance strength), enabling the system to dynamically adjust the granularity of edits from fine to coarse as necessary.
>
> Due to NeurIPS rebuttal guidelines, we are unable to include additional visual examples or external links here. However, in our revised manuscript, we will clearly highlight this existing capability, include examples of fine-grained editing (such as performing a ‘creative edit’ on a small object (e.g. a vase) in a busy living room scene) and  explicitly discuss the straightforward extensibility of our framework to incorporate additional visual inputs for even finer control.
>
> **Q3: In Fig. 3, the user’s initial intention or purpose is not clearly illustrated, and it’s hard to evaluate the visual quality of the generated images.**
>
> We apologize for any confusion regarding Fig. 3. To clarify, Fig. 3(a) illustrates our framework's "creative editing" capabilities. Specifically, the first column contains input images provided to our method, and columns two through four depict three "creatively edited" variations per input, produced without any manual intervention by the user. Additionally, the second set within Fig. 3(a) includes input images containing complex backgrounds, highlighting our framework's effectiveness in performing creative edits even when background elements are present.
>
> Moreover, Fig. 3(b) demonstrates the "creative generation" ability of our method. Here, the system receives only a simple textual keyword (e.g., "a car" or "a dress") without any visual input, and autonomously generates corresponding images.
> Regarding the visual quality, we kindly direct your attention to the supplementary materials. Specifically, our supplementary website (under “website” folder) provides high-resolution images you can view in more detail. Furthermore, the supplementary document (NeurIPS_2025_CREA.pdf) includes an additional 600 examples generated by our framework. We will incorporate these clarifications into the final manuscript.
>
> **Q4: In the middle group of Fig. 3, the background is often modified when it is not supposed to be touched.**
>
> We acknowledge that minor unintended background modifications can occur even with state-of-the-art image editing methods. Such artifacts typically arise due to inherent limitations of inversion mechanisms: these approaches aim to reconstruct input images in latent space, but often produce slightly imperfect inversions that do not exactly match the original images. A more appropriate comparison would therefore involve evaluating our edited outputs against inverted images. Unfortunately, due to the updated NeurIPS rebuttal guidelines, we are unable to provide external URLs or supplementary PDFs containing these comparative images at this stage. However, we will include these comparisons in the final manuscript.
>
> **Q5: During the evaluation phase, only using creativity criteria is limited to obtain a comprehensive evaluation on the generated image. Metrics that measure aspects such as background preservation, identity preservation are missing.**
>
> We appreciate the reviewer’s insightful suggestion. Our initial evaluation already includes comprehensive perceptual metrics such as LPIPS, CLIP, and DINO (see Table 1, main paper), all demonstrating that our method outperforms competing approaches in both editing and generation tasks. Specifically, the DINO metric is particularly effective for assessing background and identity preservation. Notably, our method achieves the highest DINO score, indicating that the edited images generated by our approach preserve essential details.
>
> However, as recommended, we have now incorporated additional quantitative metrics to further assess critical preservation aspects:  FID and KID as detailed below.
>
> | Method (Editing) | FID ↓      | KID ↓     |
> | ---------------- | ---------- | --------- |
> | LEDITS++         | 312.50     | 21.45     |
> | InstructPix2Pix  | 314.42     | 22.51     |
> | SDEdit           | 304.37     | 18.78     |
> | TurboEdit        | 320.53     | 24.79     |
> | RF-Inversion     | 309.73     | 21.76     |
> | **Ours**         | **294.19** | **14.02** |
>
> And for generation:
>
> | Method (Generation) | FID ↓      | KID ↓    |
> | ------------------- | ---------- | -------- |
> | SDXL                | 282.22     | 9.67     |
> | Flux                | 270.69     | 9.94     |
> | ConceptLab          | 272.97     | 6.49     |
> | **Ours**            | **248.67** | **5.94** |
>
> The newly computed metrics reinforce our prior findings, further confirming the superior performance of our proposed method.
>
> **Q6: In the user study results, the proposed method does not show a very obvious advantage compared with the other baselines.**
>
> We would like to clarify that our user studies comprised two distinct evaluation tasks with separate criteria:
>
> For the generation Task:
> - Usability (Q1): This measures how accurately the generated images represent the specified object (e.g., a cup). All methods performed similarly on this metric, as each can generate images resembling the intended object.
>
> - Creativity (Q2): This assesses the creativity of the generated images. Our method significantly outperformed the closest competitor (4.16 vs. 3.56), with the difference being statistically significant (p-value = 0.0056).
>
> For the editing Task:
> - Preservation  (Q1): Our method showed a better performance in preserving original image details during editing, though the margin over baselines is modest.
>
> - Creativity (Q2), our method achieved the highest score, though the numerical difference is smaller.  We also emphasize that other methods usually generated “colorful” edits to reflect ‘creativity’, whereas our method was able to generate a diverse range of creative outputs.
>
> **Q7: In the comparisons with the baselines, the authors simply use “creative [object]” as the prompt, which is too ambiguous to guide the generation**
>
> We appreciate the reviewer’s point regarding the prompt ambiguity. However, we clarify that the baseline methods were explicitly provided with the prompt "creative [object]" (e.g., "a creative couch") precisely to reflect our task definition - creative editing with minimal guidance. Our framework takes only [object] names (e.g. a couch) and autonomously generates creative transformations guided by underlying creativity principles. To further substantiate fairness, we conducted extensive ablations (Sec. 4.3, Fig. 6d) demonstrating our method’s robustness and effectiveness across various prompting strategies.
>
> **Q8: Does CREA often tend to generate imaginative objects that do not exist in the real world?**
>
> We appreciate the reviewer’s question about the realism and applicability of CREA-generated outputs. Indeed, our framework intentionally emphasizes creative and imaginative transformations, however CREA is not limited exclusively to imaginative outputs. The underlying agentic pipeline is designed flexibly, allowing users to explicitly guide the degree of creativity or realism desired.
>
> As demonstrated in Figure 3(a), CREA can reimagine familiar objects (like a dress or a table) in novel yet plausible ways. Also, CREA includes a user-in-the-loop mechanism (Section 3.4.1, Figure 5a) that allows users to guide outputs via stylistic cues or constraints (e.g., "+skull" to steer the generation to involve skull patterns), ensuring fine-grained control when desired. This control mechanism ensures that in high-fidelity domains, like product design or marketing, realism can be preserved while still benefiting from stylistic innovation. Quantitatively, CREA balances creativity and realism well, achieving top DINO (0.744), LPIPS (0.414), and VENDI (3.70) scores, indicating structural alignment, diversity, and coherence. While we are not allowed to share more realistic edits due to NeurIPS guidelines, we will add more realistic examples to the camera ready.
>
> **Q9: I am curious about the performance of a workflow that uses GPT-4o as both the agents and image generator.**
>
> As suggested, we experimented with GPT-4o for both agents and image generator, and obtained similar results. While we are not able to include the images here due to NeurIPS guidelines, these experiments will be included in the camera ready version.

---

> ### Author Response · Authors · 2025-08-06
>
> Dear reviewer,
>
> We appreciate your valuable review, and we addressed all your concerns in our rebuttal. As the NeurIPS discussion phase will conclude soon, we kindly remind you that if you have any further questions or require additional clarifications on our paper, we are more than happy to address them promptly.
>
> Best,
>
> Authors of CREA

---

> > ### Comment · Reviewer_LYr2 · 2025-08-08
> > **Update on review**
> >
> > Thank you for the very detailed rebuttal.
> > Here're my additional comments:
> >
> > - Replying to Q2: thanks for the clarification. It would be very helpful if the additional results conditioned on segmentation maps, edge maps can be shown in the revised version.
> > - Replying to Q5: thanks for providing comparisons of more metrics.
> > - The images shown in the papers seem too simple. I'm curious about image editing on images with multiple foreground objects. Will they all be changed/edited?
> > - The application scope is still limited. Most prompts shown are `a creative [object]`. The generation ability on more complicated prompts is still not clear. I would suggest adding more visual results using the prompts from benchmarks such as GenAI-Bench.
> >
> > Based on above and the common concerns from the other reviewers (e.g., unfair comparison, limited editing ability), I will keep my current rating.

---

> ### Author Response · Authors · 2025-08-08
>
> We are glad to hear that the reviewer found our rebuttal detailed, as we devoted substantial time to addressing all concerns as thoroughly as possible. We also recognize that some points in the most recent review may stem from misunderstandings about our framework, which we hope to clarify below.
>
> ***On “Most prompts shown are a creative < object >”***
>
> ***We respectfully note that this is not correct***. The `creative < object >`  format was only used for baselines for comparison. Since these baselines lack mechanisms to generate creative objects in a principled way, we followed prior state-of-the-art works (e.g., [1]) in adopting this simplified prompting strategy for them.
>
> In contrast, our method’s prompts are *highly detailed and autonomously generated by our agents according to creativity principles*. This is the core contribution of our framework; rather than relying on simple phrases such as `creative < object >`, our system generates its own detailed, creativity-guided transformations. Evidence of this can be seen in Fig. 5 of our Appendix, which shows examples of the exact prompts our framework produces. Quoting the exact prompt in Fig 5  (Appendix) generated by our framework below:
>
> ~~~
> Picture a luxurious couch crafted from an intricate blend of materials—velvet cushions that morph into soft clouds gently floating above, juxtaposed with richly textured marble armrests. This central piece, hovering against a pure white background, holds the eye with its elegant curves and vibrant emerald vines that cascade over its form, ending in delicate cherry blossom petals. Unexpectedly, geometric kaleidoscopes can be glimpsed within the cushions, offering a changeable pattern that weaves in and out of perception.
> ~~~
>
> This prompt was generated entirely by our multi-agent framework in accordance with creativity principles; something that would otherwise require substantial time, effort, and domain expertise from a human user. We acknowledge that this distinction could have been made clearer in the main paper and will revise the manuscript to explicitly include the exact prompts used.
>
> [1]: Han et.al. 2025 "Enhancing Creative Generation on Stable Diffusion-based Models" (CVPR'25)
>
>
> ***On “Unfair Comparison”***
>
> We ***respectfully disagree***. Our work is the *first to define creative editing as a research problem* explicitly guided by creativity principles. Baselines ***do not offer capabilities beyond the `creative < object >` format***. To ensure fairness, we used ***exactly the same established practices in creative image generation literature*** ([1] from CVPR'25) when designing baseline prompts.
>
> Furthermore, we conducted comprehensive ablations to test alternative prompting strategies for baselines, including `rare < object >`, `innovative < object >`, `unusual < object >`, and negative prompts such as a `normal < object >` to steer generation toward more creative outputs. These results are presented in Fig. 6(d) of the main paper and Fig. 3 in the Appendix.
>
> *While our comparison protocol follows established state-of-the-art practices, we welcome the reviewer’s suggestions for alternative comparison strategies and would be happy to explore them.*
>
> [1]: Han et.al. 2025 "Enhancing Creative Generation on Stable Diffusion-based Models" (CVPR'25)
>
>
> ***On “Limited Editing Capability”***
>
> Our framework is fully capable of editing complex, multi-object scenes (e.g., a realistic living room containing numerous distinct objects). While NeurIPS guidelines prevent us from sharing new images in the rebuttal, we conducted additional experiments on such complex scenes (for a similar request done by Reviewer  mq9P earlier).
>
> Specifically, we ran a user study with 39 participants evaluating CREA’s disentangled editing across multiple edit types (style, color change, object addition, removal, and creative transformations). Results showed overwhelming agreement that CREA preserved global structure while making targeted edits: 97.4% rated color changes as highly disentangled, 100% rated object addition, removal, and creative transformations as successful without undesired changes to other elements.  We will include these quantitative results, visual examples, and GenAI-Bench comparisons in the final manuscript to demonstrate the framework’s versatility.
>
> That said, we also wish to emphasize that the ***main contribution of our work is not to serve as a universal editing framework for all types of edits***, but to establish the ***first creativity-driven editing framework in the literature***. We provided over ***600 visual results*** in the supplementary materials to illustrate its range and fidelity.  We hope the reviewer will take this extensive evidence into account, as it already demonstrates the breadth of our framework’s capabilities, and we are prepared to expand these results in the final version with the more complex transformations created during the rebuttal period.

---

### Note · Authors · 2025-08-11

We sincerely thank the reviewers for their time, effort, and constructive feedback. During the rebuttal and discussion phases, we carefully addressed all ***38 individual questions and concerns*** raised across the five reviews, providing detailed clarifications, additional experiments, and expanded analyses. We were pleased to engage in ***productive exchanges with two reviewers*** during the discussion period (and we kindly remind these reviewers to check our ***follow-up responses for further clarifications***). For the remaining ***three reviewers***, we understand that time constraints and other commitments may have ***limited their ability to respond to our rebuttal***; nonetheless, we would be grateful if they could ***review our clarifications and new results when possible, so these contributions can be fully considered in the final evaluation***.

CREA introduces the `first creative editing task in the literature`, supported by ***extensive experiments, ablation studies, and over 600 visual examples*** in the supplementary material demonstrating its ability to produce novel, diverse, and creativity-guided visualizations. We showcase the performance of our multi-agent framework ***not only on images, but also on videos, personalization tasks, and human-in-the-loop generation***. The valuable feedback from reviewers enabled us to add further experiments, expanded baseline comparisons, and detailed clarifications. We agree that some agent-specific details should be expanded (kindly note that full details are provided under code/crea/agents in the supplementary material for transparency) and we will expand these details as well as incorporating all the feedback into the final manuscript.

We believe this paper is particularly ***timely and relevant for NeurIPS***, as multi-agent reasoning for computer vision is an ***emerging field with significant potential*** for shaping next-generation creative systems. CREA moves this field forward by not only formalizing the new task of `creative editing` but also ***surpassing other multi-agent works in image generation*** such as GenArtist (as shown by our additional rebuttal experiments). We hope this work will ***inspire further research at the intersection of multi-agent systems and generative vision.***

---

### Decision · Program_Chairs · 2025-09-17

**Decision:**

Accept (poster)

**Comment:**

This paper received borderline accept and borderline reject scores. Reviewers appreciated the new collaborative agentic framework to enhance creativity in the image generation process. On the other hand, multiple reviewers raised concerns related to limited technical novelty and experimental evaluations. AC carefully looked at the reviews and the author response. Authors addressed several of the concerns in the rebuttal. Two reviewers raised their scores after the rebuttal. It is felt that the strengths outweigh the weaknesses. The reviewers did raise some valuable concerns that should be addressed in the final camera-ready version of the paper, which include adding the relevant rebuttal discussions in the main paper. The authors are encouraged to make the necessary changes to the best of their ability.